# A REPRESENTATION-LEARNING GAME FOR CLASSES OF PREDICTION TASKS

**Neria Uzan**
Technion - Israel Institute of Technology
neriauzan@gmail.com

**Nir Weinberger**
Technion - Israel Institute of Technology
nirwein@technion.ac.il

## ABSTRACT

We propose a game-based formulation for learning dimensionality-reducing representations of feature vectors, when only a prior knowledge on future prediction tasks is available. In this game, the first player chooses a representation, and then the second player adversarially chooses a prediction task from a given class, representing the prior knowledge. The first player aims to minimize, and the second player to maximize, the *regret*: The minimal prediction loss using the representation, compared to the same loss using the original features. We consider the canonical setting in which the representation, the response to predict and the predictors are all linear functions, and the loss function is the mean squared error. We derive the theoretically optimal representation in pure strategies, which shows the effectiveness of the prior knowledge, and the optimal regret in mixed strategies, which shows the usefulness of randomizing the representation. For general representation, prediction and loss functions, we propose an efficient algorithm to optimize a randomized representation. The algorithm only requires the gradients of the loss function, and is based on incrementally adding a representation rule to a mixture of such rules.

## 1 INTRODUCTION

Commonly, data of unlabeled feature vectors $\{\boldsymbol{x}_i\} \subset \mathcal{X}$ is collected without a knowledge of the *specific* downstream prediction task it will be used for. When a prediction task becomes of interest, labels $\boldsymbol{y}_i \in \mathcal{Y}$ are also collected, and a learning algorithm is trained on $\{(\boldsymbol{x}_i, \boldsymbol{y}_i)\}$. Modern sources, such as high-definition images or genomic sequences, have high dimensionality, and this necessitates *dimensionality-reduction*, either for better generalization (Goodfellow et al., 2016), for storage/communication savings (Tsitsiklis, 1989; Nguyen et al., 2009; Duchi et al., 2018), or for interpretability (Schapire and Freund, 2012). The goal is thus to find a low-dimensional *representation* $\boldsymbol{z} = R(\boldsymbol{x}) \in \mathbb{R}^r$, that preserves the relevant part of the features, without a full knowledge of the downstream prediction task. In this paper, we propose a game-theoretic framework for this goal, by assuming that the learner has prior knowledge on the *class* of downstream prediction tasks.

Unsupervised methods for dimensionality reduction, such as *principal component analysis* (PCA) (Pearson, 1901; Jolliffe, 2005; Cunningham and Ghahramani, 2015; Johnstone and Paul, 2018), and non-linear extensions such as kernel PCA (Schölkopf et al., 1998) and *auto-encoders* (AE) (Kramer, 1991; Hinton and Salakhutdinov, 2006; Lee et al., 2011; Goodfellow et al., 2016), aim that the representation $\boldsymbol{z}$ will maximally preserve the *variation* in $\boldsymbol{x}$, and thus ignore any prior knowledge on future prediction tasks. This prior knowledge may indicate, e.g., that highly varying directions in the feature space are, in fact, irrelevant for downstream prediction tasks. From the supervised learning perspective, the *information bottleneck* (IB) principle (Tishby et al., 2000; Chechik et al., 2003; Slonim et al., 2006; Harremoës and Tishby, 2007) was used to postulate that efficient supervised learning necessitates representations that are both low-complexity and relevant (Tishby and Zaslavsky, 2015; Shwartz-Ziv and Tishby, 2017; Shwartz-Ziv, 2022; Achille and Soatto, 2018a;b) (see Appendix A). To corroborate this claim, Dubois et al. (2020) proposed a game-theoretic formulation (based on a notion of *usable information*, introduced by Xu et al. (2020)), in which Alice selects a prediction problem of $\boldsymbol{y}$ given $\boldsymbol{x}$, and then Bob selects the representation $\boldsymbol{z}$. The pay-off is the minimal risk possible using this representation. Dubois et al. (2020) showed that ideal generalization is obtained for representations that solve the resulting *decodable IB problem*.

In this paper, we assume the learner collected unlabeled feature vectors, and has a prior knowledge that the downstream prediction problem belongs to a class $\mathcal{F}$ of response functions. We propose a game different from that of Dubois et al. (2020): First, the *representation player* chooses a rule $R$ to obtain $\boldsymbol{z} = R(\boldsymbol{x}) \in \mathbb{R}^r$. Second, the *response function player* chooses $\boldsymbol{y} = f(\boldsymbol{x})$ where $f \in \mathcal{F}$ is possibly random. The payoff for $(R, f)$ is the *regret*: The minimal prediction loss of $\boldsymbol{y}$ based on $\boldsymbol{z}$ compared to the minimal prediction loss based on $\boldsymbol{x}$. The goal of the representation player (resp. response function player) is to minimize (resp. maximize) the payoff, and the output of this game is the saddle-point representation. Compared to Dubois et al. (2020), the representation is chosen based only on the *class* of response functions, rather than a specific one. The *minimax regret* measures the worst-case regret (over functions in $\mathcal{F}$) of a learner that uses the saddle-point representation. We derive the minimax representation both in *pure strategies* and in *mixed strategies* (Owen, 2013). Mixed strategies use *randomized* representation rules, whose utilization is illustrated as follows: Assume that routinely collected images are required to be compressed. There are two compression (representation) methods. The first smoothens the images, and the second preserves their edges; exclusively using the first method would hinder any possibility of making predictions reliant on the image's edges. This is prevented by randomly alternating the use of both methods.

The class $\mathcal{F}$ manifests the prior knowledge on the downstream prediction tasks that will use the represented features, and may stem from domain specific considerations; imposed by privacy or fairness constraints; or emerge from transfer or continual learning settings; see Appendix B for an extended discussion. The resulting formulation encompasses an entire spectrum of possibilities: (1) *Supervised learning*: $\mathcal{F} = \{f\}$ is a singleton, and thus known to the learner. (2) *Multitask learning* (Baxter, 2000; Maurer et al., 2016; Tripuraneni et al., 2020; 2021): $\mathcal{F} = \{f_1, f_2, \cdots, f_t\}$ is a finite set of functions. (3) *Prior expert knowledge*: $\mathcal{F}$ represents a *continuous, yet restricted* set of functions. For example, the response functions in $\mathcal{F}$ may be more sensitive to certain features than others. (4) *No supervision*: $\mathcal{F}$ is the class of *all* possible response functions, which is essentially an unsupervised learning problem, since no valuable information is known. We focus on the intermediate regimes of partial supervision, in which the learner should (and can) optimize its representation to be jointly efficient for all tasks in $\mathcal{F}$. In this respect, multitask learning (case 2) is a generalization of supervised learning, since the learner can simulate the $t$ functions in $\mathcal{F}$. Prior knowledge (case 3) is more similar to unsupervised learning, since the learner will not be able to efficiently do so.

**Theoretical contribution** We address the fundamental setting in which the representation, the response, and the prediction are all linear functions, under the mean squared error (MSE) loss (Section 3). The class is $\mathcal{F}_S = \{\|f\|_S \le 1\}$ for a known symmetric matrix $S$. Combined with the covariance matrix of the features, they both determine the relevant directions of the function in the feature space, in contrast to just the features variability, as in standard unsupervised learning. We establish the optimal representation and regret in pure strategies, which shows the utility of the prior information, and in mixed strategies, which shows that randomizing the representation yields *strictly lower* regret. We prove that randomizing between merely $\ell^*$ different representation rules suffices, where $r + 1 \le \ell^* \le d$ is a precisely characterized *effective dimension*.

**Algorithmic contribution** We develop an algorithm for optimizing mixed representations (Section 4) for general representations/response/predictors and loss functions, based only on their gradients. Similarly to boosting (Schapire and Freund, 2012), the algorithm operates incrementally. At each iteration it finds the response function in $\mathcal{F}$ that is most poorly predicted by the current mixture of representation rules. An additional representation rule is added to the mixture, based on this function and the ones from previous iterations. The functions generated by the algorithm can be considered a "self-defined signals", similarly to *self-supervised learning* (Oord et al., 2018; Shwartz-Ziv and LeCun, 2023). To optimize the weights of the representation, the algorithm solves a two-player game using the classic multiplicative weights update (MWU) algorithm (Freund and Schapire, 1999) (a follow-the-regularized-leader algorithm (Shalev-Shwartz, 2012; Hazan, 2016)).

**Related work** In Appendix A we discuss: The IB principle and compare the game of Dubois et al. (2020) with ours; The generalization error bounds of Maurer et al. (2016); Tripuraneni et al. (2020) for multi-task learning and learning-to-learn, and how our regret bound complements these results; The use of randomization in representation learning, similarly to our mixed strategies solution; Iterative algorithms for solving minimax games, and specifically the incremental algorithm approach for learning mixture models (Schapire and Freund, 2012; Tolstikhin et al., 2017), which we adopt.

## 2 PROBLEM FORMULATION

Notation conventions are mostly standard, and are detailed in Appendix C. Specifically, the eigenvalues of $S \in \mathbb{S}_+^d$ are denoted as $\lambda_{\max}(S) \equiv \lambda_1(S) \geq \cdots \geq \lambda_d(S) = \lambda_{\min}(S)$ and $v_i(S)$ denotes an eigenvector corresponding to $\lambda_i(S)$ such that $V = V(S) := [v_1(S), v_2(S), \cdots, v_d(S)] \in \mathbb{R}^{d \times d}$ and $S = V(S)\Lambda(S)V^\top(S)$ is an eigenvalue decomposition. $W_{i:j} := [w_i, \ldots, w_j] \in \mathbb{R}^{(j-i+1) \times d}$ is the matrix comprised of the columns of $W \in \mathbb{R}^{d \times d}$ indexed by $\{i, \ldots, j\}$. The probability law of a random variable $x$ is denoted as $\mathsf{L}(x)$.

Let $x \in \mathcal{X}$ be a random feature vector, with probability law $P_x := \mathsf{L}(x)$. Let $y \in \mathcal{Y}$ be a corresponding response drawn according to a probability kernel $y \sim f(\cdot \mid x = x)$, where for brevity, we will refer to $f$ as the *response function* (which can be *random*). It is known that $f \in \mathcal{F}$ for some given class $\mathcal{F}$. Let $z := R(x) \in \mathbb{R}^r$ be an $r$-dimensional representation of $x$ where $R: \mathcal{X} \to \mathbb{R}^r$ is chosen from a class $\mathcal{R}$ of representation functions, and let $Q: \mathcal{X} \to \mathcal{Y}$ be a prediction rule from a class $\mathcal{Q}_\mathcal{X}$, with the loss function $\mathsf{loss}: \mathcal{Y} \times \mathcal{Y} \to \mathbb{R}_+$. The pointwise *regret* of $(R, f)$ is

$$\mathsf{regret}(R, f \mid P_x) := \min_{Q \in \mathcal{Q}_{\mathbb{R}^r}} \mathbb{E}\left[\mathsf{loss}(y, Q(R(x)))\right] - \min_{Q \in \mathcal{Q}_\mathcal{X}} \mathbb{E}\left[\mathsf{loss}(y, Q(x))\right]. \tag{1}$$

The *minimax regret in mixed strategies* is the regret of the worst case response function in $\mathcal{F}$ to an optimal random representation, given by

$$\mathsf{regret}_{\mathsf{mix}}(\mathcal{R}, \mathcal{F} \mid P_x) := \min_{\mathsf{L}(\boldsymbol{R}) \in \mathcal{P}(\mathcal{R})} \max_{f \in \mathcal{F}} \mathbb{E}\left[\mathsf{regret}(\boldsymbol{R}, f \mid P_x)\right], \tag{2}$$

where $\mathcal{P}(\mathcal{R})$ is a set of probability measures on the possible set of representations $\mathcal{R}$. The *minimax regret in pure strategies* restricts $\mathcal{P}(\mathcal{R})$ to degenerated measures (deterministic), and so the expectation in (2) is removed. Our main goal is to determine the optimal representation strategy, either in pure $R^* \in \mathcal{R}$ or mixed strategies $\mathsf{L}(\boldsymbol{R}^*) \in \mathcal{P}(\mathcal{R})$. To this end, we will also utilize the *maximin* version of (2). Specifically, let $\mathcal{P}(\mathcal{F})$ denote a set of probability measures supported on $\mathcal{F}$, and assume that for any $R \in \mathcal{R}$, there exists a measure in $\mathcal{P}(\mathcal{R})$ that puts all its mass on $R$. Then, the *minimax theorem* (Owen, 2013, Chapter 2.4) (Sion, 1958) implies that

$$\mathsf{regret}_{\mathsf{mix}}(\mathcal{R}, \mathcal{F} \mid P_x) = \max_{\mathsf{L}(\boldsymbol{f}) \in \mathcal{P}(\mathcal{F})} \min_{R \in \mathcal{R}} \mathbb{E}\left[\mathsf{regret}(R, \boldsymbol{f} \mid P_x)\right]. \tag{3}$$

The right-hand side of (3) is the *maximin regret in mixed strategies,* and the maximizing probability law $\mathsf{L}(\boldsymbol{f}^*)$ is known as the *least favorable prior*. In general, $\mathsf{regret}_{\mathsf{mix}}(\mathcal{R}, \mathcal{F} \mid P_x) \leq \mathsf{regret}_{\mathsf{pure}}(\mathcal{R}, \mathcal{F} \mid P_x)$, and the inequality can be strict. We focus on the representation aspect, and thus assumed that $P_x$ is known and that sufficient labeled data will be provided to the learner from the subsequent prediction task. We also note that, as common in game-theory, the mixed minimax regret is achieved for *repeating* representation games (Owen, 2013), which fits the scenario in which routinely collected data is to be represented. By contrast, the pure minimax regret guarantee is valid for a single representation, and thus more conservative from this aspect.

## 3 THE LINEAR SETTING UNDER MSE LOSS

In this section, we focus on linear classes and the MSE loss function. The response function class $\mathcal{F}$ is characterized by a quadratic constraint specified by a matrix $S \in \mathbb{S}_{++}^d$, which represents the relative importance of each direction in the feature space in determining $y$.

**Definition 1** (The linear MSE setting)**.** Assume that $\mathcal{X} = \mathbb{R}^d$, that $\mathcal{Y} = \mathbb{R}$ and a squared error loss function $\mathsf{loss}(y_1, y_2) = |y_1 - y_2|^2$. Assume that $\mathbb{E}[x] = 0$ and let $\Sigma_x := \mathbb{E}[xx^T] \in \mathbb{S}_{++}^d$ be its invertible covariance matrix. The classes of representations, response functions, and predictors are all linear, that is: (1) The representation is $z = R(x) = R^\top x$ for $R \in \mathcal{R} := \mathbb{R}^{d \times r}$ where $d > r$; (2) The response function is $f \in \mathcal{F} \subset \mathbb{R}^d$, and $y = f^\top x + n \in \mathbb{R}$, where $n \in \mathbb{R}$ is a heteroscedastic noise that satisfies $\mathbb{E}[n \mid x] = 0$, and given some specified $S \in \mathbb{S}_{++}^d$, it holds that

$$f \in \mathcal{F}_S := \left\{ f \in \mathbb{R}^d : \|f\|_S^2 \leq 1 \right\}, \tag{4}$$

where $\|f\|_S := \|S^{-1/2}f\|_2 = (f^\top S^{-1} f)^{1/2}$ is the Mahalanobis norm; (3) The predictor is $Q(z) = q^\top z \in \mathbb{R}$ for $q \in \mathbb{R}^r$. Since the regret will depend on $P_x$ only via $\Sigma_x$, we will abbreviate the notation of the pure (resp. mixed) minimax regret to $\mathsf{regret}_{\mathsf{pure}}(\mathcal{F} \mid \Sigma_x)$ (resp. $\mathsf{regret}_{\mathsf{mix}}(\mathcal{F} \mid \Sigma_x)$).

In Appendix E.1 we show that standard PCA can be similarly formulated, by assuming that $\mathcal{F}$ is a singleton containing the noiseless identity function, so that $\boldsymbol{y} = \boldsymbol{x}$ surely holds, and $\hat{x} = Q(z) \in \mathbb{R}^d$. Proposition 15 therein shows that the pure and mixed minimax representations are both $R = V_{1:r}(\Sigma_{\boldsymbol{x}})$, and so randomization is unnecessary. We begin with the pure minimax regret.

**Theorem 2.** *For the linear MSE setting (Definition 1)*

$$\mathsf{regret}_{\mathsf{pure}}(\mathcal{F}_S \mid \Sigma_{\boldsymbol{x}}) = \lambda_{r+1}\left(\Sigma_{\boldsymbol{x}}^{1/2} S \Sigma_{\boldsymbol{x}}^{1/2}\right). \tag{5}$$

*A minimax representation matrix is*

$$R^* := \Sigma_{\boldsymbol{x}}^{-1/2} \cdot V_{1:r}\left(\Sigma_{\boldsymbol{x}}^{1/2} S \Sigma_{\boldsymbol{x}}^{1/2}\right), \tag{6}$$

*and the worst case response function is*

$$f^* := S^{1/2} \cdot v_{r+1}\left(\Sigma_{\boldsymbol{x}}^{1/2} S \Sigma_{\boldsymbol{x}}^{1/2}\right). \tag{7}$$

The optimal representation thus whitens the feature vector $\boldsymbol{x}$, and then projects it on the top $r$ eigenvectors of the adjusted covariance matrix $\Sigma_{\boldsymbol{x}}^{1/2} S \Sigma_{\boldsymbol{x}}^{1/2}$, which reflects the prior knowledge that $f \in \mathcal{F}_S$. The proof in Appendix E.2 has the following outline: Plugging the optimal predictor into the regret results a quadratic form in $f \in \mathbb{R}^d$, determined by a matrix which depends on the subspace spanned by the representation $R$. The worst-case $f$ is the determined via the *Rayleigh quotient theorem*, and the optimal $R$ is found via the *Courant–Fischer variational characterization* (see Appendix D for a summary of these results). We next consider the mixed minimax regret, which turns out to be more involved:

**Theorem 3.** *For the linear MSE setting (Definition 1), let $\lambda_i \equiv \lambda_i(S^{1/2}\Sigma_{\boldsymbol{x}}S^{1/2})$ for $i \in [d]$ and $\lambda_{d+1} \equiv 0$, and let $\ell^*$ be any member of*

$$\left\{\ell \in [d]\backslash[r]: (\ell - r)\cdot\lambda_\ell^{-1} \leq \sum_{i=1}^{\ell}\lambda_i^{-1} \leq (\ell - r)\cdot\lambda_{\ell+1}^{-1}\right\}. \tag{8}$$

- *The minimax regret in mixed strategies is*

$$\mathsf{regret}_{\mathsf{mix}}(\mathcal{F}_S \mid \Sigma_{\boldsymbol{x}}) = \frac{\ell^* - r}{\sum_{i=1}^{\ell^*}\lambda_i^{-1}}. \tag{9}$$

- *The covariance matrix of the least favorable prior of $\boldsymbol{f}$ is*

$$\Sigma_{\boldsymbol{f}}^* := \frac{V^\top \Lambda_{\ell^*}^{-1} V}{\sum_{i=1}^{\ell^*}\lambda_i^{-1}}. \tag{10}$$

*where $\Lambda_\ell := \mathrm{diag}(\lambda_1, \ldots, \lambda_{\ell^*}, 0, \cdots, 0)$, and $V \equiv V(S^{1/2}\Sigma_{\boldsymbol{x}}S^{1/2})$.*

- *The probability law of the minimax representation: Let $\overline{A} \in \{0,1\}^{\ell^* \times \binom{\ell^*}{r}}$ be a matrix whose columns are the members of the set $\overline{\mathcal{A}} := \{\overline{a} \in \{0,1\}^{\ell^*}: \|\overline{a}\|_1 = \ell^* - r\}$ (in some order). Let $\overline{b} = (b_1, \ldots, b_{\ell^*})^\top$ be such that*

$$b_i = (\ell^* - r) \cdot \frac{\lambda_i^{-1}}{\sum_{j=1}^{\ell^*}\lambda_j^{-1}}. \tag{11}$$

*Then, there exists a solution $p \in [0,1]^{\binom{\ell^*}{r}}$ to $\overline{A}p = \overline{b}$ with support size at most $\ell^* + 1$. For $j \in [\binom{\ell^*}{r}]$, let $\mathcal{I}_j := \{i \in [\ell^*]: \overline{A}_{ij} = 0\}$ be the zero indices on the $j$th column of $\overline{A}$, and let $V_{\mathcal{I}_j}$ denote the $r$ columns of $V$ whose index is in $\mathcal{I}_j$. A minimax representation is $\boldsymbol{R}^* = \Sigma_{\boldsymbol{x}}^{-1/2}V_{\mathcal{I}_j}$ with probability $p_j$, for $j \in [\binom{\ell^*}{r}]$.*

The proof of Theorem 3 is also in Appendix E.2, and is substantially more complicated than for the pure regret. The reason is that directly maximizing over $\mathsf{L}(\boldsymbol{R})$ is challenging, and so we take a two-step indirect approach. The outline is as follows: First, we solve the *maximin problem* (3), and

find the least favorable prior $\mathsf{L}(\boldsymbol{f}^*)$. Second, we propose a probability law for the representation $\mathsf{L}(\boldsymbol{R})$, and show that its regret equals the maximin value, and thus also the minimax. With more detail, in the first step, we show that the regret only depends on $\mathsf{L}(\boldsymbol{f})$ via $\Sigma_{\boldsymbol{f}} = \mathbb{E}[\boldsymbol{f}\boldsymbol{f}^\top]$, and we explicitly construct $\mathsf{L}(\boldsymbol{f})$ that is supported on $\mathcal{F}_S$ and has this covariance matrix. This reduces the problem from optimizing $\mathsf{L}(\boldsymbol{f})$ to optimizing $\Sigma_{\boldsymbol{f}}$, whose solution (Lemma 17) results the least favorable $\Sigma_{\boldsymbol{f}}^*$, and then the maximin value. In the second step, we construct a representation that achieves the maximin regret. Concretely, we construct representation matrices that use $r$ of the $\ell^*$ principal components of $\Sigma_{\boldsymbol{x}}^{1/2} S \Sigma_{\boldsymbol{x}}^{1/2}$, where $\ell^* > r$. The defining property of $\ell^*$ (8), established in the maximin solution, is utilized to find proper weights on the $\binom{\ell^*}{r}$ possible representations, which achieve the maximin solution, and thus also the minimax. The proof then uses Carathéodory's theorem (see Appendix D) to establish that the optimal $\{p_j\}$ is supported on at most $\ell^* + 1$ matrices, much less than the potential support size of $\binom{\ell^*}{r}$. We next make a few comments:

1. Computing the mixed minimax probability: This requires solving $\overline{A}^\top p = \overline{b}$ for a probability vector $p$. This is a linear-program feasibility problem, which is routinely solved (Bertsimas and Tsitsiklis, 1997). For illustration, if $r = 1$ then $p_j = 1 - (\ell^* - 1)\lambda_j^{-1}/(\sum_{i=1}^{\ell^*} \lambda_i^{-1})$ for $j \in [\ell^*]$, and if $\ell^* = r + 1$ then $p_j = (\lambda_j^{-1})/(\sum_{j'=1}^{\ell^*} \lambda_{j'}^{-1})$ on the first $\ell^*$ standard basis vectors. Nonetheless, the dimension of $p$ is $\binom{\ell^*}{r}$ and thus increases fast as $\Theta((\ell^*)^r)$, which is infeasible for high dimensions. In this case, the algorithm of Section 4 can be used. As we empirically show it approximately achieves the optimal regret, with slightly more than $\ell^* + 1$ atoms (see Example 6 henceforth).

2. The rank of $\Sigma_{\boldsymbol{f}}^*$: The rank of the covariance matrix of the least favorable prior is an *effective dimension*, satisfying (see (9))

$$\ell^* = \operatorname*{argmax}_{\ell \in [d] \setminus [r]} \frac{1 - (r/\ell)}{\frac{1}{\ell} \sum_{i=1}^{\ell} \lambda_i^{-1}}. \tag{12}$$

By convention, $\{\lambda_i^{-1}\}_{i \in [d]}$ is a monotonic non-decreasing sequence, and so is the partial Cesàro mean $\psi(\ell) := \frac{1}{\ell} \sum_{i=1}^{\ell} \lambda_i^{-1}$. For example, if $\lambda_i = i^{-\alpha}$ with $\alpha > 0$ then $\psi(\ell) = \Theta(\ell^\alpha)$. So, if, e.g., $\psi(\ell) = \ell^\alpha$, then it is easily derived that $\ell^* \approx \min\{\frac{\alpha+1}{\alpha} r, d\}$. Hence, if $\alpha \geq \frac{r}{d-r}$ is large enough and the decay rate of $\{\lambda_i\}$ is fast enough then $\ell^* < d$, but otherwise $\ell^* = d$. As the decay rate of $\{\lambda_i\}$ becomes faster, the rank of $\Sigma_{\boldsymbol{f}}^*$ decreases from $d$ to $r + 1$. Importantly, $\ell^* \geq r + 1$ always holds, and so the optimal mixed representation is not deterministic even if $S^{1/2} \Sigma_{\boldsymbol{x}} S^{1/2}$ has less than $r$ significant eigenvalues (which can be represented by a single matrix $R \in \mathbb{R}^{d \times r}$). Hence, the mixed minimax regret is always *strictly lower* than the pure minimax regret. Thus, even when $S = I_d$, and no valuable prior knowledge is known on the response function, the mixed minimax representation is different from the standard PCA solution of top $r$ eigenvectors of $\Sigma_{\boldsymbol{x}}$.

3. Uniqueness of the optimal representation: Since one can always post-multiply $R^\top x$ by some invertible matrix, and then pre-multiply $z = R^\top x$ by its inverse, the following holds: If $\mathcal{R}$ and $\mathcal{Q}$ are not further restricted, then if $\boldsymbol{R}$ is a minimax representation, and $W(\boldsymbol{R}) \in \mathbb{R}^{r \times r}$ is an invertible matrix, then $\boldsymbol{R} \cdot W(\boldsymbol{R})$ is also a minimax representation.

4. Infinite-dimensional features: Theorems 2 and 3 assume a finite dimensional feature space. We show in Appendix F that the results can be generalized to an infinite dimensional Hilbert space $\mathcal{X}$, in the more restrictive setting that the noise $\boldsymbol{n}$ is statistically independent of $\boldsymbol{x}$.

**Example 4.** Assume $S = I_d$, and denote, for brevity, $V \equiv V(\Sigma_{\boldsymbol{x}}) := [v_1, \ldots, v_d]$ and $\Lambda \equiv \Lambda(\Sigma_{\boldsymbol{x}}) := \operatorname{diag}(\lambda_1, \ldots, \lambda_d)$. The optimal minimax representation in pure strategies (Theorem 2) is

$$R^* = \Sigma_{\boldsymbol{x}}^{-1/2} \cdot V_{1:r} = V \Lambda_{\boldsymbol{x}}^{-1/2} V^\top V_{1:r} = V \Lambda_{\boldsymbol{x}}^{-1/2} \cdot [e_1, \ldots, e_r] = \left[ \lambda_1^{-1/2} \cdot v_1, \ldots, \lambda_r^{-1/2} \cdot v_r \right], \tag{13}$$

which is comprised of the top $r$ eigenvectors of $\Sigma_{\boldsymbol{x}}$, scaled so that $v_i^\top \boldsymbol{x}$ has unit variance. By Comment 3 above, $V_{1:r}$ is also an optimal minimax representation. The worst case response is $f = v_{r+1}(\Sigma_{\boldsymbol{x}})$ and, as expected, since $R$ uses the first $r$ principal directions

$$\mathsf{regret}_{\mathsf{pure}}(\mathcal{F} \mid \Sigma_{\boldsymbol{x}}) = \lambda_{r+1}. \tag{14}$$

The minimax regret in mixed strategies (Theorem 3) is different, and given by

$$\mathsf{regret}_{\mathsf{mix}}(\mathcal{F} \mid \Sigma_{\boldsymbol{x}}) = \frac{\ell^* - r}{\sum_{i=1}^{\ell^*} \lambda_i^{-1}}, \tag{15}$$

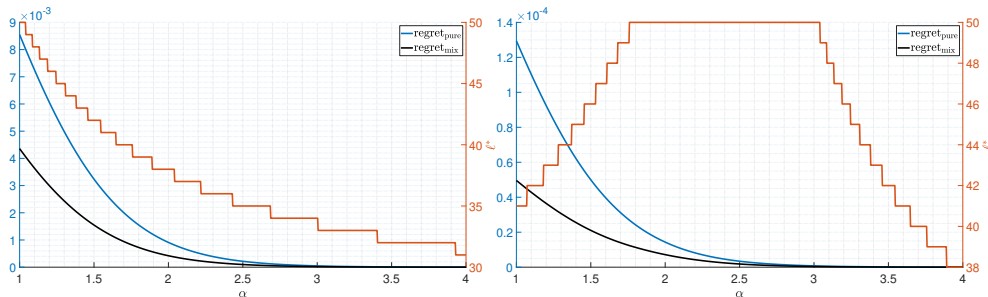

Figure 1: Left: Pure and mixed minimax regret and $\ell_*$ for Example 4, for $d = 50, r = 25$, with $\lambda_i = \sigma_i^2 \propto i^{-\alpha}$. Right: Pure and mixed minimax regret and $\ell_*$ for Example 5, for $d = 50, r = 25$, with $\sigma_i^2 \propto i^{-\alpha}$ and $s_i \propto i^2$. The trend of $\ell_*$ is reversed for $\alpha > 2$.

where $\ell^*$ is determined by the decay rate of the eigenvalues of $\Sigma_{\boldsymbol{x}}$ (see (8)). The least favorable covariance matrix is given by (Theorem 3)

$$\Sigma_{\boldsymbol{f}}^* = \left[ \sum_{i=1}^{\ell^*} \lambda_i^{-1} \right]^{-1} \cdot V \, \mathrm{diag} \left( \lambda_1^{-1}, \ldots, \lambda_{\ell^*}^{-1}, 0, \cdots, 0 \right) \cdot V^\top. \tag{16}$$

Intuitively, $\Sigma_{\boldsymbol{f}}^*$ equalizes the first $\ell^*$ eigenvalues of $\Sigma_{\boldsymbol{x}}\Sigma_{\boldsymbol{f}}^*$ (and nulls the other $d - \ell^*$), to make the representation indifferent to these $\ell^*$ directions. As evident from the regret, the "equalization" of the $i$th eigenvalue adds a term of $\lambda_i^{-1}$ to the denominator, and if $\lambda_i$ is too small then $v_i$ is not chosen for the representation. This agrees with Comment 2, which states that a fast decay of $\{\lambda_i\}$ reduces $\ell_*$ away from $d$. A derivation similar to (13) shows that the mixed minimax representation sets

$$\boldsymbol{R}^* = \Sigma_{\boldsymbol{x}}^{-1/2} \cdot V_{\mathcal{I}_j} = \left[ \lambda_{i_{j,1}}^{-1/2} \cdot v_{i_{j,1}}, \ldots, \lambda_{i_{j,r}}^{-1/2} \cdot v_{i_{j,r}} \right] \tag{17}$$

with probability $p_j$, where $\mathcal{I}_j \equiv \{i_{j,1}, \ldots, i_{j,r}\}$. Thus, the optimal representation chooses a random subset of $r$ vectors from $\{v_1, \ldots, v_{\ell^*}\}$. See the left panel of Figure 1 for a numerical example.

**Example 5.** To demonstrate the effect of prior knowledge on the response function, we assume $\Sigma_{\boldsymbol{x}} = \mathrm{diag}(\sigma_1^2, \ldots, \sigma_d^2)$ and $S = \mathrm{diag}(s_1, \ldots, s_d)$, where $\sigma_1^2 \geq \sigma_2^2 \geq \cdots \geq \sigma_d^2$ (but $\{s_i\}_{i \in [d]}$ are not necessarily ordered). Letting $f = (f_1, \ldots, f_d)$, the class of response functions is $\mathcal{F}_S := \{f \in \mathbb{R}^d : \sum_{i=1}^d (f_i^2/s_i) \leq 1\}$, and so coordinates $i \in [d]$ with a large $s_i$ have large influence on the response. Let $(i_{(1)}, \ldots, i_{(d)})$ be a permutation of $[d]$ so that $\sigma_{i(j)}^2 s_{i(j)}$ it the $j$th largest value of $(\sigma_i^2 s_i)_{i \in [d]}$. The pure minimax regret is (Theorem 2)

$$\mathsf{regret}_{\mathsf{pure}}(\mathcal{F} \mid \Sigma_{\boldsymbol{x}}) = \sigma_{i_{r+1}}^2 s_{i_{r+1}}. \tag{18}$$

The optimal representation is $R = [e_{i_{(1)}}, e_{i_{(2)}}, \ldots, e_{i_{(r)}}]$, that is, uses the most influential coordinates, according to $\{s_i\}$, which may be different from the $r$ principal directions of $\Sigma_{\boldsymbol{x}}$. For the minimax regret in mixed strategies, Theorem 3 results

$$\mathsf{regret}_{\mathsf{mix}}(\mathcal{F} \mid \Sigma_{\boldsymbol{x}}) = \frac{\ell^* - r}{\sum_{j=1}^{\ell^*} (s_{i_j} \sigma_{i_j}^2)^{-1}} \tag{19}$$

for $\ell^* \in [d] \backslash [r]$ satisfying (8), and the covariance matrix of the least favorable prior is given by

$$\Sigma_{\boldsymbol{f}}^* = \frac{\sum_{j=1}^{\ell^*} \sigma_{i_j}^{-2} \cdot e_{i_j} e_{i_j}^\top}{\sum_{j=1}^{\ell^*} (s_{i_j} \sigma_{i_j}^2)^{-1}}. \tag{20}$$

That is, the matrix is diagonal, and the $k$th term on the diagonal is $\Sigma_{\boldsymbol{f}}^*(k,k) \propto \sigma_k^{-2}$ if $k = i_j$ for some $j \in [\ell^*]$ and $\Sigma_{\boldsymbol{f}}^*(k,k) = 0$ otherwise. As in Example 4, $\Sigma_{\boldsymbol{f}}^*$ equalizes the first $\ell^*$ eigenvalues of $\Sigma_{\boldsymbol{x}}\Sigma_{\boldsymbol{f}}$ (and nulls the other $d - \ell^*$). However, it does so in a manner that chooses them according to their influence on the response $\boldsymbol{f}^\top \boldsymbol{x}$. The minimax representation in mixed strategies is

$$\boldsymbol{R}^* = \left[ \sigma_{i_{j,1}}^{-1} \cdot e_{i_{j,1}}, \ldots, \sigma_{i_{j,r}}^{-1} \cdot e_{i_{j,r}} \right] \tag{21}$$

with probability $p_j$. Again, the first $\ell^*$ coordinates are used, and not just the top $r$. See the right panel of Figure 1 for a numerical example. Naturally, in the non-diagonal case, the minimax regret will also depend on the relative alignment between $S$ and $\Sigma_{\boldsymbol{x}}$.

## 4 AN ALGORITHM FOR GENERAL CLASSES AND LOSS FUNCTIONS

In this section, we propose an iterative algorithm for optimizing the representation in mixed strategies, i.e., solving (2) for general classes and loss functions. The algorithm will find a *finite* mixture of $m$ representations, and so we let $\boldsymbol{R} = R^{(j)} \in \mathcal{R}$ with probability $p^{(j)}$ for $j \in [m]$ (this suffices for the linear MSE setting of Section 3, but possibly sub-optimal in general). The main idea is to *incrementally* add representations $R^{(j)}$ to the mixture. Loosely speaking, the algorithm is initialized with a single representation rule $R^{(1)}$. Then, the response function in $\mathcal{F}$ which is most poorly predicted when $\boldsymbol{x}$ is represented by $R^{(1)}(\boldsymbol{x})$ is found. The added representation component $R^{(2)}$ aims to allow for accurate prediction of this function. At the next iteration, the function in $\mathcal{F}$ which is most poorly predicted by a mixed representation that randomly uses either $R^{(1)}$ or $R^{(2)}$ is found, and $R^{(3)}$ is then optimized to reduce the regret for this function, and so on.

The actual algorithm is more complicated, since it also finds proper weights $\{p^{(j)}\}_{j \in [m]}$ for the representation rules, and also randomizes the response player. Thus we set $\boldsymbol{f} = f^{(i)} \in \mathcal{F}$ with probability $o^{(i)}$ where $i \in [\overline{m}]$, and $\overline{m} = m_0 + m$ for some $m_0 \geq 0$. The resulting optimization problem then becomes

$$\min_{\{p^{(j)}, R^{(j)} \in \mathcal{R}\}} \max_{\{o^{(i)}, f^{(i)} \in \mathcal{F}\}} \sum_{j \in [m]} \sum_{i \in [\overline{m}]} p^{(j)} \cdot o^{(i)} \cdot \mathsf{regret}(R^{(j)}, f^{(i)} \mid P_{\boldsymbol{x}}), \qquad (22)$$

where $\{p^{(j)}\}_{j \in [m]}$ and $\{o^{(i)}\}_{i \in [\overline{m}]}$ are probability vectors. The ultimate goal of solving (22) is just to extract the optimal randomized representation $\boldsymbol{R}$, given by $\{(R^{(j)}, p^{(j)})\}_{j \in [m]}$.

Henceforth, the index $k \in [m]$ will denote the current number of representations. Initialization requires a representation $R^{(1)}$, as well as a *set* of functions $\{f^{(i)}\}_{i \in m_0}$, so that the final support size of $\boldsymbol{f}$ will be $\overline{m} = m_0 + m$. Finding this initial representation and the set of functions is based on the specific loss function and the set of representations/predictors (see Appendix G for examples). The algorithm has two phases for each iteration. In the first phase, a new adversarial function is added to the set of functions, as the worse function for the current random representation. In the second phase, a new representation atom is added to the set of possible representations. This representation is determined based on the given set of functions. Concretely, at iteration $k$:

• Phase 1 (Finding adversarial function): Given $k$ representations $\{R^{(j)}\}_{j \in (k)}$ with weights $\{p^{(j)}\}_{j \in [k]}$, the algorithm determines $f^{(m_0+k)}$ as the worst function, by solving

$$\mathsf{reg}_k := \mathsf{regret}_{\mathsf{mix}}(\{R^{(j)}, p^{(j)}\}_{j \in [k]}, \mathcal{F} \mid P_{\boldsymbol{x}}) := \max_{f \in \mathcal{F}} \sum_{j \in [k]} p^{(j)} \cdot \mathsf{regret}(R^{(j)}, f \mid P_{\boldsymbol{x}}), \qquad (23)$$

and $f^{(m_0+k)}$ is set to be the maximizer. This simplifies (22) in the sense that $m$ is replaced by $k$, that the random representation $\boldsymbol{R}$ is kept fixed, and that $f \in \mathcal{F}$ is optimized as a pure strategy (the previous functions $\{f^{(i)}\}_{i \in [m_0+k-1]}$ are ignored).

• Phase 2 (Adding a representation atom): Given fixed $\{f^{(j)}\}_{j \in [m_0+k]}$ and $\{R^{(j)}\}_{j \in [k]}$, a new representation $R^{(k+1)}$ is found as the most incrementally valuable representation atom, by solving

$$\min_{R^{(k+1)} \in \mathcal{R}} \mathsf{regret}_{\mathsf{mix}}(\{R^{(j_1)}\}, \{f^{(j_2)}\} \mid P_{\boldsymbol{x}}) :=$$
$$\min_{R^{(k+1)} \in \mathcal{R}} \min_{\{p^{(j_1)}\}} \max_{\{o^{(j_2)}\}} \sum_{j_1} \sum_{j_2} p^{(j_1)} \cdot o^{(j_2)} \cdot \mathsf{regret}(R^{(j_1)}, f^{(j_2)} \mid P_{\boldsymbol{x}}) \qquad (24)$$

where $j_1 \in [k + 1]$ and $j_2 \in [m_0 + k]$, the solution $R^{(k+1)}$ is added to the representations, and the weights are updated to the optimal $\{p^{(j_1)}\}$. Compared to (22), the response functions and first $k$ representations are kept fixed, and only the weights $\{p^{(j_1)}\}$ $\{o^{(j_2)}\}$ and $R^{(k+1)}$ are optimized.

The procedure is described in Algorithm 1, where, following the main loop, $m^* = \operatorname{argmin}_{k \in [m]} \mathsf{reg}_k$ representation atoms are chosen and the output is $\{R^{(j)}, p^{(j)}\}_{j \in [m^*]}$. Algorithm 1 relies on solvers for the Phase 1 (23) and Phase 2 (24) problems. In Appendix G we propose two algorithms for these problems, which are based on gradient steps for updating the adversarial response (assuming $\mathcal{F}$ is

a continuous class) and the new representation, and on the MWU algorithm (Freund and Schapire, 1999) for updating the weights. In short, the Phase 1 algorithm updates the response function $f$ via a projected gradient step of the expected loss, and then adjusts the predictors $\{Q^{(j)}\}$ to the updated response function $f$ and the current representations $\{R^{(j)}\}_{j\in[k]}$. The Phase 2 algorithm only updates the new representation $R^{(k+1)}$ via projected gradient steps, while keeping $\{R^{(j)}\}_{j\in[k]}$ fixed. Given the representations $\{R^{(j)}\}_{j\in[k+1]}$ and the functions $\{f^{(i)}\}_{i\in[m_0+k]}$, a predictor $Q^{(j,i)}$ is then fitted to each representation-function pair, which also determines the loss for this pair. The weights $\{p^{(j)}\}_{j\in[k+1]}$ and $\{o^{(i)}\}_{i\in[m_0+k]}$ are updated towards the equilibrium of the two-player game determined by the loss of the predictors $\{Q^{(j,i)}\}_{j\in[k+1],i\in[m_0+k]}$ via the MWU algorithm. In a multi-task learning setting, in which the class $\mathcal{F}$ is finite, the Phase 1 solver can be replaced by a performing a simple maximization over the functions.

---

**Algorithm 1** Solver of (22): An iterative algorithm for learning mixed representations.

1: **input** $P_{\boldsymbol{x}}, \mathcal{R}, \mathcal{F}, \mathcal{Q}, d, r, m, m_0$     ▷ Feature distribution, classes, dimensions and parameters
2: **input** $R^{(1)}, \{f^{(j)}\}_{j\in[m_0]}$     ▷ Initial representation and initial function (set)
3: **begin**
4: **for** $k = 1$ to $m$ **do**
5:     **phase 1:** $f^{(m_0+k)}$ is set by a solver of (23) and:     ▷ Solved using Algorithm 2

$$\mathsf{reg}_k \leftarrow \mathsf{regret}_{\mathsf{mix}}(\{R^{(j)}, p^{(j)}\}_{j\in[k]}, \mathcal{F} \mid P_{\boldsymbol{x}}) \tag{25}$$

6:     **phase 2:** $R^{(k+1)}, \{p_k^{(j)}\}_{j\in[k+1]}$ is set by a solver of (24)     ▷ Solved using Algorithm 3
7: **end for**
8: **set** $m^* = \operatorname{argmin}_{k\in[m]} \mathsf{reg}_k$
9: **return** $\{R^{(j)}\}_{j\in[m^*]}$ **and** $p_{m_*} = \{p_k^{(j)}\}_{j\in[m^*]}$

---

Since Algorithm 1 aims to solve a non convex-concave minimax game, deriving theoretical bounds on its convergence seems to be elusive at this point (see Appendix A for a discussion). We next describe a few experiments with the algorithm (See Appendix H for details).

**Example 6.** We validated Algorithm 1 in the linear MSE setting (Section 3), for which a closed-form solution exists. We ran Algorithm 1 on randomly drawn diagonal $\Sigma_{\boldsymbol{x}}$, and computed the ratio between the regret obtained by the algorithm to the theoretical value. The left panel of Figure 2 shows that the ratio is between $1.15 - 1.2$ in a wide range of $d$ values. Algorithm 1 is also useful for this setting since finding an $(\ell^* + 1)$-sparse solution to $\overline{A}p = \overline{b}$ is computationally difficult when $\binom{\ell^*}{r}$ is very large (e.g., in the largest dimension of the experiment $\binom{d}{r} = \binom{19}{5} = 11,628$).

Our next example pertains to a logistic regression setting, under the cross-entropy loss function.

**Definition 7** (The linear cross-entropy setting). Assume that $\mathcal{X} = \mathbb{R}^d$, that $\mathcal{Y} = \{\pm1\}$ and that $\mathbb{E}[\boldsymbol{x}] = 0$. Assume that the class of representation is linear $z = R(x) = R^\top x$ for some $R \in \mathcal{R} := \mathbb{R}^{d\times r}$ where $d > r$. Assume that a response function and a prediction rule determine the probability that $y = 1$ via logistic regression modeling, as $f(\boldsymbol{y} = \pm1 \mid x) = 1/[1 + \exp(\mp f^\top x)]$. Assume the cross-entropy loss function, where given that the prediction that $\boldsymbol{y} = 1$ with probability $q$ results the loss $\mathsf{loss}(y, q) := -\frac{1}{2}(1 + y)\log q - \frac{1}{2}(1 - y)\log(1 - q)$. The set of predictor functions is $\mathcal{Q} := \{Q(z) = 1/[1 + \exp(-q^\top z)], q \in \mathbb{R}^r\}$. As for the linear case, we assume that $f \in \mathcal{F}_S$ for some $S \in \mathbb{S}_{++}^d$. The regret is then given by the expected binary Kullback-Leibler (KL) divergence

$$\mathsf{regret}(R, f \mid P_{\boldsymbol{x}}) = \min_{q\in\mathbb{R}^r} \mathbb{E}\left[D_{\mathrm{KL}}\left([1 + \exp(-f^\top \boldsymbol{x})]^{-1} \,\|\, [1 + \exp(-q^\top R^\top \boldsymbol{x})]^{-1}\right)\right]. \tag{26}$$

**Example 8.** We drawn empirical distributions of features from an isotropic normal distribution, in the linear cross-entropy setting. We ran Algorithm 1 using the closed-form regret gradients from Appendix H. The right panel of Figure 2 shows the reduced regret obtained by increasing the support size $m$ of the random representation, and thus the effectiveness of mixed representations.

The last example compares the optimized representation with that of PCA.

**Example 9** (Comparison with PCA for multi-label classification). We constructed a dataset of images, each containing 4 shapes randomly selected from a dictionary of 6 shapes. The class $\mathcal{F}$ is

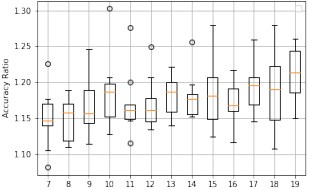 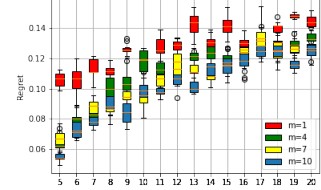

Figure 2: Results of Algorithm 1. Left: $r = 5$, varying $d$. The ratio between the regret achieved by Algorithm 1 and the theoretical regret in the linear MSE setting. Right: $r = 3$, varying $d$. The regret achieved by Algorithm 1 in the linear cross-entropy setting, various $m$.

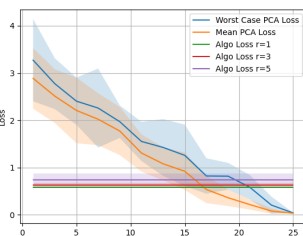 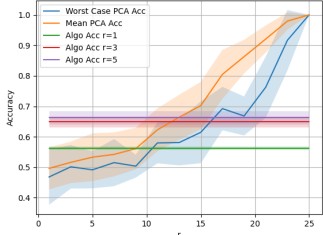

Figure 3: Results on the dataset of images. Comparison between optimized minimax representation (simplified version of Algorithm 1) vs. PCA. Worst-case function in blue, and average-case function in orange. Left: Cross entropy loss. Right: Accuracy.

finite and contains the $t = 6$ binary classification functions given by indicators for each shape. The representation is linear and the predictor is based on logistic regression. In this setting, Algorithm 1 can be simplified; See Appendix H.3 for details (specifically Definition 23 of the setting and Figure 6 for an image example). We ran the simplified version of Algorithm 1 on a dataset of 1000 images, and compared the cross-entropy loss and the accuracy of optimized representation to that of PCA on a fresh dataset of 1000 images. The results in Figure 3 show that the regret of PCA is much larger, not only for the worse-case function but also for the average-case function. For example, using the representation obtained by the algorithm with $r = 3 < t = 6$ is as good as PCA with at least $r = 12$.

## 5 CONCLUSION

We proposed a game-theoretic formulation for learning feature representations when prior knowledge on the class of downstream prediction tasks is available. Interestingly, our formulation links between the problem of finding jointly efficient representations for multiple prediction tasks, and the problem of finding saddle points of non-convex/non-concave games. The latter is a challenging problem and is under active research (see Appendix A). So, any advances in that domain can be translated to improve representation learning. Similarly to our problem, foundation models also adapt to wide range of downstream prediction tasks, however, efficient learning is achieved therein without explicit prior knowledge on the class of downstream tasks, albeit using fine-tuning. It is interesting to incorporate minimax formulations into the training procedure of foundation models, or to explain them using similar game formulations.

For future research it would be interesting: (1) To generalize the class $\mathcal{F}_S = \{f \colon \|f\|_S \leq 1\}$ used for linear functions to the class $\mathcal{F}_{S_x} := \mathbb{E}[\|\nabla_x f(x)\|^2_{S_x}] \leq 1$ for non-linear functions, where $\{S_x\}_{x \in \mathbb{R}^d}$ is now locally specified (somewhat similarly to the regularization term used in contractive AE (Rifai et al., 2011), though for different reasons). (2) To efficiently learn $S$ from previous experience, e.g., improving $S$ from one episode to another in a meta-learning setup (Hospedales et al., 2021). (3) To evaluate the effectiveness of the learned representation in our formulation, as an initialization for further optimization when labeled data is collected. One may postulate that our learned representation may serve as a *universal* initialization for such training.

ETHICS STATEMENT

The research described in this paper is foundational, and does not aim for any specific application. Nonetheless, the learned representation is based on a prior assumption on the class of response functions, and the choice of this prior may have positive or negative impacts: For example, a risk of this choice of prior is that the represented features completely ignore a viable feature for making downstream predictions. A benefit that can stem from choosing a proper prior is that the representation will null the effect of features that lead to unfair advantages for some particular group in downstream predictions. Anyhow, the results presented in the paper are indifferent to such future utilization, and any usage of these results should take into account the aforementioned possible implications.

REPRODUCIBILITY STATEMENT

For the theoretical statements, full proofs appear in Appendices E and F. In order to ease the reader, a summary of the main mathematical results used appears in Appendix D. For the experimental results, a detailed description of the algorithm and experiments appears in Appendix H. This description includes an accurate description of the Phase 1 and 2 solvers, initialization methods, explicit formulas for the gradients used, algorithm parameters, details on the generation of the dataset, and implementations details. The code for the experiments is available at this link.

ACKNOWLEDGMENTS

This research was partially supported by the Israel Science Foundation (ISF), grant no. 1782/22.

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

The outline of the supplementary material is as follows. In Appendix A, we review additional related work. In Appendix B, we discuss in detail the origin of prior knowledge of classes of prediction problems. In Appendix C, we set our notation conventions. In Appendix D, we summarize a few mathematical results that are used in later proofs. In Appendix E, we show that PCA can be cast as a degenerate setting of our formulation, and provide the proofs of the main theorems in the paper (the linear MSE setting). In Appendix F, we generalize these results to an infinite dimensional Hilbert space. In Appendix G, we provide two algorithms for solving the Phase 1 and Phase 2 problems in Algorithm 1. In Appendix H, we provide details on the examples for the experiments with the iterative algorithm, and additional experiments.

## A    ADDITIONAL RELATED WORK

**The information bottleneck principle**    The IB principle is a prominent approach to feature relevance in the design of representations (Tishby et al., 2000; Chechik et al., 2003; Slonim et al., 2006; Harremoës and Tishby, 2007), and proposes to optimize the representation in order to maximize its relevance to the response $y$. Letting $I(z; y)$ and $I(x; z)$ denote the corresponding mutual information terms (Cover and Thomas, 2006), the IB principle aims to maximize the former while constraining the latter from above, and this is typically achieved via a Lagrangian formulation (Boyd et al., 2004). The resulting representation, however, is tailored to the joint distribution of $(x, y)$, i.e., to a specific prediction task. In practice, this is achieved using a labeled dataset (generalization bounds were derived by Shamir et al. (2010)). As in our mixed representation approach, the use of randomized representation dictated by a probability kernel $z \sim f(\cdot \mid x = x)$ is inherent to the IB principle.

A general observation made from exploring deep neural networks (DNNs) (Goodfellow et al., 2016) used for classification, is that practically good predictors in fact first learn an efficient representation $z$ of the features, and then just train a simple predictor from $z$ to the response $y = f(x)$. This is claimed to be quantified by the IB where low mutual information $I(x; z)$ indicates an efficient representation, and high mutual information $I(z; y)$ indicates that the representation allows for efficient prediction. This idea has been further developed using the IB principle, by hypothesizing that modern prediction algorithms must intrinsically include learning an efficient representation (Tishby and Zaslavsky, 2015; Shwartz-Ziv and Tishby, 2017; Shwartz-Ziv, 2022; Achille and Soatto, 2018a;b) (this spurred a debate, see, e.g., (Saxe et al., 2019; Geiger, 2021)).

However, this approach is inadequate in our setting, since the prediction task is not completely specified to the learner, and in the IB formulation the optimal representation depends on the response variable (so that labeled data should be provided to the learner). In addition, as explained by Dubois et al. (2020), while the resulting IB solution provides a fundamental limit for the problem, it also suffers from multiple theoretical and practical issues. The first main issue is that the mutual information terms are inherently difficult to estimate from finite samples (Shamir et al., 2010; Nguyen et al., 2010; Poole et al., 2018; Wu et al., 2020; McAllester and Stratos, 2020), especially at high dimensions, and thus require resorting to approximations, e.g., variational bounds (Chalk et al., 2016; Alemi et al., 2016; Belghazi et al., 2018; Razeghi et al., 2022). The resulting generalization bounds (Shamir et al., 2010; Vera et al., 2018) are still vacuous for modern settings (Rodriguez Galvez, 2019). The second main issue is that the IB formulation does not constrain the complexity of the representation and the prediction rule, which can be arbitrarily complex. These issues were addressed by Dubois et al. (2020) using the notion of *usable information*, previously introduced by Xu et al. (2020): The standard mutual information $I(z; y)$ can be described as the log-loss difference between a predictor for $z$ which does not use or does use $y$ (or vice-versa, since mutual information is symmetric). Usable information, or $\mathcal{F}$-information $I_{\mathcal{F}}(z \rightarrow y)$, restricts the predictor to a class $\mathcal{F}$, which is computationally constrained. Several desirable properties were established in Xu et al. (2020) for the $\mathcal{F}$-information, e.g., probably approximate correct (PAC) bounds via Rademacher-complexity based bounds (Bartlett and Mendelson, 2002) (Wainwright, 2019, Chapter 5)(Shalev-Shwartz and Ben-David, 2014, Chapters 26-28).

Dubois et al. (2020) used the notion of $\mathcal{F}$-information to define the *decodable IB* problem, with the goal of assessing the generalization capabilities of this IB problem, and shedding light on the necessity of efficient representation for generalization. To this end, a game was proposed, in which the data available to the learner are feature-response pairs $(x, y)$ (in our notation $y = f(x)$). In

the game proposed by Dubois et al. (2020), Alice chooses a prediction problem $f(\cdot)$, i.e., a feature-response pair $(\boldsymbol{x}; \boldsymbol{y})$, and Bob chooses a representation $\boldsymbol{z} = R(\boldsymbol{x})$. For comparison, in this paper, we assume that the learner is given features $\boldsymbol{x}$ and a *class* $\mathcal{F}$ of prediction problems. We ask how to choose the representation $R$ if it is only known that the response function $Y = f(X)$ can be chosen adversarially from $\mathcal{F}$. In our formulated game, the order of plays is thus different. First, the representation player chooses $R$, and then the adversarial function player chooses $f \in \mathcal{F}$. Beyond those works, the IB framework has drawn a significant recent attention, and a remarkable number of extensions and ramifications have been proposed (Sridharan and Kakade, 2008; Amjad and Geiger, 2019; Kolchinsky et al., 2019; Strouse and Schwab, 2019; Pensia et al., 2020; Asoodeh and Calmon, 2020; Ngampruetikorn and Schwab, 2021; Yu et al., 2021; Ngampruetikorn and Schwab, 2022; Gündüz et al., 2022; Razeghi et al., 2022; Ngampruetikorn and Schwab, 2023). An IB framework for self-supervised learning was recently discussed in (Ngampruetikorn et al., 2020).

**Multitask learning and learning-to-learn** In the problem of multitask learning and learning-to-learn (Baxter, 2000; Argyriou et al., 2006; Maurer et al., 2016; Du et al., 2020; Tripuraneni et al., 2020; 2021), the goal of the learner is to jointly learn multiple downstream prediction tasks. The underlying implicit assumption is that the tasks are similar in some way, specifically, that they share a common low dimensional representation. In the notation of this paper, a class $\mathcal{F} = \{f_1, f_2, \cdots, f_t\}$ of $t$ prediction task is given, and a predictor from $R \colon \mathbb{R}^d \to \mathcal{Y}$ is decomposed as $Q_i(R(\boldsymbol{x}))$, where the representation $\boldsymbol{z} = R(\boldsymbol{x}) \in \mathbb{R}^r$ is common to all tasks. The learner is given a dataset for each of the tasks, and its goal is to learn the common representation, as well as the $t$ individual predictors in the multitask setting. In the learning-to-learn setting, the learner will be presented with a new prediction tasks, and so only the representation is retained.

Maurer et al. (2016) assumed that the representation is chosen from a class $\mathcal{R}$ and the predictors from a class $\mathcal{Q}$ (in our notation), and generalization bounds on the average excess risk for an empirical risk minimization (ERM) learning algorithm were derived, highlighting the different complexity measures associated with the problem. In the context of learning-to-learn, the bound in (Maurer et al., 2016, Theorem 2) scales as $O(1/\sqrt{t}) + O(1/\sqrt{m})$, where $m$ is the number of samples provided to the learner from the new task. In practice, it is often the case that learning-to-learn is efficient even for small $t$, which implies that this bound is loose. It was then improved by Tripuraneni et al. (2020), whose main statement is that a proper notion of task diversity is crucial for generalization. They then proposed an explicit notion of task diversity, controlled by a term $\nu$ and obtained a bound of the form $\tilde{O}(\frac{1}{\nu} \left( \sqrt{\frac{C(\mathcal{R}) + tC(\mathcal{Q})}{nt}} + \sqrt{\frac{C(\mathcal{Q})}{m}} \right))$ (in our notation), where $C(\mathcal{R})$ and $C(\mathcal{Q})$ are complexity measures for the representation class and the prediction class. For comparison, in this paper we focus on finding the optimal representation, either theoretically (in the fundamental linear MSE setting) or algorithmically, rather than relying on a generic ERM. To this end we side-step the generalization error, and so the regret we define can be thought of as an *approximation error*. One direct consequence of this difference is that in our case task diversity (rich $\mathcal{F}$) leads to a large regret, whereas in (Tripuraneni et al., 2020) task diversity leads to low generalization bound. From this aspect, our results complement those of Tripuraneni et al. (2020) to the non-realizable case (that is, when the prediction tasks cannot be decomposed as a composition $f(\boldsymbol{x}) = Q(R(\boldsymbol{x}))$.

**Randomization in representation learning** Randomization is classically used in data representation, most notably, utilizing the seminal Johnson-Lindenstrauss Lemma Johnson (1984) or more generally, *sketching* algorithms (e.g., (Vempala, 2005; Mahoney et al., 2011; Woodruff et al., 2014; Yang et al., 2021)). Our use of randomization is different and is inspired by the classical Nash equilibrium (Nash Jr, 1950). Rather than using a single deterministic representation that was randomly chosen, we consider randomizing multiple representation rules. Such randomization is commonly used in the face of future uncertainty, which in our setting is the downstream prediction task.

**Game-theoretic formulations in statistics and machine-learning** The use of game-theoretic formulations in statistics, between a player choosing a prediction algorithm and an adversary choosing a prediction problem (typically Nature), was established by Wald (1939) in his classical statistical decision theory (see, e.g., (Wasserman, 2004, Chapter 12)). It is a common approach both in classic statistics and learning theory (Yang and Barron, 1999; Grünwald and Dawid, 2004; Haussler and Opper, 1997; Farnia and Tse, 2016), as well as in modern high-dimensional statistics (Wainwright, 2019). The effect of the representation (quantizer) on the consistency of learning algorithms when

a surrogate convex loss function replaces the loss function of interest was studied in (Nguyen et al., 2009; Duchi et al., 2018; Grünwald and Dawid, 2004) (for binary and multiclass classification, respectively). A relation between information loss and minimal error probability was recently derived by Silva and Tobar (2022).

Iterative algorithms for the solution of minimax games have drawn much attention in the last few years due to their importance in optimizing generative adversarial networks (GANs) (Goodfellow et al., 2020; Creswell et al., 2018), adversarial training (Madry et al., 2017), and robust optimization (Ben-Tal et al., 2009). The notion of convergence is rather delicate, even for the basic convex-concave two-player setting (Salimans et al., 2016). While the value output by the MWU algorithm (Freund and Schapire, 1999), or improved versions (Daskalakis et al., 2011; Rakhlin and Sridharan, 2013) converges to a no-regret solution, the actual strategies used by the players are, in fact, repelled away from the equilibrium point to the boundary of the probability simplex (Bailey and Piliouras, 2018). For general games, the gradient descent ascent (GDA) is a natural and practical choice, yet despite recent advances, its theory is still partial (Zhang et al., 2022). Various other algorithms have been proposed, e.g., (Schäfer and Anandkumar, 2019; Mescheder et al., 2017; Letcher et al., 2019; Gidel et al., 2019; Zhang and Wang, 2021).

**Incremental learning of mixture models** Our proposed Algorithm 1 operates iteratively, and each main iteration adds a representation rule as a new component to the existing mixture of representation rules. More broadly, the efficiency of algorithms that follow this idea is based on two principles: (i) A powerful model can be obtained from a mixture of a few weak models; (ii) Mixture models can be efficiently learned by a gradual addition of components to the mixture, if the new component aims to address the most challenging problem instance. As a classic example, this idea is instantiated by the boosting method (Schapire and Freund, 2012) for classification, and was adapted for generative models by Tolstikhin et al. (2017) for GANs. In boosting, the final classifier is a mixture of simpler classifiers. Large weights are put on data points which are wrongly classified with the current mixture of classifiers, and the new component (classifier) is trained to cope with these samples. In GANs, the generated distribution is a mixture is of generative models. Large weights are put on examples which are easily discerned by the discriminator of the true and generated distributions, and the new component (generative distribution) optimizes the GAN objective on this weighted data. In our setting, the final representation is a mixture of representation rules. Weights are put on adversarial functions that cannot be accurately predicted with the current representation matrices. The new representation component aims to allow for accurate prediction of these functions. Overall, the common intuitive idea is very natural: The learning algorithm sees what is most lacking in the current mixture, and adds a new component that directly aims to minimize this shortage. We refer the reader to (Tolstikhin et al., 2017) for a more in-depth exposition of this idea. As mentioned by Tolstikhin et al. (2017), this idea dates back to the use of boosting for density estimation (Welling et al., 2002).

**Unsupervised pretraining** From a broader perspective, our method is essentially an *unsupervised pretraining* method, similar to the methods which currently enable the recent success in natural language processing. Our model is much simplified compared to transformer architecture (Vaswani et al., 2017), but the unsupervised training aspect used for prediction tasks (Devlin et al., 2018) is common, and our results may shed light on these methods. For example, putting more weight on some words compared to others during training phase that uses the masked-token prediction objective.

## B  CLASSES OF RESPONSE FUNCTIONS

Our approach to optimal representation is based on the assumption that a class $\mathcal{F}$ of downstream prediction tasks is known. This assumption may represent prior knowledge or constraints on the response function, and can stem from various considerations. To begin, it might be hypothesized that some features are less relevant than others. As a simple intuitive example, the outer pixels in images are typically less relevant to the classification of photographed objects, regardless of their variability (which may stem from other effects, such as lighting conditions). Similarly, non-coding regions of the genotype are irrelevant for predicting phenotype. The prior knowledge may encode softer variations in relevance. Moreover, such prior assumption may be imposed on the learned

function, e.g., it may be assumed that the response function respects the privacy of some features, or only weakly depends on features which provide an unfair advantage. In domain adaptation (Mansour et al., 2009), one may solve the prediction problem for feature distribution $P_{\boldsymbol{x}}$ obtaining a optimal response function $f_1$. Then, after a change of input distribution to $Q_{\boldsymbol{x}}$, the response function learned for this feature distribution $f_2$ may be assumed to belong to functions which are "compatible" with $f_1$. For example, if $P_{\boldsymbol{x}}$ and $Q_{\boldsymbol{x}}$ are supported on different subsets of $\mathbb{R}^d$, the learned response function $f_1(x)$ and $f_2(x)$ may be assumed to satisfy some type of continuity assumptions. Similar assumptions may hold for the more general setting of transfer learning (Ben-David et al., 2010). Furthermore, such assumptions may hold in a *continual learning* setting (Zenke et al., 2017; Nguyen et al., 2017; Van de Ven and Tolias, 2019; Aljundi et al., 2019), in which a sequence of response functions is learned one task at a time. Assuming that *catastrophic forgetting* is aimed to be avoided, then starting from the second task, the choice of representation may assume that the learned response function is accurate for all previously learned tasks.

## C    NOTATION CONVENTIONS

For an integer $d$, $[d] := \{1, 2, \ldots, d\}$. For $p \geq 1$, $\|x\|_p := (\sum_{i=1}^{d} |x_i|^p)^{1/p}$ is the $\ell_p$ norm of $x \in \mathbb{R}^d$. The Frobenius norm of the matrix $A$ is denoted by $\|A\|_F = \sqrt{\mathrm{Tr}[A^T A]}$ . The non-negative (resp. positive) definite cone of symmetric matrices is given by $\mathbb{S}_+^d$ (resp. $\mathbb{S}_{++}^d$). For a given positive-definite matrix $S \in \mathbb{S}_{++}^d$, the Mahalanobis norm of $x \in \mathbb{R}^d$ is given by $\|x\|_S := \|S^{-1/2} x\|_2 = (x^\top S^{-1} x)^{1/2}$, where $S^{1/2}$ is the symmetric square root of $S$. The matrix $W := [w_1, \ldots, w_r] \in \mathbb{R}^{d \times r}$ is comprised from the column vectors $\{w_i\}_{i \in [r]} \subset \mathbb{R}^d$. For a real symmetric matrix $S \in \mathbb{S}^d$, $\lambda_i(S)$ is the $i$th largest eigenvalue, so that $\lambda_{\max}(S) \equiv \lambda_1(S) \geq \lambda_2(S) \geq \cdots \geq \lambda_d(S) = \lambda_{\min}(S)$, and in accordance, $v_i(S)$ denote an eigenvector corresponding to $\lambda_i(S)$ (these are unique if there are no two equal eigenvalues, and otherwise arbitrarily chosen, while satisfying orthogonality $v_i^\top v_j = \langle v_i, v_j \rangle = \delta_{ij}$). Similarly, $\Lambda(S) := \mathrm{diag}(\lambda_1(S), \lambda_2(S), \cdots, \lambda_d(S))$ and $V(S) := [v_1(S), v_2(S), \cdots, v_d(S)]$, so that $S = V(S)\Lambda(S)V^\top(S)$ is an eigenvalue decomposition. For $j \geq i$, $V_{i:j} := [v_i, \ldots, v_j] \in \mathbb{R}^{(j-i+1) \times d}$ is the matrix comprised of the columns indexed by $\{i, \ldots, j\}$. The vector $e_i \in \mathbb{R}^d$ is the $i$th standard basis vector, that is, $e_i := [\underbrace{0, \ldots 0}_{i-1 \text{ terms}}, 1, \underbrace{0, \ldots 0}_{d-i \text{ terms}}]^\top$.

Random quantities (scalars, vectors, matrices, etc.) are denoted by boldface letters. For example, $\boldsymbol{x} \in \mathbb{R}^d$ is a random vector that takes values $x \in \mathbb{R}^d$ and $\boldsymbol{R} \in \mathbb{R}^{d \times r}$ is a random matrix. The probability law of a random element $\boldsymbol{x}$ is denoted by $\mathsf{L}(\boldsymbol{x})$. The probability of the event $\mathcal{E}$ in some given probability space is denoted by $\mathbb{P}[\mathcal{E}]$ (typically understood from context). The expectation operator is denoted by $\mathbb{E}[\cdot]$. The indicator function is denoted by $\mathbb{1}\{\cdot\}$, and the Kronecker delta is denoted by $\delta_{ij} := \mathbb{1}\{i = j\}$. We do not make a distinction between minimum and infimum (or maximum and supremum) as arbitrarily accurate approximation is sufficient for the description of the results in this paper. The binary KL divergence between $p_1, p_2 \in (0, 1)$ is denoted as

$$D_{\mathrm{KL}}(p_1 \parallel p_2) := p_1 \log \frac{p_1}{p_2} + (1 - p_1) \log \frac{1 - p_1}{1 - p_2}. \tag{27}$$

## D    USEFUL MATHEMATICAL RESULTS

In this section we provide several simplified versions of mathematical results that are used in the proofs. The following well-known result is about the optimal low-rank approximation of a given matrix:

**Theorem 10** (*Eckart-Young-Mirsky* (Wainwright, 2019, Example 8.1) (Vershynin, 2018, Section 4.1.4)). *For a symmetric matrix $S \in \mathbb{S}^d$*

$$\|S_k - S\|_F \leq \min_{S' \in \mathbb{S}^d : \mathrm{rank}(S') \leq k} \|S - S'\|_F \tag{28}$$

*where*

$$S_k = \sum_{i \in [k]} \lambda_i(S) \cdot v_i(S) v_i^\top(S) \tag{29}$$

*(more generally, this is true for any unitarily invariant norm).*

We next review a simplified version of variational characterizations of eigenvalues of symmetric matrices:

**Theorem 11** (*Rayleigh quotient* (Horn and Johnson, 2012, Theorem 4.2.2))**.** *For a symmetric matrix* $S \in \mathbb{S}^d$

$$\lambda_1(S) = \max_{x \neq 0} \frac{x^\top S x}{\|x\|_2^2}. \tag{30}$$

**Theorem 12** (*Courant–Fisher variational characterization* (Horn and Johnson, 2012, Theorem 4.2.6))**.** *For a symmetric matrix* $S \in \mathbb{S}^d$, $k \in [d]$, *and a subspace* $T$ *of* $\mathbb{R}^d$

$$\lambda_k(S) = \min_{T:\dim(T)=k} \max_{x \in T \setminus \{0\}} \frac{x^\top S x}{\|x\|_2^2} = \max_{T:\dim(T)=d-k+1} \min_{x \in T \setminus \{0\}} \frac{x^\top S x}{\|x\|_2^2}. \tag{31}$$

**Theorem 13** (*Fan's variational characterization (Horn and Johnson, 2012, Corollary 4.3.39.)*)**.** *For a symmetric matrix* $S \in \mathbb{S}^d$ *and* $k \in [d]$

$$\lambda_1(S) + \cdots + \lambda_k(S) = \min_{U \in \mathbb{R}^{d \times k}: U^\top U = I_k} \mathrm{Tr}[U^\top S U] \tag{32}$$

*and*

$$\lambda_{d-k+1}(S) + \cdots + \lambda_d(S) = \max_{U \in \mathbb{R}^{d \times k}: U^\top U = I_k} \mathrm{Tr}[U^\top S U]. \tag{33}$$

We will use the following celebrated result from convex analysis.

**Theorem 14** (*Carathéodory's theorem (Bertsekas et al., 2003, Prop. 1.3.1)*)**.** *Let* $\mathcal{A} \subset \mathbb{R}^d$ *be non-empty. Then, any point* $a$ *in the convex hull of* $\mathcal{A}$ *can be written as a convex combination of at most* $d + 1$ *points from* $\mathcal{A}$.

## E  THE LINEAR MSE SETTING: ADDITIONS AND PROOFS

### E.1  THE STANDARD PRINCIPAL COMPONENT SETTING

In order to highlight the formulation proposed in this paper, we show, as a starting point, that the well known PCA solution of representing $\boldsymbol{x} \in \mathbb{R}^d$ with the top $r$ eigenvectors of the covariance matrix of $\boldsymbol{x}$ can be obtained as a specific case of the regret formulation. In this setting, we take $\mathcal{F} = \{I_d\}$, and so $\boldsymbol{y} = \boldsymbol{x}$ with probability 1. In addition, the predictor class $\mathcal{Q}$ is a linear function from the representation dimension $r$ back to the features dimension $d$.

**Proposition 15.** *Consider the linear MSE setting, with the difference that the response is* $\boldsymbol{y} \in \mathbb{R}^d$, *the loss function is the squared Euclidean norm* $\mathsf{loss}(y_1, y_2) = \|y_1 - y_2\|^2$, *and the predictor is* $Q(z) = Q^\top z \in \mathbb{R}^d$ *for* $Q \in \mathbb{R}^{r \times d}$. *Assume* $\mathcal{F} = \{I_d\}$ *so that* $\boldsymbol{y} = \boldsymbol{x}$ *with probability* 1. *Then,*

$$\mathsf{regret}_{\mathsf{pure}}(\mathcal{F} \mid \Sigma_{\boldsymbol{x}}) = \mathsf{regret}_{\mathsf{mix}}(\mathcal{F} \mid \Sigma_{\boldsymbol{x}}) = \sum_{i=r+1}^{d} \lambda_i(\Sigma_{\boldsymbol{x}}), \tag{34}$$

*and an optimal representation is* $R = V_{1:r}(\Sigma_{\boldsymbol{x}})$.

The result of Proposition 15 verifies that the minimax and maximin formulations indeed generalize the standard PCA formulation. The proof is standard and follows from the *Eckart-Young-Mirsky theorem*, which determines the best rank $r$ approximation in the Frobenius norm.

*Proof of Proposition 15.* Since $\mathcal{F} = \{I_d\}$ is a singleton, there is no distinction between pure and mixed minimax regret. It holds that

$$\mathsf{regret}(R, f) = \mathbb{E}\left[\|\boldsymbol{x} - Q^\top R^\top \boldsymbol{x}\|^2\right] \tag{35}$$

where $A = Q^\top R^\top \in \mathbb{R}^{d \times d}$ is a rank $r$ matrix. For any $A \in \mathbb{R}^{d \times d}$

$$\mathbb{E}\left[\|\boldsymbol{x} - A\boldsymbol{x}\|^2\right] = \mathbb{E}\left[\boldsymbol{x}^\top \boldsymbol{x} - \boldsymbol{x}^\top A\boldsymbol{x} - \boldsymbol{x}^\top A^\top \boldsymbol{x} + \boldsymbol{x}^\top A^\top A\boldsymbol{x}\right] \tag{36}$$

$$= \mathrm{Tr}\left[\Sigma_{\boldsymbol{x}} - A\Sigma_{\boldsymbol{x}} - A^\top \Sigma_{\boldsymbol{x}} + A^\top A\Sigma_{\boldsymbol{x}}\right] \tag{37}$$

$$= \left\|\Sigma_{\boldsymbol{x}}^{1/2} - \Sigma_{\boldsymbol{x}}^{1/2} A\right\|_F^2 \tag{38}$$

$$= \left\| \Sigma_{\boldsymbol{x}}^{1/2} - B \right\|_F^2, \tag{39}$$

where $B := \Sigma_{\boldsymbol{x}}^{1/2} A$. By the classic *Eckart-Young-Mirsky theorem* (Wainwright, 2019, Example 8.1) (Vershynin, 2018, Section 4.1.4) (see Appendix D), the best rank $r$ approximation in the Frobenius norm is obtained by setting

$$B^* = \sum_{i=1}^{r} \lambda_i(\Sigma_{\boldsymbol{x}}^{1/2}) \cdot v_i v_i^\top = \sum_{i=1}^{r} \sqrt{\lambda_i(\Sigma_{\boldsymbol{x}})} \cdot v_i v_i^\top \tag{40}$$

where $v_i \equiv v_i(\Sigma_{\boldsymbol{x}}^{1/2}) = v_i(\Sigma_{\boldsymbol{x}})$ is the $i$th eigenvector of $\Sigma_{\boldsymbol{x}}^{1/2}$ (or $\Sigma_{\boldsymbol{x}}$). Then, the optimal $A$ is

$$A^* = \sum_{i=1}^{r} \sqrt{\lambda_i(\Sigma_{\boldsymbol{x}})} \cdot \Sigma_{\boldsymbol{x}}^{-1/2} v_i v_i^\top = \sum_{i=1}^{r} \sqrt{\lambda_i(\Sigma_{\boldsymbol{x}})} \cdot \Sigma_{\boldsymbol{x}}^{-1/2} v_i v_i^\top = \sum_{i=1}^{r} v_i v_i^\top, \tag{41}$$

since $v_i$ is also an eigenvector of $\Sigma_{\boldsymbol{x}}^{-1/2}$. Letting $R = U(R)\Sigma(R)V^\top(R)$ and $Q = U(Q)\Sigma(Q)V^\top(Q)$ be the singular value decomposition of $R$ and $Q$, respectively, it holds that

$$Q^\top R^\top = V(Q)\Sigma^\top(Q)V(Q)V(R)\Sigma^\top(R)U^\top(R). \tag{42}$$

Setting $V(Q) = V(R) = I_r$, and $\Sigma^\top(Q) = \Sigma(R) \in \mathbb{R}^{d \times r}$ to have $r$ ones on the diagonal (and all other entries are zero), as well as $U(Q) = U(R)$ to be an orthogonal matrix whose first $r$ columns are $\{v_i\}_{i \in [r]}$ results that $Q^\top R^\top = A^*$, as required. $\qquad\square$

### E.2 PROOFS OF PURE AND MIXED MINIMAX REPRESENTATIONS

Before the proof of Theorem 2, we state a simple and useful lemma, which provides the pointwise value of the regret and the optimal linear predictor for a given representation and response.

**Lemma 16.** *Consider the representation $\boldsymbol{z} = R^\top \boldsymbol{x} \in \mathbb{R}^r$. It then holds that*

$$\mathsf{regret}(R, f \mid P_{\boldsymbol{x}}) = \min_{q \in \mathbb{R}^r} \mathbb{E}\left[ \left( f^\top \boldsymbol{x} + \boldsymbol{n} - q^\top \boldsymbol{z} \right)^2 \right] \tag{43}$$

$$= \mathbb{E}\left[ \mathbb{E}[\boldsymbol{n}^2 \mid \boldsymbol{x}] \right] + f^\top \left( \Sigma_{\boldsymbol{x}} - \Sigma_{\boldsymbol{x}} R (R^\top \Sigma_{\boldsymbol{x}} R)^{-1} R^\top \Sigma_{\boldsymbol{x}} \right) f. \tag{44}$$

*Proof.* The orthogonality principle states that

$$\mathbb{E}\left[ \left( f^\top \boldsymbol{x} + \boldsymbol{n} - q^\top \boldsymbol{z} \right) \cdot \boldsymbol{z}^\top \right] = 0 \tag{45}$$

must hold for the optimal linear estimator. Using $\boldsymbol{z} = R^\top \boldsymbol{x}$ and taking expectations leads to the ordinary least-squares (OLS) solution

$$q_{\mathsf{LS}} = (R^\top \Sigma_{\boldsymbol{x}} R)^{-1} R^T \Sigma_{\boldsymbol{x}} f, \tag{46}$$

assuming that $R^\top \Sigma_{\boldsymbol{x}} R$ is invertible (which we indeed assume as if this is not the case, the representation can be reduced to a dimension lower than $r$ in a lossless manner). The resulting regret of $R$ is thus given by

$$\mathsf{regret}(R, f \mid P_{\boldsymbol{x}}) = \mathbb{E}\left[ \left( f^\top \boldsymbol{x} + \boldsymbol{n} - q_{\mathsf{LS}}^\top \boldsymbol{z} \right)^2 \right] \tag{47}$$

$$\overset{(a)}{=} \mathbb{E}\left[ \left( f^\top \boldsymbol{x} + \boldsymbol{n} \right)^\top \left( f^\top \boldsymbol{x} + \boldsymbol{n} - q_{\mathsf{LS}}^\top \boldsymbol{z} \right) \right] \tag{48}$$

$$= \mathbb{E}\left[ \left( f^\top \boldsymbol{x} + \boldsymbol{n} \right)^2 - \left( f^\top \boldsymbol{x} + \boldsymbol{n} \right)^\top q_{\mathsf{LS}}^\top \boldsymbol{z} \right] \tag{49}$$

$$\overset{(b)}{=} \mathbb{E}\left[ \mathbb{E}[\boldsymbol{n}^2 \mid \boldsymbol{x}] \right] + f^\top \Sigma_{\boldsymbol{x}} f - \mathbb{E}\left[ \boldsymbol{x}^\top f q_{\mathsf{LS}}^\top R^\top \boldsymbol{x} \right] \tag{50}$$

$$= \mathbb{E}\left[ \mathbb{E}[\boldsymbol{n}^2 \mid \boldsymbol{x}] \right] + f^\top \Sigma_{\boldsymbol{x}} f - \mathrm{Tr}\left[ f q_{\mathsf{LS}}^\top R^\top \Sigma_{\boldsymbol{x}} \right] \tag{51}$$

$$= \mathbb{E}\left[ \mathbb{E}[\boldsymbol{n}^2 \mid \boldsymbol{x}] \right] + f^\top \Sigma_{\boldsymbol{x}} f - q_{\mathsf{LS}}^\top R^\top \Sigma_{\boldsymbol{x}} f \tag{52}$$

$$\overset{(c)}{=} \mathbb{E}\left[ \mathbb{E}[\boldsymbol{n}^2 \mid \boldsymbol{x}] \right] + f^\top \left( \Sigma_{\boldsymbol{x}} - \Sigma_{\boldsymbol{x}} R (R^\top \Sigma_{\boldsymbol{x}} R)^{-1} R^\top \Sigma_{\boldsymbol{x}} \right) f, \tag{53}$$

where $(a)$ follows from the orthogonality principle in (45), $(b)$ follows from the tower property of conditional expectation and since $\mathbb{E}[\boldsymbol{x}\boldsymbol{n}] = \mathbb{E}[\boldsymbol{x} \cdot \mathbb{E}[\boldsymbol{n} \mid \boldsymbol{x}]] = 0$, and $(c)$ follows by substituting $q_{\mathsf{LS}}$ from (46). $\qquad\square$

We may now prove Theorem 2.

*Proof of Theorem 2.* For any given $f$, the optimal predictor based on $x \in \mathbb{R}^d$ achieves average loss of

$$\text{regret}(R = I_d, f \mid P_{\boldsymbol{x}}) = \min_{q \in \mathbb{R}^d} \mathbb{E}\left[\left(f^\top \boldsymbol{x} + \boldsymbol{n} - q^\top \boldsymbol{x}\right)^2\right] = \mathbb{E}\left[\mathbb{E}[\boldsymbol{n}^2 \mid \boldsymbol{x}]\right] \quad (54)$$

(obtained by setting $R = I_d$ in Lemma 16 so that $\boldsymbol{z} = \boldsymbol{x}$). Hence, the resulting regret of $R$ over an adversarial choice of $f \in \mathcal{F}_S$ is

$$\max_{f \in \mathcal{F}_S} \text{regret}(R, f) = \max_{f \in \mathcal{F}} \mathbb{E}\left[\left|f^\top \boldsymbol{x} + \boldsymbol{n} - q_{\text{LS}}^\top \boldsymbol{z}\right|^2\right] - \mathbb{E}\left[\mathbb{E}[\boldsymbol{n}^2 \mid \boldsymbol{x}]\right]$$

$$\stackrel{(a)}{=} \max_{f \in \mathcal{F}_S} f^\top \left(\Sigma_{\boldsymbol{x}} - \Sigma_{\boldsymbol{x}} R(R^\top \Sigma_{\boldsymbol{x}} R)^{-1} R^\top \Sigma_{\boldsymbol{x}}\right) f \quad (55)$$

$$\stackrel{(b)}{=} \max_{\tilde{f} : \|\tilde{f}\|_2^2 \le 1} \tilde{f}^\top \left(S^{1/2} \Sigma_{\boldsymbol{x}} S^{1/2} - S^{1/2} \Sigma_{\boldsymbol{x}} R(R^\top \Sigma_{\boldsymbol{x}} R)^{-1} R^\top \Sigma_{\boldsymbol{x}} S^{1/2}\right) \tilde{f} \quad (56)$$

$$\stackrel{(c)}{=} \lambda_1 \left(S^{1/2} \Sigma_{\boldsymbol{x}} S^{1/2} - S^{1/2} \Sigma_{\boldsymbol{x}} R(R^\top \Sigma_{\boldsymbol{x}} R)^{-1} R^\top \Sigma_{\boldsymbol{x}} S^{1/2}\right) \quad (57)$$

$$= \lambda_1 \left[S^{1/2} \Sigma_{\boldsymbol{x}}^{1/2} \left(I_d - \Sigma_{\boldsymbol{x}}^{1/2} R(R^\top \Sigma_{\boldsymbol{x}} R)^{-1} R^\top \Sigma_{\boldsymbol{x}}^{1/2}\right) \Sigma_{\boldsymbol{x}}^{1/2} S^{1/2}\right] \quad (58)$$

$$\stackrel{(d)}{=} \lambda_1 \left[S^{1/2} \Sigma_{\boldsymbol{x}}^{1/2} \left(I_d - \tilde{R}(\tilde{R}^\top \tilde{R})^{-1} \tilde{R}^\top\right) \Sigma_{\boldsymbol{x}}^{1/2} S^{1/2}\right] \quad (59)$$

$$\stackrel{(e)}{=} \lambda_1 \left[\left(I_d - \tilde{R}(\tilde{R}^\top \tilde{R})^{-1} \tilde{R}^\top\right) \Sigma_{\boldsymbol{x}}^{1/2} S \Sigma_{\boldsymbol{x}}^{1/2} \left(I_d - \tilde{R}(\tilde{R}^\top \tilde{R})^{-1} \tilde{R}^\top\right)\right], \quad (60)$$

where $(a)$ follows from Lemma 16, $(b)$ follows by letting $\tilde{f} := S^{-1/2} f$ and recalling that any $f \in \mathcal{F}$ must satisfy $\|f\|_S^2 \le 1$, $(c)$ follows from the *Rayleigh quotient theorem* (Horn and Johnson, 2012, Theorem 4.2.2) (see Appendix D), $(d)$ follows by letting $\tilde{R} := \Sigma_{\boldsymbol{x}}^{1/2} R$, and $(e)$ follows since $I_d - \tilde{R}(\tilde{R}^\top \tilde{R})^{-1} \tilde{R}^\top$ is an orthogonal projection (idempotent and symmetric matrix) of rank $d - r$.

Now, to find the minimizer of $\max_{f \in \mathcal{F}_S} \text{regret}(R, f)$ over $R$, we note that

$$\lambda_1 \left[\left(I_d - \tilde{R}(\tilde{R}^\top \tilde{R})^{-1} \tilde{R}^\top\right) \Sigma_{\boldsymbol{x}}^{1/2} S \Sigma_{\boldsymbol{x}}^{1/2} \left(I_d - \tilde{R}(\tilde{R}^\top \tilde{R})^{-1} \tilde{R}^\top\right)\right]$$

$$\stackrel{(a)}{=} \max_{u : \|u\|_2 = 1} u^\top \left(I_d - \tilde{R}(\tilde{R}^\top \tilde{R})^{-1} \tilde{R}^\top\right) \Sigma_{\boldsymbol{x}}^{1/2} S \Sigma_{\boldsymbol{x}}^{1/2} \left(I_d - \tilde{R}(\tilde{R}^\top \tilde{R})^{-1} \tilde{R}^\top\right) u \quad (61)$$

$$\stackrel{(b)}{=} \max_{u : \|u\|_2 = 1, \; \tilde{R}^\top u = 0} u^\top \Sigma_{\boldsymbol{x}}^{1/2} S \Sigma_{\boldsymbol{x}}^{1/2} u \quad (62)$$

$$\stackrel{(c)}{\ge} \min_{\mathcal{S} : \dim(\mathcal{S}) = d - r} \max_{u : \|u\|_2 = 1, \; u \in \mathcal{S}} u^\top \Sigma_{\boldsymbol{x}}^{1/2} S \Sigma_{\boldsymbol{x}}^{1/2} u \quad (63)$$

$$\stackrel{(d)}{=} \lambda_{r+1} \left(\Sigma_{\boldsymbol{x}}^{1/2} S \Sigma_{\boldsymbol{x}}^{1/2}\right), \quad (64)$$

where $(a)$ follows again from the *Rayleigh quotient theorem* (Horn and Johnson, 2012, Theorem 4.2.2), $(b)$ follows since $I_d - \tilde{R}(\tilde{R}^\top \tilde{R})^{-1} \tilde{R}^\top$ is an orthogonal projection matrix, and so we may write $u = u_\perp + u_\parallel$ so that $\|u_\perp\|^2 + \|u_\parallel\|^2 = 1$ and $\tilde{R}^\top u_\perp = 0$; Hence replacing $u$ with $u_\perp$ only increases the value of the maximum, $(c)$ follows by setting $\mathcal{S}$ to be a $d - r$ dimensional subspace of $\mathbb{R}^d$, and $(d)$ follows by the *Courant–Fischer variational characterization* (Horn and Johnson, 2012, Theorem 4.2.6) (see Appendix D). The lower bound in $(c)$ can be achieved by setting the $r$ columns of $\tilde{R} \in \mathbb{R}^{d \times r}$ to be the top eigenvectors $\{v_i(\Sigma_{\boldsymbol{x}}^{1/2} S \Sigma_{\boldsymbol{x}}^{1/2})\}_{i \in [r]}$. This leads to the minimax representation $\tilde{R}^*$. From (60), the worst case $\tilde{f}$ is the top eigenvector of

$$\left(I_d - \tilde{R}^*((\tilde{R})^{*\top} \tilde{R}^*)^{-1} (\tilde{R})^{*\top}\right) \Sigma_{\boldsymbol{x}}^{1/2} S \Sigma_{\boldsymbol{x}}^{1/2} \left(I_d - \tilde{R}^*((\tilde{R})^{*\top} \tilde{R}^*)^{-1} (\tilde{R})^{*\top}\right). \quad (65)$$

This is a symmetric matrix, whose top eigenvector is the $(r + 1)$th eigenvector $v_{r+1}(\Sigma_{\boldsymbol{x}}^{1/2} S \Sigma_{\boldsymbol{x}}^{1/2})$. $\square$

We next prove Theorem 3.

*Proof of Theorem 3.* We follow the proof strategy mentioned after the statement of the theorem. We assume that $\boldsymbol{n} \equiv 0$ with probability 1, since, as for the pure minimax regret, this unavoidable additive term of $\mathbb{E}\left[\mathbb{E}[\boldsymbol{n}^2 \mid \boldsymbol{x}]\right]$ to the loss does not affect the regret.

*The minimax problem – a direct computation:* As in the derivations leading to (60), the minimax regret in (2) is given by

$$
\begin{aligned}
& \mathsf{regret}_{\mathrm{mix}}(\mathcal{F}_S \mid \Sigma_{\boldsymbol{x}}) \\
&= \min_{\mathsf{L}(\boldsymbol{R}) \in \mathcal{P}(\mathcal{R})} \max_{f \in \mathcal{F}_S} \mathbb{E}\left[\mathsf{regret}(\boldsymbol{R}, f \mid \Sigma_{\boldsymbol{x}})\right] && (66)
\end{aligned}
$$

$$
= \min_{\mathsf{L}(\boldsymbol{R}) \in \mathcal{P}(\mathcal{R})} \max_{\tilde{f}:\|\tilde{f}\|_2^2 \leq 1} \mathbb{E}\left[\tilde{f}^{\top}\left(S^{1/2}\Sigma_{\boldsymbol{x}}S^{1/2} - S^{1/2}\Sigma_{\boldsymbol{x}}\boldsymbol{R}(\boldsymbol{R}^{\top}\Sigma_{\boldsymbol{x}}\boldsymbol{R})^{-1}\boldsymbol{R}^{\top}\Sigma_{\boldsymbol{x}}S^{1/2}\right)\tilde{f}\right] \quad (67)
$$

$$
= \min_{\mathsf{L}(\Sigma_{\boldsymbol{x}}^{-1/2}\tilde{\boldsymbol{R}}) \in \mathcal{P}(\mathcal{R})} \max_{\tilde{f}:\|\tilde{f}\|_2^2 \leq 1} \tilde{f}^{\top} S^{1/2}\Sigma_{\boldsymbol{x}}^{1/2}\mathbb{E}\left[I_d - \tilde{\boldsymbol{R}}(\tilde{\boldsymbol{R}}^{\top}\tilde{\boldsymbol{R}})^{-1}\tilde{\boldsymbol{R}}^{\top}\right]\Sigma_{\boldsymbol{x}}^{1/2}S^{1/2}\tilde{f} \quad (68)
$$

$$
= \min_{\mathsf{L}(\Sigma_{\boldsymbol{x}}^{-1/2}\tilde{\boldsymbol{R}}) \in \mathcal{P}(\mathcal{R})} \lambda_1\left(S^{1/2}\Sigma_{\boldsymbol{x}}^{1/2}\mathbb{E}\left[I_d - \tilde{\boldsymbol{R}}(\tilde{\boldsymbol{R}}^{\top}\tilde{\boldsymbol{R}})^{-1}\tilde{\boldsymbol{R}}^{\top}\right]\Sigma_{\boldsymbol{x}}^{1/2}S^{1/2}\right), \quad (69)
$$

where $\tilde{\boldsymbol{R}} = \Sigma_{\boldsymbol{x}}^{1/2}\boldsymbol{R}$. Determining the optimal distribution of the representation directly from this expression seems to be intractable. We thus next solve the maximin problem, and then return to the maximin problem (69), set a specific random representation, and show that it achieves the maximin value. This, in turn, establishes the optimality of this choice.

*The maximin problem:* Let an arbitrary $\mathsf{L}(\boldsymbol{f})$ be given. Then, taking the expectation of the regret over the random choice of $\boldsymbol{f}$, for any given $R \in \mathcal{R}$,

$$
\mathbb{E}\left[\mathsf{regret}(R, \boldsymbol{f})\right] \stackrel{(a)}{=} \mathbb{E}\left[\mathrm{Tr}\left[\left(S^{1/2}\Sigma_{\boldsymbol{x}}S^{1/2} - S^{1/2}\Sigma_{\boldsymbol{x}}R(R^{\top}\Sigma_{\boldsymbol{x}}R)^{-1}R^{\top}\Sigma_{\boldsymbol{x}}S^{1/2}\right)\tilde{\boldsymbol{f}}\tilde{\boldsymbol{f}}^{\top}\right]\right] \quad (70)
$$

$$
\stackrel{(b)}{=} \mathrm{Tr}\left[\left(S^{1/2}\Sigma_{\boldsymbol{x}}S^{1/2} - S^{1/2}\Sigma_{\boldsymbol{x}}R(R^{\top}\Sigma_{\boldsymbol{x}}R)^{-1}R^{\top}\Sigma_{\boldsymbol{x}}S^{1/2}\right)\tilde{\Sigma}_{\boldsymbol{f}}\right] \quad (71)
$$

$$
= \mathrm{Tr}\left[\tilde{\Sigma}_{\boldsymbol{f}}^{1/2}\left(S^{1/2}\Sigma_{\boldsymbol{x}}S^{1/2} - S^{1/2}\Sigma_{\boldsymbol{x}}R(R^{\top}\Sigma_{\boldsymbol{x}}R)^{-1}R^{\top}\Sigma_{\boldsymbol{x}}S^{1/2}\right)\tilde{\Sigma}_{\boldsymbol{f}}^{1/2}\right] \quad (72)
$$

$$
\stackrel{(c)}{=} \mathrm{Tr}\left[\tilde{\Sigma}_{\boldsymbol{f}}^{1/2}S^{1/2}\Sigma_{\boldsymbol{x}}^{1/2}\left(I_d - \tilde{R}(\tilde{R}^{\top}\tilde{R})^{-1}\tilde{R}^{\top}\right)\Sigma_{\boldsymbol{x}}^{1/2}S^{1/2}\tilde{\Sigma}_{\boldsymbol{f}}^{1/2}\right] \quad (73)
$$

$$
= \mathrm{Tr}\left[\left(I - \tilde{R}(\tilde{R}^{\top}\tilde{R})^{-1}\tilde{R}^{\top}\right)\Sigma_{\boldsymbol{x}}^{1/2}S^{1/2}\tilde{\Sigma}_{\boldsymbol{f}}S^{1/2}\Sigma_{\boldsymbol{x}}^{1/2}\right] \quad (74)
$$

$$
\stackrel{(d)}{=} \mathrm{Tr}\left[\left(I - \tilde{R}(\tilde{R}^{\top}\tilde{R})^{-1}\tilde{R}^{\top}\right)\Sigma_{\boldsymbol{x}}^{1/2}S^{1/2}\tilde{\Sigma}_{\boldsymbol{f}}S^{1/2}\Sigma_{\boldsymbol{x}}^{1/2}\left(I - \tilde{R}(\tilde{R}^{\top}\tilde{R})^{-1}\tilde{R}^{\top}\right)\right] \quad (75)
$$

$$
\stackrel{(e)}{\geq} \min_{W \in \mathbb{R}^{d \times (d-r)}:W^{\top}W=I_{d-r}} \mathrm{Tr}\left[W^{\top}\Sigma_{\boldsymbol{x}}^{1/2}S^{1/2}\tilde{\Sigma}_{\boldsymbol{f}}S^{1/2}\Sigma_{\boldsymbol{x}}^{1/2}W\right] \quad (76)
$$

$$
\stackrel{(f)}{=} \sum_{i=r+1}^{d}\lambda_i(\Sigma_{\boldsymbol{x}}^{1/2}S^{1/2}\tilde{\Sigma}_{\boldsymbol{f}}S^{1/2}\Sigma_{\boldsymbol{x}}^{1/2}) \quad (77)
$$

$$
= \sum_{i=r+1}^{d}\lambda_i\left(\tilde{\Sigma}_{\boldsymbol{f}}S^{1/2}\Sigma_{\boldsymbol{x}}S^{1/2}\right), \quad (78)
$$

where $(a)$ follows from Lemma 16 and setting $\tilde{\boldsymbol{f}} := S^{-1/2}\boldsymbol{f}$, $(b)$ follows by setting $\tilde{\Sigma}_{\boldsymbol{f}} \equiv \Sigma_{\tilde{\boldsymbol{f}}} = \mathbb{E}[\tilde{\boldsymbol{f}}\tilde{\boldsymbol{f}}^{\top}]$, $(c)$ follows by setting $\tilde{R} := \Sigma_{\boldsymbol{x}}^{1/2}R$, $(d)$ follows since $I - \tilde{R}(\tilde{R}^{\top}\tilde{R})^{-1}\tilde{R}^{\top}$ is an orthogonal projection (idempotent and symmetric matrix) of rank $d - r$, $(e)$ follows since any orthogonal projection can be written as $WW^{\top}$ where $W \in \mathbb{R}^{d \times (d-r)}$ is an orthogonal matrix $W^{\top}W = I_{d-r}$, $(f)$ follows from *Fan's variational characterization (Fan, 1949)*(Horn and Johnson, 2012, Corollary 4.3.39.) (see Appendix D). Equality in $(e)$ can be achieved by letting $\tilde{R}$ be the top $r$ eigenvectors of $\Sigma_{\boldsymbol{x}}^{1/2}S^{1/2}\tilde{\Sigma}_{\boldsymbol{f}}S^{1/2}\Sigma_{\boldsymbol{x}}^{1/2}$.

The next step of the derivation is to maximize the expected regret over the probability law of $\boldsymbol{f}$ (or $\tilde{\boldsymbol{f}}$). Evidently, $\mathbb{E}[\text{regret}(R, \boldsymbol{f})] = \sum_{i=r+1}^{d} \lambda_i(\tilde{\Sigma}_{\boldsymbol{f}} S^{1/2} \Sigma_{\boldsymbol{x}} S^{1/2})$ only depends on the random function $\tilde{\boldsymbol{f}}$ via $\tilde{\Sigma}_{\boldsymbol{f}}$. The covariance matrix $\tilde{\Sigma}_{\boldsymbol{f}}$ is constrained as follows. Recall that $\boldsymbol{f}$ is supported on $\mathcal{F}_S := \{f \in \mathbb{R}^d : \|f\|_S^2 \leq 1\}$ (see (4)), and let $\Sigma_{\boldsymbol{f}} = \mathbb{E}[\boldsymbol{f} \boldsymbol{f}^\top]$ be its covariance matrix. Then, it must hold that $\text{Tr}[S^{-1} \Sigma_{\boldsymbol{f}}] \leq 1$. Then, it also holds that

$$1 \geq \text{Tr}[S^{-1} \Sigma_{\boldsymbol{f}}] = \text{Tr}\left[\mathbb{E}[S^{-1} \boldsymbol{f} \boldsymbol{f}^\top]\right] \tag{79}$$

$$= \mathbb{E}\left[\boldsymbol{f}^\top S^{-1} \boldsymbol{f}\right] \tag{80}$$

$$= \mathbb{E}\left[\tilde{\boldsymbol{f}}^\top \tilde{\boldsymbol{f}}\right] \tag{81}$$

$$= \text{Tr}\left[\tilde{\Sigma}_{\boldsymbol{f}}\right] \tag{82}$$

where $\tilde{\Sigma}_{\boldsymbol{f}} = S^{-1/2} \Sigma_{\boldsymbol{f}} S^{-1/2}$. Conversely, given any covariance matrix $\tilde{\Sigma}_{\boldsymbol{f}} \in \mathbb{S}_{++}^d$ such that $\text{Tr}[\tilde{\Sigma}_{\boldsymbol{f}}] \leq 1$ there exists a random vector $\boldsymbol{f}$ supported on $\mathcal{F}_S$ such that

$$\mathbb{E}[\boldsymbol{f} \boldsymbol{f}^\top] = S^{1/2} \tilde{\Sigma}_{\boldsymbol{f}} S^{-1/2}. \tag{83}$$

We show this by an explicit construction. Let $\tilde{\Sigma}_{\boldsymbol{f}} = \tilde{V}_{\boldsymbol{f}} \tilde{\Lambda}_{\boldsymbol{f}} \tilde{V}_{\boldsymbol{f}}^\top$ be the eigenvalue decomposition of $\tilde{\Sigma}_{\boldsymbol{f}}$, and, for brevity, denote by $\tilde{\lambda}_i \equiv \lambda_i(\tilde{\Sigma}_{\boldsymbol{f}})$ the diagonal elements of $\tilde{\Lambda}_{\boldsymbol{f}}$. Let $\{\boldsymbol{q}_i\}_{i \in [d]}$ be a set of independent and identically (IID) distributed random variables, so that $\boldsymbol{q}_i$ is Rademacher, that is $\mathbb{P}[\boldsymbol{q}_i = 1] = \mathbb{P}[\boldsymbol{q}_i = -1] = 1/2$. Define the random vector

$$\boldsymbol{g} := \left(\boldsymbol{q}_1 \cdot \sqrt{\tilde{\lambda}_1}, \cdots, \boldsymbol{q}_d \cdot \sqrt{\tilde{\lambda}_d}\right)^\top. \tag{84}$$

The constraint $\text{Tr}[\tilde{\Sigma}_{\boldsymbol{f}}] \leq 1$ implies that $\sum \tilde{\lambda}_i \leq 1$ and so $\|\boldsymbol{g}\|^2 = \sum_{i=1}^d \tilde{\lambda}_i \leq 1$ with probability 1. Then, letting $\tilde{\boldsymbol{f}} = \tilde{V}_{\boldsymbol{f}} \boldsymbol{g}$ it also holds that $\|\tilde{\boldsymbol{f}}\|_2^2 = \|\boldsymbol{g}\|_2^2 \leq 1$ with probability 1, and furthermore,

$$\mathbb{E}\left[\tilde{\boldsymbol{f}} \tilde{\boldsymbol{f}}^\top\right] = \tilde{V}_{\boldsymbol{f}} \mathbb{E}\left[\boldsymbol{g} \boldsymbol{g}^\top\right] \tilde{V}_{\boldsymbol{f}}^\top = \tilde{V}_{\boldsymbol{f}} \tilde{\Lambda}_{\boldsymbol{f}} \tilde{V}_{\boldsymbol{f}}^\top = \tilde{\Sigma}_{\boldsymbol{f}}. \tag{85}$$

Consequently, letting $\boldsymbol{f} = S^{1/2} \boldsymbol{f}$ assures that $\|\boldsymbol{f}\|_S = \|\tilde{\boldsymbol{f}}\|_2 \leq 1$ and $\mathbb{E}[\boldsymbol{f} \boldsymbol{f}^\top] = S^{1/2} \tilde{\Sigma}_{\boldsymbol{f}} S^{-1/2}$, as was required to obtain. Therefore, instead of maximizing over probability laws on $\mathcal{P}(\mathcal{F}_S)$, we may equivalently maximize over $\tilde{\Sigma}_{\boldsymbol{f}} \in \mathbb{S}_{++}^d$ such that $\text{Tr}[\tilde{\Sigma}_{\boldsymbol{f}}] \leq 1$, i.e., to solve

$$\text{regret}_{\text{mix}}(\mathcal{F}_S \mid \Sigma_{\boldsymbol{x}}) = \max_{\tilde{\Sigma}_{\boldsymbol{f}} : \text{Tr}[\tilde{\Sigma}_{\boldsymbol{f}}] \leq 1} \sum_{i=r+1}^{d} \lambda_i(\tilde{\Sigma}_{\boldsymbol{f}} S^{1/2} \Sigma_{\boldsymbol{x}} S^{1/2}). \tag{86}$$

The optimization problem in (86) is solved in Lemma 17, and is provided after this proof. Setting $\Sigma = S^{1/2} \Sigma_{\boldsymbol{x}} S^{1/2}$ in Lemma 17, and letting $\lambda_i \equiv \lambda_i(S^{1/2} \Sigma_{\boldsymbol{x}} S^{1/2})$, the solution is given by

$$\frac{\ell^* - r}{\sum_{i=1}^{\ell^*} \frac{1}{\lambda_i}} \tag{87}$$

where $\ell^* \in [d] \setminus [r]$ satisfies

$$\frac{\ell^* - r}{\lambda_{\ell^*}} \leq \sum_{i=1}^{\ell^*} \frac{1}{\lambda_i} \leq \frac{\ell^* - r}{\lambda_{\ell^* + 1}}. \tag{88}$$

Lemma 17 also directly implies that an optimal $\tilde{\Sigma}_{\boldsymbol{f}}$ is given as in (10). The value in (87) is exactly $\text{regret}_{\text{mix}}(\mathcal{F}_S \mid \Sigma_{\boldsymbol{x}})$ claimed by the theorem, and we next show it is indeed achievable by a properly constructed random representation.

*The minimax problem – a solution via the maximin certificate:* Given the value of the regret game in mixed strategies found in (87), we may also find a minimax representation in mixed strategies. To this end, we return to the minimax expression in (69), and propose a random representation which achieves the maximin value in (87). Note that for any given $\tilde{R}$, the matrix $I_d - \tilde{R}(\tilde{R}^\top \tilde{R})^{-1} \tilde{R}^\top$ is an orthogonal projection, that is, a symmetric matrix whose eigenvalues are all either 0 or 1, and it has at

most $r$ eigenvalues equal to zero. We denote its eigenvalue decomposition by $I_d - \tilde{R}(\tilde{R}^\top \tilde{R})^{-1}\tilde{R}^\top = U\Omega U^\top$. Then, any probability law on $\tilde{R}$ induces a probability law on $U$ and $\Omega$ (and vice-versa). To find the mixed minimax representation, we propose setting $U = V(\Sigma_{\boldsymbol{x}}^{1/2}S\Sigma_{\boldsymbol{x}}^{1/2}) \equiv V$ with probability 1, that is, to be deterministic, and thus only randomize $\Omega$. With this choice, and by denoting, for brevity, $\Lambda \equiv \Lambda(\Sigma_{\boldsymbol{x}}^{1/2}S\Sigma_{\boldsymbol{x}}^{1/2}) = \Lambda(S^{1/2}\Sigma_{\boldsymbol{x}}S^{1/2})$, the value of the objective function in (69) is given by

$$\lambda_1\left(S^{1/2}\Sigma_{\boldsymbol{x}}^{1/2}V \cdot \mathbb{E}\left[\Omega\right] \cdot V^\top \Sigma_{\boldsymbol{x}}^{1/2}S^{1/2}\right)$$

$$= \lambda_1\left(\mathbb{E}\left[\Omega\right] \cdot V^\top \Sigma_{\boldsymbol{x}}^{1/2}S\Sigma_{\boldsymbol{x}}^{1/2}V\right) \tag{89}$$

$$= \lambda_1\left(\mathbb{E}\left[\Omega\right] \cdot \Lambda\right). \tag{90}$$

Now, the distribution of $\Omega$ is equivalent to a distribution on its diagonal, which is supported on the finite set $\mathcal{A} := \{a \in \{0,1\}^d : \|a\|_1 \geq d - r\}$. Our goal is thus to find a probability law on $\boldsymbol{a}$, supported on $\mathcal{A}$, which solves

$$\min_{\mathsf{L}(\Omega)}\max_{i\in[d]} \lambda_1\left(\mathbb{E}\left[\Omega\right] \cdot \Lambda\right) = \min_{\mathsf{L}(\boldsymbol{a})}\max_{i\in[d]} \mathbb{E}[\boldsymbol{a}_i]\lambda_i \tag{91}$$

where $\lambda_i \equiv \lambda_i(S^{1/2}\Sigma_{\boldsymbol{x}}S^{1/2})$ are the diagonal elements of $\Lambda$. Consider $\ell^*$, the optimal dimension of the maximin problem, which satisfies (88). We then set $\boldsymbol{a}_{\ell^*+1} = \cdots = \boldsymbol{a}_d = 1$ to hold with probability 1, and so it remains to determine the probability law of $\overline{\boldsymbol{a}} := (\boldsymbol{a}_1, \ldots, \boldsymbol{a}_{\ell^*})$, supported on $\tilde{\mathcal{A}} := \{a \in \{0,1\}^{\ell^*} : \|a\|_1 \geq \ell^* - r\}$. Clearly, reducing $\|a\|_1$ only reduces the objective function $\max_{i\in[d]} \mathbb{E}[\boldsymbol{a}_i\lambda_i]$, and so we may in fact assume that $\overline{\boldsymbol{a}}$ is supported on $\overline{\mathcal{A}} := \{\overline{a} \in \{0,1\}^{\ell^*} : \|\overline{a}\|_1 = \ell^* - r\}$, a finite subset of cardinality $\binom{\ell^*}{r}$. Suppose that we find a probability law $\mathsf{L}(\overline{\boldsymbol{a}})$ supported on $\overline{\mathcal{A}}$ such that

$$\mathbb{E}[\boldsymbol{a}_i] = (\ell^* - r) \cdot \frac{1/\lambda_i}{\sum_{i=1}^{\ell^*} 1/\lambda_i} := b_i, \tag{92}$$

for all $i \in [\ell^*]$. Then, since $\mathbb{E}[\boldsymbol{a}_i] = 1$ for $i \in [d]\setminus[\ell^*]$

$$\max_{i\in[d]} \mathbb{E}[\boldsymbol{a}_i]\lambda_i = \max\left\{\frac{\ell^* - r}{\sum_{i=1}^{\ell^*} \frac{1}{\lambda_i}}, \lambda_{\ell^*+1}, \cdots, \lambda_d\right\} \tag{93}$$

$$= \max\left\{\frac{\ell^* - r}{\sum_{i=1}^{\ell^*} \frac{1}{\lambda_i}}, \lambda_{\ell^*+1}\right\} \tag{94}$$

$$\overset{(*)}{=} \frac{\ell^* - r}{\sum_{i=1}^{\ell^*} \frac{1}{\lambda_i}}, \tag{95}$$

where $(*)$ follows from the condition on $\ell^*$ in the right inequality of (88). This proves that such probability law achieves the minimax regret in mixed strategies. This last term is $\mathrm{regret}_{\mathsf{mix}}(\mathcal{F}_S \mid \Sigma_{\boldsymbol{x}})$ claimed by the theorem. It remains to construct $\mathsf{L}(\overline{\boldsymbol{a}})$ which satisfies (92). To this end, note that the set

$$\mathcal{C} := \left\{c \in [0,1]^{\ell^*} : \|c\|_1 = \ell^* - r\right\} \tag{96}$$

is convex and compact, and $\overline{\mathcal{A}}$ is the set of its *extreme points* ($\mathcal{C}$ is the convex hull of $\overline{\mathcal{A}}$). Letting $\overline{b} = (b_1, \ldots, b_{\ell^*})^\top$ as denoted in (92), it holds that $\overline{b}_i \geq 0$ and $\{\overline{b}_i\}_{i=1}^{\ell^*}$ is a non-decreasing sequence. Using the condition on $\ell^*$ in the left inequality of (88), it then holds that

$$\overline{b}_1 \leq \cdots \leq \overline{b}_{\ell^*} = (\ell^* - r) \cdot \frac{1/\lambda_{\ell^*}}{\sum_{i=1}^{\ell^*} 1/\lambda_i} \leq 1. \tag{97}$$

Hence, $\overline{b} \in \mathcal{C}$. By Carathéodory's theorem (Bertsekas et al., 2003, Prop. 1.3.1) (see Appendix D), any point inside a convex compact set in $\mathbb{R}^{\ell^*}$ can be written as a convex combination of at most $\ell^* + 1$ extreme points. Thus, there exists $\{p_{\overline{a}}\}_{\overline{a}\in\overline{\mathcal{A}}}$ such that $p_{\overline{a}} \in [0,1]$ and $\sum_{\overline{a}\in\overline{\mathcal{A}}} p_{\overline{a}} = 1$ so that $\overline{b} = \sum_{\overline{a}\in\overline{\mathcal{A}}} p_{\overline{a}} \cdot \overline{a}$, and moreover the support of $p_{\overline{a}}$ has cardinality at most $\ell^* + 1$. Let $\overline{A} \in \{0,1\}^{\ell^* \times |\overline{\mathcal{A}}|}$ be such that its $j$th column is given by the $j$th member of $\overline{\mathcal{A}}$ (in an arbitrary order). Let $p \in [0,1]^{|\overline{\mathcal{A}}|}$

be a vector whose $j$th element corresponds to the $j$th member of $\overline{\mathcal{A}}$. Then, $p$ is the solution to $\overline{A}p = \overline{b}$, and as claimed above, such a solution with at most $\ell^* + 1$ nonzero entries always exists. Setting $\boldsymbol{a} = (\overline{a}, \underbrace{1\ldots,1}_{d-\ell^* \text{ terms}})$ with probability $p_{\overline{a}}$ then assures that (92) holds, as was required to be proved.

Given the above, we observe that setting $\tilde{R}$ as in the theorem induces a distribution on $\boldsymbol{\Omega}$ for which the random entries of its diagonal $\boldsymbol{a}$ satisfy (92), and thus achieve $\mathrm{regret}_{\mathrm{mix}}(\mathcal{F}_S \mid \Sigma_{\boldsymbol{x}})$. $\qquad\square$

We next turn to complete the proof of Theorem 3 by solving the optimization problem in (86). Assume that $\Sigma \in \mathbb{S}_{++}^d$ is a strictly positive covariance matrix $\Sigma \succ 0$, and consider the optimization problem

$$v_r^* = \max_{\tilde{\Sigma}_{\boldsymbol{f}} \in \mathbb{S}_+^d} \sum_{i=r+1}^d \lambda_i(\tilde{\Sigma}_{\boldsymbol{f}}\Sigma)$$

$$\text{subject to } \mathrm{Tr}[\tilde{\Sigma}_{\boldsymbol{f}}] \le 1 \tag{98}$$

for some $r \in [d-1]$. Note that the objective function refers to the maximization of the $d-r$ minimal eigenvalues of $\Sigma^{1/2}\tilde{\Sigma}_{\boldsymbol{f}}\Sigma^{1/2}$.

**Lemma 17.** *Let*

$$a_\ell := \frac{\ell - r}{\sum_{i=1}^\ell \frac{1}{\lambda_i(\Sigma)}}. \tag{99}$$

*The optimal value of (98) is $v^* = \max_{[d]\setminus[r]} a_\ell$ and $\ell^* \in \mathrm{argmax}_{[d]\setminus[r]} a_\ell$ iff*

$$\frac{\ell^* - r}{\lambda_{\ell^*}(\Sigma)} \le \sum_{i=1}^{\ell^*} \frac{1}{\lambda_i(\Sigma)} \le \frac{\ell^* - r}{\lambda_{\ell^*+1}(\Sigma)}. \tag{100}$$

*An optimal solution is*

$$\tilde{\Sigma}_{\boldsymbol{f}}^* = \left[\sum_{i=1}^{\ell^*} \frac{1}{\lambda_i(\Sigma)}\right]^{-1} \cdot V(\Sigma)\,\mathrm{diag}\left(\frac{1}{\lambda_1(\Sigma)}, \ldots, \frac{1}{\lambda_{\ell^*}(\Sigma)}, 0, \cdots, 0\right) \cdot V(\Sigma)^\top. \tag{101}$$

*Proof.* Let $\overline{\Sigma}_{\boldsymbol{f}} = \Sigma^{1/2}\tilde{\Sigma}_{\boldsymbol{f}}\Sigma^{1/2}$, let $\overline{\Sigma}_{\boldsymbol{f}} = \overline{U}_{\boldsymbol{f}}\overline{\Lambda}_{\boldsymbol{f}}\overline{U}_{\boldsymbol{f}}^\top$ be its eigenvalue decomposition, and, for brevity, denote $\overline{\lambda}_i \equiv \lambda_i(\overline{\Sigma}_{\boldsymbol{f}})$. Then, the trace operation appearing in the constraint of (98) can be written as

$$\mathrm{Tr}[\tilde{\Sigma}_{\boldsymbol{f}}] = \mathrm{Tr}\left[\Sigma^{-1/2}\overline{\Sigma}_{\boldsymbol{f}}\Sigma^{-1/2}\right] \tag{102}$$

$$= \mathrm{Tr}\left[\Sigma^{-1/2}\overline{U}_{\boldsymbol{f}}\overline{\Lambda}_{\boldsymbol{f}}\overline{U}_{\boldsymbol{f}}^\top\Sigma^{-1/2}\right] \tag{103}$$

$$= \mathrm{Tr}\left[\Sigma^{-1/2}\left(\sum_{i=1}^d \lambda_i\overline{u}_i\overline{u}_i^\top\right)\Sigma^{-1/2}\right] \tag{104}$$

$$= \sum_{i=1}^d \overline{\lambda}_i \cdot \left(\overline{u}_i^\top\Sigma^{-1}\overline{u}_i\right) \tag{105}$$

$$= \sum_{i=1}^d c_i\overline{\lambda}_i, \tag{106}$$

where $\overline{u}_i = v_i(\overline{U}_{\boldsymbol{f}})$ (that is, the $i$th column of $\overline{U}_{\boldsymbol{f}}$), and $c_i := \overline{u}_i^\top\Sigma^{-1}\overline{u}_i$ (which satisfies $c_i > 0$). Thus, the optimization problem in (98) over $\tilde{\Sigma}_{\boldsymbol{f}}$ is equivalent to an optimization problem over $\{\overline{\lambda}_i, \overline{u}_i\}_{i\in[d]}$, given by

$$v_r^* = \max_{\{\overline{u}_i, \overline{\lambda}_i\}_{i\in[d]}} \sum_{i=r+1}^d \overline{\lambda}_i$$

$$\text{subject to } \sum_{i=1}^{d} c_i \overline{\lambda}_i \leq 1,$$
$$c_i = \overline{u}_i^\top \Sigma^{-1} \overline{u}_i,$$
$$\overline{u}_i^\top \overline{u}_j = \delta_{ij},$$
$$\overline{\lambda}_1 \geq \overline{\lambda}_2 \geq \cdots \geq \overline{\lambda}_d \geq 0. \tag{107}$$

To solve the optimization problem (107), let us fix feasible $\{\overline{u}_i\}_{i \in [d]}$, so that $\{c_i\}_{i \in [d]}$ are fixed too. This results the problem

$$v_r^*(\{\overline{u}_i\}) \equiv v_r^*(\{c_i\}) = \max_{\{\overline{\lambda}_i\}_{i \in [d]}} \sum_{i=r+1}^{d} \overline{\lambda}_i$$

$$\text{subject to } \sum_{i=1}^{d} c_i \overline{\lambda}_i \leq 1,$$
$$\overline{\lambda}_1 \geq \overline{\lambda}_2 \geq \cdots \geq \overline{\lambda}_d \geq 0. \tag{108}$$

The objective function of (108) is linear in $\{\overline{\lambda}_i\}_{i \in [d]}$ and its constraint set is a convex bounded polytope. So the solution to (108) must be obtained on the boundary of the constraint set. Clearly, the optimal value satisfies $v_r^*(\{c_i\}) \geq 0$, and thus the solution $\{\overline{\lambda}_i^*\}_{i \in [d]}$ must be obtained when the constraint $\sum_{i=1}^{d} c_i \overline{\lambda}_i \leq 1$ is satisfied with equality. Indeed, if this is not the case then one may scale all $\overline{\lambda}_i^*$ by a constant larger than 1, and obtain larger value of the objective, while still satisfying the constraint.

To find the optimal solution to (108), we consider feasible points for which $\ell := \max\{i \in [d]: \overline{\lambda}_i > 0\}$ is fixed. Let $\{\overline{\lambda}_i^*\}_{i \in [d]}$ be the optimal solution of (108), under the additional constraint that $\overline{\lambda}_{\ell+1} = \cdots = \overline{\lambda}_d = 0$. We next prove that $\overline{\lambda}_1^* = \cdots = \overline{\lambda}_\ell^*$ must hold. To this end, assume by contradiction that there exists $j \in [\ell]$ so that $\overline{\lambda}_{j-1}^* > \overline{\lambda}_j^* > 0$. There are two cases to consider, to wit, whether $j - 1 < r + 1$ and so only $\overline{\lambda}_j$ appears in the objective of (108), or, otherwise, $j - 1 \geq r + 1$ and then $\overline{\lambda}_{j-1} + \overline{\lambda}_j$ appears in the objective of (108). Assuming the first case, let $\alpha = \overline{\lambda}_{j-1}^* c_{j-1} + \overline{\lambda}_j^* c_j$ and consider the optimization problem

$$\max_{\hat{\lambda}_{j-1}, \hat{\lambda}_j} \hat{\lambda}_j$$
$$\text{subject to } \hat{\lambda}_{j-1} c_{j-1} + \hat{\lambda}_j c_j = \alpha,$$
$$\hat{\lambda}_{j-1} \geq \hat{\lambda}_j > 0. \tag{109}$$

It is easy to verify that the optimum of this problem is $\hat{\lambda}_{j-1}^* = \hat{\lambda}_j^* = \frac{\alpha}{c_{j-1}+c_j}$. Thus, if $\overline{\lambda}_{j-1}^* > \overline{\lambda}_j^*$ then one can replace this pair with $\overline{\lambda}_{j-1}^* = \overline{\lambda}_j^* = \hat{\lambda}_{j-1}^* = \hat{\lambda}_j^*$ so that the value of the constraint $\sum_{i=1}^{d} \overline{\lambda}_i^* c_i$ remains the same, and thus $(\overline{\lambda}_1^*, \cdots, \hat{\lambda}_{j-1}^*, \hat{\lambda}_j^*, \overline{\lambda}_{j+1}^*, \ldots \overline{\lambda}_d^*)$ is a feasible point, while the objective function value of (108) is smaller; a contradiction. Therefore, it must hold for the first case that $\overline{\lambda}_{j-1}^* = \overline{\lambda}_j^*$. For the second case, in a similar fashion, let now $\alpha = \overline{\lambda}_{j-1}^* c_{j-1} + \overline{\lambda}_j^* c_j$, and consider the optimization problem

$$\max_{\hat{\lambda}_{j-1}, \hat{\lambda}_j} \hat{\lambda}_j + \hat{\lambda}_{j-1}$$
$$\text{subject to } \hat{\lambda}_{j-1} c_{j-1} + \hat{\lambda}_j c_j = \alpha,$$
$$\hat{\lambda}_{j-1} \geq \hat{\lambda}_j > 0. \tag{110}$$

The solution for this optimization problem is at one of the two extreme points of the feasible interval for $\hat{\lambda}_j$. Since $\lambda_j^* > 0$ was assumed it therefore must hold that $\hat{\lambda}_{j-1}^* = \hat{\lambda}_j^*$, and hence also $\overline{\lambda}_{j-1}^* = \overline{\lambda}_j^*$. Thus, $\lambda_{j-1}^* < \lambda_j^*$ leads to a contradiction. From the above, we deduce that the optimal solution of (108) under the additional constraint that $\overline{\lambda}_{\ell+1} = \cdots = \overline{\lambda}_d = 0$ is

$$\overline{\lambda}_1^* = \cdots = \overline{\lambda}_\ell^* = \frac{1}{\sum_{i=1}^{\ell} c_i} \tag{111}$$

$$\overline{\lambda}^*_{\ell+1} = \cdots = \overline{\lambda}^*_d = 0, \tag{112}$$

and that the optimal value is $\frac{\ell-r}{\sum^{\ell}_{i=1} c_i}$. Since $\ell \in [d]\backslash[r]$ can be arbitrarily chosen, we deduce that the value of (108) is

$$v^*(\{c_i\}) = \max_{\ell \in [d]\backslash[r]} \frac{\ell - r}{\sum^{\ell}_{i=1} c_i}. \tag{113}$$

For any given $\ell \in [d]\backslash[r]$, we may now optimize over $\{\overline{u}_i\}$, which from (113) is equivalent to minimizing $\sum^{\ell}_{i=1} c_i$. It holds that

$$\min_{\{\overline{u}_i\}} \sum^{\ell}_{i=1} c_i = \min_{\{\overline{u}_i : \overline{u}_i^\top \overline{u}_j = \delta_{ij}\}} \sum^{\ell}_{i=1} \overline{u}_i^\top \Sigma^{-1} \overline{u}_i \tag{114}$$

$$= \min_{\{\overline{u}_i : \overline{u}_i^\top \overline{u}_j = \delta_{ij}\}} \mathrm{Tr} \left[ \Sigma^{-1} \sum^{\ell}_{i=1} \overline{u}_i \overline{u}_i^\top \right] \tag{115}$$

$$\overset{(a)}{=} \min_{\grave{U} \in \mathbb{R}^{d \times \ell} : \grave{U}^\top \grave{U} = I_\ell} \mathrm{Tr} \left[ \Sigma^{-1} \grave{U} \grave{U}^\top \right] \tag{116}$$

$$= \min_{\grave{U} \in \mathbb{R}^{d \times \ell} : \grave{U}^\top \grave{U} = I_\ell} \mathrm{Tr} \left[ \grave{U}^\top \Sigma^{-1} \grave{U} \right] \tag{117}$$

$$\overset{(b)}{=} \sum^{\ell}_{i=1} \frac{1}{\lambda_i(\Sigma)}, \tag{118}$$

where in $(a)$ $\grave{U} \in \mathbb{R}^{d \times \ell}$ whose $\ell$ columns are $\{\overline{u}_i\}_{i \in [\ell]}$ and $\grave{U}^\top \grave{U} = I_\ell$, and in $(b)$ we have used *Fan (1949)'s variational characterization* (Horn and Johnson, 2012, Corollary 4.3.39.) (see Appendix D). Substituting back to (113) results that

$$v_r^* = \max_{\ell \in [d]\backslash[r]} \frac{\ell - r}{\sum^{\ell}_{i=1} \frac{1}{\lambda_i(\Sigma)}} = \max_{\ell \in [d]\backslash[r]} a_\ell. \tag{119}$$

Let us denote that maximizer index by $\ell^*$. Then, Fan's characterization is achieved by setting $\overline{U}_{\boldsymbol{f}} = V$ (so that the $\ell^*$ columns of $\grave{U}$ are the $\ell^*$ eigenvectors $v_i(\Sigma)$, corresponding to the $\ell^*$ largest eigenvalues of $\Sigma$), so that

$$\overline{\Sigma}^*_{\boldsymbol{f}} = \left[ \sum^{\ell^*}_{i=1} \frac{1}{\lambda_i(\Sigma)} \right]^{-1} \cdot V \cdot \mathrm{diag} \left( \underbrace{1, \ldots, 1}_{\ell^* \text{ terms}}, 0, \cdots, 0 \right) \cdot V^\top, \tag{120}$$

and then

$$\tilde{\Sigma}^*_{\boldsymbol{f}} = \Sigma^{-1/2} \overline{\Sigma}^*_{\boldsymbol{f}} \Sigma^{-1/2} \tag{121}$$

$$= \left[ \sum^{\ell^*}_{i=1} \frac{1}{\lambda_i(\Sigma)} \right]^{-1} \cdot V\Lambda^{-1/2}V^\top V \cdot \mathrm{diag}(1, \ldots, 1, 0, \cdots, 0) V^\top V\Lambda^{-1/2}V^\top \tag{122}$$

$$= \left[ \sum^{\ell^*}_{i=1} \frac{1}{\lambda_i(\Sigma)} \right]^{-1} \cdot V \cdot \mathrm{diag} \left( \frac{1}{\lambda_1(\Sigma)}, \ldots, \frac{1}{\lambda_{\ell^*}(\Sigma)}, 0, \cdots, 0 \right) \cdot V^\top \tag{123}$$

as claimed in (101).

To complete the proof, it remains to characterize $\ell^*$, which belongs to the set possible indices maximizing $\{a_\ell\}_{\ell \in [d]\backslash[r]}$. Since $\ell^*$ maximizes $a_\ell$ it must be a local maximizer, that is, it must hold that $a_{\ell^*-1} \leq a_{\ell^*} \geq a_{\ell^*+1}$. By simple algebra, these conditions are equivalent to those in (100). It remains to show that any $\ell \in [d]\backslash[r]$ which satisfies (100) has the same value, and thus any local maxima is a global maxima. We will show this by proving that the sequence $\{a_\ell\}^d_{\ell=r}$ is *unimodal*, as follows. Let $\Delta_\ell := a_{\ell+1} - a_\ell$ be the discrete derivative of $\{a_\ell\}_{\ell \in [d]}$, and consider the sequence $\{\Delta_\ell\}_{\ell \in [d]\backslash[r]}$. We show that as $\ell$ increases from $r$ to $d$, $\{\Delta_\ell\}_{\ell \in [d]\backslash[r]}$ is only changing its sign at most once. To this end, we first note that

$$\Delta_\ell = \frac{\ell+1-r}{\sum^{\ell+1}_{i=1} \frac{1}{\lambda_i(\Sigma)}} - \frac{\ell-r}{\sum^{\ell}_{i=1} \frac{1}{\lambda_i(\Sigma)}} = \frac{\sum^{\ell}_{i=1} \frac{1}{\lambda_i(\Sigma)} - (\ell-r)\frac{1}{\lambda_{\ell+1}(\Sigma)}}{\left[ \sum^{\ell+1}_{i=1} \frac{1}{\lambda_i(\Sigma)} \right] \left[ \sum^{\ell}_{i=1} \frac{1}{\lambda_i(\Sigma)} \right]}. \tag{124}$$

Since the denominator of (124) is strictly positive, it suffices to prove that the sequence comprised of the numerator of (124), to wit $\{\zeta_\ell\}_{\ell \in [d] \backslash [r]}$ with

$$\zeta_\ell := \sum_{i=1}^{\ell} \frac{1}{\lambda_i(\Sigma)} - (\ell - r) \frac{1}{\lambda_{\ell+1}(\Sigma)}, \tag{125}$$

is only changing its sign at most once. Indeed, this claim is true because $\zeta_r = \sum_{i=1}^{\ell} \frac{1}{\lambda_i(\Sigma)} > 0$ and because $\{\zeta_\ell\}_{\ell \in [d] \backslash [r]}$ is a monotonic non-increasing sequence,

$$\zeta_\ell - \zeta_{\ell+1} = (\ell - r + 1) \left[ \frac{1}{\lambda_{\ell+2}(\Sigma)} - \frac{1}{\lambda_{\ell+1}(\Sigma)} \right] \geq 0. \tag{126}$$

Therefore, $\{\zeta_\ell\}_{\ell \in [d] \backslash [r]}$ has at most a single sign change (its has a positive value at $\ell = r$ and is monotonically non-increasing with $\ell$ up to $\ell = d$), and so is $\{\Delta_\ell\}_{\ell=r}^d$. The single sign change property of the finite difference $\{\Delta_\ell\}_{\ell=r}^d$ is equivalent to the fact that $\{a_\ell\}_{\ell=r}^d$ is *unimodal*. Thus, any local maximizer of $a_\ell$ is also a global maximizer. $\square$

## F   THE HILBERT SPACE MSE SETTING

In this section, we show that the regret expressions in Section 3 can be easily generalized to an infinite dimensional Hilbert space, for responses with noise that is statistically independent of the features. We still assume the MSE loss function ($\mathcal{Y} = \mathbb{R}$, and $\text{loss}(y_1, y_2) = (y_1 - y_2)^2$), and that the predictor is a linear function. However, we allow the the representation and response function to be functions in a Hilbert space. As will be evident, the resulting regret is not very different from the finite-dimensional case. Formally, this is defined as follows:

**Definition 18** (The Hilbert space MSE setting). Assume that $\boldsymbol{x} \sim P_{\boldsymbol{x}}$ is supported on a compact subset $\mathcal{X} \subset \mathbb{R}^d$, and let $L_2(P_{\boldsymbol{x}})$ be the Hilbert space of functions from $\mathcal{X} \to \mathbb{R}$ such that $\mathbb{E}[f^2(\boldsymbol{x})] = \int_{\mathcal{X}} f^2(\boldsymbol{x}) \cdot \mathrm{d}P_{\boldsymbol{x}} < \infty$, with the inner product,

$$\langle f, g \rangle := \int_{\mathcal{X}} f(\boldsymbol{x}) g(\boldsymbol{x}) \cdot \mathrm{d}P_{\boldsymbol{x}} \tag{127}$$

for $f, g \in L_2(P_{\boldsymbol{x}})$. Let $\{\phi_j(x)\}_{j=1}^\infty$ be an orthonormal basis for $L_2(P_{\boldsymbol{x}})$.

A representation is comprised of a set of functions $\{\psi_i\}_{i \in [r]} \subset L_2(P_{\boldsymbol{x}})$, $\psi_i \colon \mathcal{X} \to \mathbb{R}$, so that

$$\mathcal{R} := \{R(x) = (\psi_1(x), \ldots, \psi_r(x))^\top \in \mathbb{R}^r\}. \tag{128}$$

Let $\{\lambda_j\}_{j \in \mathbb{N}}$ be a positive monotonic non-increasing sequence for which $\lambda_j \downarrow 0$ as $j \to \infty$, and let $\mathcal{F}$ be the set of functions from $\mathcal{X} \to \mathbb{R}$ such that given $f \in \mathcal{F}$, the response is given by

$$\boldsymbol{y} = f(\boldsymbol{x}) + \boldsymbol{n} \in \mathbb{R} \tag{129}$$

where

$$f \in \mathcal{F}_{\{\lambda_j\}} := \left\{ f(x) = \sum_{j=1}^\infty f_j \phi_j(x) \colon \{f_j\}_{j \in \mathbb{N}} \in \ell_2(\mathbb{N}), \quad \sum_{j=1}^\infty \frac{f_j^2}{\lambda_j} \leq 1 \right\}, \tag{130}$$

where $\boldsymbol{n} \in \mathbb{R}$ is a homoscedastic noise that is statistically independent of $\boldsymbol{x}$ and satisfies $\mathbb{E}[\boldsymbol{n}] = 0$. Infinite-dimensional ellipsoids such as $\mathcal{F}_{\{\lambda_j\}}$ naturally arise in reproducing kernel Hilbert spaces (RKHS) (Wainwright, 2019, Chapter 12) (Shalev-Shwartz and Ben-David, 2014, Chapter 16), in which $\{\lambda_j\}$ is the eigenvalues of the kernel. In this case, the set $\mathcal{F}_{\{\lambda_i\}} = \{f \colon \|f\|_{\mathcal{H}} \leq 1\}$ where $\|\cdot\|_{\mathcal{H}}$ is the norm of the RKHS $\mathcal{H}$. For example, $\mathcal{H}$ could be the first-order Sobolev space of functions with finite first derivative energy.

Let the set of predictor functions be the set of linear functions from $\mathbb{R}^d \to \mathbb{R}$, that is

$$\mathcal{Q} := \{Q(z) = q^\top z = \sum_{i=1}^r q_i \cdot \psi_i(x), \; q \in \mathbb{R}^r\}. \tag{131}$$

We denote the pure (resp. mixed) minimax regret as $\text{regret}_{\text{pure}}(\mathcal{R}, \mathcal{F}_{\{\lambda_j\}} \mid P_{\boldsymbol{x}})$ (resp. $\text{regret}_{\text{mix}}(\mathcal{R}, \mathcal{F}_{\{\lambda_j\}} \mid P_{\boldsymbol{x}})$). We begin with pure strategies.

**Theorem 19.** *For the Hilbert space MSE setting (Definition 18)*

$$\text{regret}_{\text{pure}}(\mathcal{R}, \mathcal{F}_{\{\lambda_j\}} \mid P_{\boldsymbol{x}}) = \lambda_{r+1}. \tag{132}$$

*A minimax representation is*

$$R^*(x) = (\phi_1(x), \dots, \phi_r(x))^\top, \tag{133}$$

*and the worst case response function is $f^* = \sqrt{\lambda_{r+1}} \cdot \phi_{r+1}$.*

We now turn to the minimax representation in mixed strategies.

**Theorem 20.** *For the Hilbert space MSE setting (Definition 18)*

$$\text{regret}_{\text{mix}}(\mathcal{R}, \mathcal{F}_{\{\lambda_j\}} \mid P_{\boldsymbol{x}}) = \frac{\ell^* - r}{\sum_{i=1}^{\ell^*} \frac{1}{\lambda_i}}, \tag{134}$$

*where $\ell^*$ is defined as (8) of Theorem 3 (with the replacement $d \to \mathbb{N}_+$). Let $\{\boldsymbol{b}_j\}_{i=1}^\infty$ be an IID sequence of Rademacher random variables, $\mathbb{P}[\boldsymbol{b}_i = 1] = \mathbb{P}[\boldsymbol{b}_i = -1] = 1/2$. Then, a least favorable prior $\boldsymbol{f}^*$ is*

$$\boldsymbol{f}_i^* = \begin{cases} \boldsymbol{b}_i \cdot \dfrac{1}{\sqrt{\sum_{i=1}^{\ell^*} \frac{1}{\lambda_i}}}, & 1 \le i \le \ell_* \\ 0, & i \ge \ell_* + 1 \end{cases}, \tag{135}$$

*and a law of minimax representation is to choose*

$$\boldsymbol{R}^*(x) = \{\phi_{\mathcal{I}_j}(x)\}_{j=1}^r \tag{136}$$

*with probability $p_j$, $j \in [\binom{\ell^*}{r}]$, defined as in Theorem 3.*

**Discussion** Despite having countably infinite possible number of representations, the optimal representation only utilizes a *finite* set of orthogonal functions, as determined by the radius of $\mathcal{F}_{\{s_i\}}$. The proof of Theorems 19 and 20 is obtained by reducing the infinite dimensional problem to a $d$-dimensional problem via an approximation argument, then showing the the finite dimensional case is similar to the problem of Section 3, and then taking limit $d \uparrow \infty$.

### F.1 PROOFS

Let us denote the $d$-dimensional *slice* of $\mathcal{F}_{\{\lambda_j\}}$ by

$$\mathcal{F}_{\{\lambda_j\}}^{(d)} := \left\{ f(x) \in \mathcal{F}_{\{\lambda_j\}} : f_j = 0 \text{ for all } j \ge d+1 \right\}. \tag{137}$$

Further, let us consider the restricted representation class, in which the representation functions $\psi_i(t)$ belong to the span of the first $d$ basis functions, that is

$$\mathcal{R}^{(d)} := \{ R(x) \in \mathcal{R} := \psi_i(x) \in \text{span}(\{\phi_i\}_{i \in [d]}) \text{ for all } i \in [r]\}. \tag{138}$$

The following proposition implies that the regret in the infinite-dimensional Hilbert space is obtained as the limit of finite-dimensional regrets, as the one characterized in Section 3:

**Proposition 21.** *It holds that*

$$\text{regret}_{\text{pure}}(\mathcal{R}, \mathcal{F}_{\{\lambda_j\}} \mid P_{\boldsymbol{x}}) = \lim_{d \uparrow \infty} \text{regret}_{\text{pure}}(\mathcal{R}^{(d)}, \mathcal{F}_{\{\lambda_j\}}^{(d)} \mid P_{\boldsymbol{x}}) \tag{139}$$

*and*

$$\text{regret}_{\text{mix}}(\mathcal{R}, \mathcal{F}_{\{\lambda_j\}} \mid P_{\boldsymbol{x}}) = \lim_{d \uparrow \infty} \text{regret}_{\text{mix}}(\mathcal{R}^{(d)}, \mathcal{F}_{\{\lambda_j\}}^{(d)} \mid P_{\boldsymbol{x}}). \tag{140}$$

*Proof.* Let $\{c_{ij}\}_{j \in \mathbb{N}}$ be the coefficients of the orthogonal expansion of $\psi_i$, $i \in [r]$, that is, $\psi_i = \sum_{j=1}^\infty c_{ij} \phi_j$. With a slight abuse of notation, we also let $c_i := (c_{i1}, c_{i2} \dots) \in \ell_2(\mathbb{N})$. We use a sandwich argument. On one hand,

$$\text{regret}_{\text{pure}}(\mathcal{R}, \mathcal{F}_{\{\lambda_j\}} \mid P_{\boldsymbol{x}}) = \min_{R \in \mathcal{R}} \max_{f \in \mathcal{F}_{\{\lambda_j\}}} \text{regret}(R, f) \tag{141}$$

$$\geq \min_{R \in \mathcal{R}} \max_{f \in \mathcal{F}_{\{\lambda_j\}}^{(d)}} \mathsf{regret}(R, f) \tag{142}$$

$$\overset{(*)}{=} \min_{R \in \mathcal{R}^{(d)}} \max_{f \in \mathcal{F}_{\{\lambda_j\}}^{(d)}} \mathsf{regret}(R, f) \tag{143}$$

$$= \mathsf{regret}_{\mathsf{pure}}(\mathcal{R}^{(d)}, \mathcal{F}_{\{\lambda_j\}}^{(d)} \mid P_{\boldsymbol{x}}), \tag{144}$$

where $(*)$ follows from the following reasoning: For any $(R \in \mathcal{R}, f \in \mathcal{F}_{\{\lambda_j\}}^{(d)})$,

$$\mathsf{regret}(R, f) = \min_{q \in \mathbb{R}^r} \mathbb{E}\left[\left(\sum_{j=1}^{d} f_j \phi_j(\boldsymbol{x}) + \boldsymbol{n} - \sum_{j=1}^{\infty} \sum_{i=1}^{r} q_i c_{ij} \phi_j(\boldsymbol{x})\right)^2\right] - \mathbb{E}\left[\boldsymbol{n}^2\right] \tag{145}$$

$$\overset{(a)}{=} \min_{q \in \mathbb{R}^r} \mathbb{E}\left[\left(\sum_{j=1}^{d} f_j \phi_j(\boldsymbol{x}) - \sum_{j=1}^{\infty} \sum_{i=1}^{r} q_i c_{ij} \phi_j(\boldsymbol{x})\right)\right] \tag{146}$$

$$\overset{(b)}{=} \min_{q \in \mathbb{R}^r} \sum_{j=1}^{d}\left(f_j - \sum_{i=1}^{r} q_i c_{ij}\right)^2 + \sum_{j=d+1}^{\infty}\left(\sum_{i=1}^{r} q_i c_{ij}\right)^2, \tag{147}$$

where here $(a)$ follows since the noise $\boldsymbol{n}$ is independent of $\boldsymbol{x}$, and since, similarly to the finite-dimensional case (Section 3), the prediction loss based on the features $x \in \mathcal{X}$ is $\mathbb{E}[\boldsymbol{n}^2]$, for any given $f \in \mathcal{F}$, $(b)$ follows from Parseval's identity and the orthonormality of $\{\phi_j\}_{j \in \mathbb{N}}$. So,

$$\min_{R \in \mathcal{R}} \max_{f \in \mathcal{F}_{\{\lambda_j\}}^{(d)}} \mathsf{regret}(R, f)$$

$$= \min_{\{c_{ij}\}_{i \in [r], j \in \mathbb{N}}} \max_{f \in \mathcal{F}_{\{\lambda_j\}}^{(d)}} \min_{q \in \mathbb{R}^r} \sum_{j=1}^{d}\left(f_j - \sum_{i=1}^{r} q_i c_{ij}\right)^2 + \sum_{j=d+1}^{\infty}\left(\sum_{i=1}^{r} q_i c_{ij}\right)^2. \tag{148}$$

Evidently, since $\sum_{j=d+1}^{\infty}(\sum_{i=1}^{r} q_i c_{ij})^2 \geq 0$, an optimal representation may satisfy that $c_{ij} = 0$ for all $j \geq d + 1$. Thus, the optimal representation belongs to $\mathcal{R}^{(d)}$.

On the other hand,

$$\mathsf{regret}_{\mathsf{pure}}(\mathcal{R}, \mathcal{F}_{\{\lambda_j\}} \mid P_{\boldsymbol{x}}) = \min_{R \in \mathcal{R}} \max_{f \in \mathcal{F}_{\{\lambda_j\}}} \mathsf{regret}(R, f) \tag{149}$$

$$\leq \min_{R \in \mathcal{R}^{(d)}} \max_{f \in \mathcal{F}_{\{\lambda_j\}}} \mathsf{regret}(R, f) \tag{150}$$

$$\overset{(*)}{\leq} \min_{R \in \mathcal{R}^{(d)}} \max_{f \in \mathcal{F}_{\{\lambda_j\}}^{(d)}} \mathsf{regret}(R, f) + \lambda_{d+1} \tag{151}$$

$$= \mathsf{regret}_{\mathsf{pure}}(\mathcal{R}^{(d)}, \mathcal{F}_{\{\lambda_j\}}^{(d)} \mid P_{\boldsymbol{x}}) + \lambda_{d+1}, \tag{152}$$

where $(*)$ follows from the following reasoning: For any $(R \in \mathcal{R}^{(d)}, f \in \mathcal{F}_{\{\lambda_j\}})$,

$$\mathsf{regret}(R, f) = \min_{q \in \mathbb{R}^r} \mathbb{E}\left[\left(\sum_{j=1}^{\infty} f_j \phi_j(\boldsymbol{x}) + \boldsymbol{n} - \sum_{j=1}^{\infty} \sum_{i=1}^{r} q_i c_{ij} \phi_j(\boldsymbol{x})\right)^2\right] - \mathbb{E}[\boldsymbol{n}^2] \tag{153}$$

$$\overset{(a)}{=} \min_{q \in \mathbb{R}^r} \sum_{j=1}^{d}\left(f_j - \sum_{i=1}^{r} q_i c_{ij}\right)^2 + \sum_{j=d+1}^{\infty} f_j^2 \tag{154}$$

$$\overset{(b)}{\leq} \min_{q \in \mathbb{R}^r} \sum_{j=1}^{d}\left(f_j - \sum_{i=1}^{r} q_i c_{ij}\right)^2 + \lambda_{d+1}, \tag{155}$$

where $(a)$ follows similarly to the analysis made in the previous step, and $(b)$ follows since for any $f \in \mathcal{F}_{\{\lambda_j\}}$ it holds that

$$\sum_{j=d+1}^{\infty} f_j^2 \leq \lambda_{d+1} \sum_{j=d+1}^{\infty} \frac{f_j^2}{\lambda_j} \leq \lambda_{d+1} \sum_{j=1}^{\infty} \frac{f_j^2}{\lambda_j} \leq \lambda_{d+1}. \tag{156}$$

Combining (144) and (152) and using $\lambda_{d+1} \downarrow 0$ completes the proof for the pure minimax regret. The proof for the mixed minimax is analogous and thus is omitted. $\qquad\square$

We also use the following simple and technical lemma.

**Lemma 22.** *For $R \in \mathcal{R}^{(d)}$ and $f \in \mathcal{F}^{(d)}$[1]*

$$\mathsf{regret}(R, f) = f^\top \left( I_d - R^\top (RR^\top)^{-1} R \right) f, \tag{157}$$

*where $R \in \mathbb{R}^{r \times d}$ is the matrix of coefficients of the orthogonal expansion of $\psi_i = \sum_{j=1}^{d} c_{ij}\phi_j$ for $i \in [r]$, so that $R(i, j) = c_{ij}$.*

*Proof.* It holds that

$$\mathsf{regret}(R, f) = \min_{q \in \mathbb{R}^r} \mathbb{E}\left[ \left( \sum_{j=1}^{d} f_j \phi_j(\boldsymbol{x}) + \boldsymbol{n} - \sum_{i=1}^{r} q_i \sum_{j=1}^{d} c_{ij}\phi_j(\boldsymbol{x}) \right)^2 \right] - \mathbb{E}\left[\boldsymbol{n}^2\right] \tag{158}$$

$$= \min_{q \in \mathbb{R}^r} \mathbb{E}\left[ \sum_{j=1}^{d} \left( f_j - \sum_{i=1}^{r} q_i c_{ij} \right) \phi_j(\boldsymbol{x}) \right] \tag{159}$$

$$= \min_{q \in \mathbb{R}^r} \sum_{j=1}^{d} \left( f_j - \sum_{i=1}^{r} q_i c_{ij} \right)^2 \tag{160}$$

$$= \min_{q \in \mathbb{R}^r} \sum_{j=1}^{d} \left[ f_j^2 - 2 f_j \sum_{i=1}^{r} q_i c_{ij} + \sum_{i_1=1}^{r} \sum_{i_2=1}^{r} q_{i_1} c_{i_1 j} q_{i_2} c_{i_2 j} \right] \tag{161}$$

$$= \min_{q \in \mathbb{R}^r} f^\top f - 2 q^\top R f + q^\top R R^\top q \tag{162}$$

$$= f^\top \left( I_d - R^\top (RR^\top)^{-1} R \right) f, \tag{163}$$

where the last equality is obtained by the minimizer $q^* = (RR^\top)^{-1} R f$. $\qquad\square$

*Proof of Theorems 19 and 20.* By Proposition 21, we may first consider the finite dimensional case, and then take the limit $d \uparrow \infty$. By Lemma 22, in the $d$-dimensional case (for both the representation and the response function), the regret is formally as in the linear setting under the MSE of Theorem 2, by setting therein $\Sigma_{\boldsymbol{x}} = I_d$, and $S = \mathrm{diag}(\lambda_1, \ldots, \lambda_d)$ (c.f. Lemma 16). The claim of the Theorem 19 then follows by taking $d \uparrow \infty$ and noting that $\lambda_{d+1} \downarrow 0$. The proof of Theorem 20 is analogous and thus is omitted. $\qquad\square$

## G  ITERATIVE ALGORITHMS FOR THE PHASE 1 AND PHASE 2 PROBLEMS

In this section, we describe our proposed algorithms for solving the Phase 1 and Phase 2 problems of Algorithm 1. Those algorithms are general, and only require providing gradients of the regret function (1) and an initial representation and a set of adversarial functions. These are individually determined for each setting. See Appendix H for the way these are determined in Examples 6 and 8.

---

[1]Note that any $f \in \mathcal{F}^{(d)}$ may be uniquely identified with a $d$-dimensional vector $f \in \mathbb{R}^d$. With a slight abuse of notation we do not distinguish between the two.

### G.1 PHASE 1: FINDING A NEW ADVERSARIAL FUNCTION

We propose an algorithm to solve the Phase 1 problem (23), which is based on an iterative algorithm. We denote the function's value at the $t$th iteration by $f_{(t)}$. The proposed Algorithm 2 operates as follows. At initialization, the function $f_{(1)} \in \mathcal{F}$ is arbitrarily initialized (say at random), and then the optimal predictor $Q^{(j)}$ is found for each of the $k$ possible representations $R^{(j)}$, $j \in [k]$. Then, the algorithm iteratively repeats the following steps, starting with $t = 2$: (1) Updating the function from $f_{(t-1)}$ to $f_{(t)}$ based on a gradient step of

$$\sum_{j \in [k]} p^{(j)} \cdot \mathbb{E}\left[\mathsf{loss}(f_{(t-1)}(\boldsymbol{x}), Q^{(j)}(R^{(j)}(\boldsymbol{x})))\right], \tag{164}$$

that is, the weighted loss function of the previous iteration function, which is then followed by a projection to the feasible class of functions $\mathcal{F}$, denoted as $\Pi_{\mathcal{F}}(\cdot)$ (2) Finding the optimal predictor $Q^{(j)}$ for the current function $f_{(t)}$ and the given representations $\{R^{(j)}\}_{j \in [k]}$, and computing the respective loss for each representation,

$$L^{(j)} := \mathbb{E}\left[\mathsf{loss}(f_{(t)}(\boldsymbol{x}), Q^{(j)}(R^{(j)}(\boldsymbol{x})))\right]. \tag{165}$$

This loop iterates for $T_f$ iterations, or until convergence.

---

**Algorithm 2** A procedure for finding a new function via the solution of (23)

---

1: **procedure** PHASE 1 SOLVER($\{R^{(j)}, p^{(j)}\}_{j \in [k]}, \mathcal{F}, \mathcal{Q}, d, r, P_{\boldsymbol{x}}$)
2:     **begin**
3:     **initialize** $T_f$                                                 ▷ Number of iterations parameters
4:     **initialize** $\eta_f$                                                          ▷ Step size parameter
5:     **initialize** $f_{(1)} \in \mathcal{F}$                              ▷ Function initialization, e.g., at random
6:     **for** $j = 1$ to $k$ **do**
7:         **set** $Q^{(j)} \leftarrow \operatorname{argmin}_{Q \in \mathcal{Q}} \mathbb{E}\left[\mathsf{loss}(f_{(1)}(\boldsymbol{x}), Q(R^{(j)}(\boldsymbol{x})))\right]$
8:     **end for**
9:     **for** $t = 2$ to $T_f$ **do**
10:         **update** $f_{(t-1/2)} = f_{(t-1)} + \eta_f \cdot \sum_{j \in [k]} p^{(j)}_{(t-1)} \cdot \nabla_f \mathbb{E}\left[\mathsf{loss}(f_{(t-1)}(\boldsymbol{x}), Q^{(j)}(R^{(j)}(\boldsymbol{x})))\right]$
    ▷ A gradient update of the function
11:         **project** $f_{(t)} = \Pi_{\mathcal{F}}(f_{(t-1/2)})$                       ▷ Projection on the class $\mathcal{F}$
12:         **for** $j = 1$ to $k$ **do**
13:             **set** $Q^{(j)} \leftarrow \operatorname{argmin}_{Q \in \mathcal{Q}} \mathbb{E}\left[\mathsf{loss}(f_{(t)}(\boldsymbol{x}), Q(R^{(j)}(\boldsymbol{x})))\right]$     ▷ Update of predictors
14:             **set** $L^{(j)} \leftarrow \mathbb{E}\left[\mathsf{loss}(f_{(t)}(\boldsymbol{x}), Q^{(j)}(R^{(j)}(\boldsymbol{x})))\right]$ ▷ Compute loss of each representation
15:         **end for**
16:     **end for**
17:     **return** $f_{(T)}$ **and the regret** $\sum_{j \in [k]} p^{(j)} \cdot L^{(j)}$
18: **end procedure**

---

**Design choices and possible variants of the basic algorithm**   At initialization, we have chosen a simple random initialization for $f_{(1)}$, but it may also be initialized based on some prior knowledge of the adversarial function. For the update of the predictors, we have specified a full computation of the optimal predictor, which can be achieved in practice by running another iterative algorithm such as stochastic gradient descent (SGD) until convergence. If this is too computationally expensive, the number of gradient steps may be limited. The update of the function is done via projected SGD with a constant step size $\eta_f$, yet it is also possible to modify the step size with the iteration, e.g., the common choice $\eta_f / \sqrt{t}$ at step $t$ (Hazan, 2016). Accelerated algorithms, e.g., moment-based, may also be deployed.

**Convergence analysis**   A theoretical analysis of the convergence properties of the algorithm appears to be challenging. Evidently, this is a minimax game between the response player and a player cooperating with the representation player, which optimizes the prediction rule in order to minimize the loss. This is, however, not a concave-convex game. As described in Appendix A, even concave-convex games are not well understood at this point. We thus opt to validate this algorithm numerically.

**Running-time complexity analysis** Algorithm 2 runs for a fixed number of iterations $T_f$, accepts $k$ representations, and makes $kT_f$ updates. Each update is comprised from a gradient step of for the adversarial function (cost $C_1$), and optimization of the predictor (cost $C_2$). So the total computation complexity is $kT_f \cdot (C_1 + C_2)$. The most expensive part is the optimization of the predictor $C_2$, and this can be significantly reduced by running a few gradients steps of the predictor instead of a full optimization. If we take $g$ gradient steps then $C_2$ is replaced by $C_1 g$ and the total computational cost is $kT_f(g+1)C_1$.

## G.2 Phase 2: finding a new representation

We propose an iterative algorithm to solve the Phase 2 problem (24), and thus finding a new representation $R^{(k+1)}$. To this end, we first note that the objective function in (24) can be separated into a part that depends on existing representations and a part that depends on the new one, specifically, as

$$
\sum_{j_1 \in [k]} \sum_{j_2 \in [m_0+k]} p^{(j_1)} \cdot o^{(j_2)} \cdot \mathbb{E}\left[\mathsf{loss}(f^{(j_2)}(\boldsymbol{x}), Q^{(j_1,j_2)}(R^{(j_1)}(\boldsymbol{x})))\right]
$$
$$
+ \sum_{j_2 \in [m_0+k]} p^{(k+1)} \cdot o^{(j_2)} \cdot \mathbb{E}\left[\mathsf{loss}(f^{(j_2)}(\boldsymbol{x}), Q^{(k+1,j_2)}(R^{(k+1)}(\boldsymbol{x})))\right]
$$
$$
= \sum_{j_1 \in [k]} \sum_{j_2 \in [m_0+k]} p^{(j_1)} \cdot o^{(j_2)} \cdot L^{(j_1,j_2)}
$$
$$
+ \sum_{j_2 \in [m_0+k]} p^{(k+1)} \cdot o^{(j_2)} \cdot \mathbb{E}\left[\mathsf{loss}(f^{(j_2)}(\boldsymbol{x}), Q^{(k+1,j_2)}(R^{(k+1)}(\boldsymbol{x})))\right], \tag{166}
$$

where

$$
L^{(j_1,j_2)} := \mathbb{E}\left[\mathsf{loss}(f^{(j_2)}(\boldsymbol{x}), Q^{(j_1,j_2)}(R^{(j_2)}(\boldsymbol{x})))\right], \tag{167}
$$

and the predictors $\{Q^{(j_1,j_2)}\}_{j_1 \in [k], j_2 \in [m_0+k]}$ can be optimized independently of the new representation $R^{(k+1)}$. We propose an iterative algorithm for this problem, and denote the new representation at the $t$th iteration of the algorithm by $R_{(t)}^{(k+1)}$. The algorithm's input is a set of $m_0 + k$ adversarial functions $\{f^{(i)}\}_{i \in [m_0+k]}$, and the current set of representations $\{R^{(j)}\}_{j \in [k]}$. Based on these, the algorithm may find the optimal predictor for $f^{(j_2)}$ based on the representation $R^{(j_1)}$, and thus compute the loss

$$
L_*^{(j_1,j_2)} := \min_{Q \in \mathcal{Q}} \mathbb{E}\left[\mathsf{loss}(f^{(j_2)}(\boldsymbol{x}), Q(R^{(j_1)}(\boldsymbol{x})))\right] \tag{168}
$$

for $j_1 \in [k]$ and $j_2 \in [m_0 + k]$. In addition, the new representation is arbitrarily initialized (say, at random) as $R_{(1)}^{(k+1)}$, and the predictors $\{Q_{(1)}^{(k+1,j_2)}\}_{j_2 \in [m_0+k]}$ are initialized as the optimal predictors for $f^{(j_2)}$ given the representation $R_{(1)}^{(k+1)}$. The algorithm keeps track of weights for the representations (including the new one), which are initialized uniformly, i.e., $p_{(1)}^{(j_1)} = \frac{1}{k+1}$ for $j_1 \in [k+1]$ (including a weight for the new representation). The algorithm also keeps track of weights for the functions, which are also initialized uniformly as $o_{(1)}^{(j_2)} = \frac{1}{m_0+k}$ for $j_2 \in [m_0 + k]$. Then, the algorithm iteratively repeats the following steps, starting with $t = 2$: (1) Updating the new representation from $R_{(t-1)}^{(k+1)}$ to $R_{(t)}^{(k+1)}$ based on a gradient step of the objective function (24) as a function of $R^{(k+1)}$. Based on the decomposition in (166) the term of the objective which depends on $R^{(k+1)}$ is

$$
p_{(t-1)}^{(k+1)} \sum_{j_2 \in [m_0+k]} o_{(t-1)}^{(j_2)} \cdot \mathbb{E}\left[\mathsf{loss}(f^{(j_2)}(\boldsymbol{x}), Q^{(k+1,j_2)}(R^{(k+1)}(\boldsymbol{x})))\right], \tag{169}
$$

that is, the loss function of the previous iteration new representation, weighted according to the current function weights $o_{(t-1)}^{(j_2)}$. Since the multiplicative factor $p_{(t-1)}^{(k+1)}$ is common to all terms, it is removed from the gradient computation (this aids in the choice of the gradient step). This gradient step is then possibly followed by normalization or projection, which we denote by the operator $\Pi_{\mathcal{R}}(\cdot)$. For example, in the linear case, it make sense to normalize $R^{(k+1)}$ to have unity norm (in

some matrix norm of choice). After updating the new representation to $R_{(t)}^{(k+1)}$, optimal predictors are found for each function, the loss is computed

$$L_{(t)}^{(k+1,j_2)} := \min_{Q \in \mathcal{Q}} \mathbb{E}\left[\mathsf{loss}(f^{(j_2)}(\boldsymbol{x}), Q(R_{(t)}^{(k+1)}(\boldsymbol{x})))\right] \tag{170}$$

for all $j_2 \in [m_0 + k]$, and the optimal predictor is updated to $\{Q_{(t)}^{(k+1,j_2)}\}_{j_2 \in [m_0+k]}$ based on this solution. (2) Given the current new representation $R_{(t)}^{(k+1)}$, the loss matrix

$$\{L_{(t)}^{(j_1,j_2)}\}_{j_1 \in [k], j_2 \in [m_0+k]} \tag{171}$$

is constructed where for $j_1 \in [k]$ it holds that $L_{(t)}^{(j_1,j_2)} = L^{(j_1,j_2)}$ for all $t$ (i.e., the loss of previous representations and functions is kept fixed). This is considered to be the loss matrix of a two-player zero-sum game between the representation player and the function player, where the representation player has $k + 1$ possible strategies and the function player has $m_0 + k$ strategies. The weights $\{p_{(t)}^{(j_1)}\}_{j_1 \in [k+1]}$ and $\{o_{(t)}^{(j_2)}\}_{j_2 \in [m_0+k]}$ are then updated according to the MWU rule. Specifically, for an *inverse temperature parameter* $\beta$ (or a *regularization parameter*), the update is given by

$$p_{(t)}^{(j)} = \frac{p_{(t-1)}^{(j)} \cdot \beta^{L^{(j)}}}{\sum_{\tilde{j} \in [k]} p_{(t-1)}^{(\tilde{j})} \cdot \beta^{L^{(\tilde{j})}}} \tag{172}$$

for the representation weights and, analogously, by

$$o_{(t)}^{(j)} = \frac{o_{(t-1)}^{(j)} \cdot \beta^{-L^{(j)}}}{\sum_{\tilde{j} \in [k]} o_{(t-1)}^{(\tilde{j})} \cdot \beta^{-L^{(\tilde{j})}}} \tag{173}$$

for the function weights (as the function player aims to maximize the loss). This can be considered as a regularized gradient step on the probability simplex, or more accurately, a *follow-the-regularized-leader* (Hazan, 2016). The main reasoning of this algorithm is that at each iteration the weights $\{p^{(j)}\}_{j \in [k+1]}$ and $\{o^{(j)}\}_{j \in [m_0+k]}$ are updated towards the solution of the two-player zero-sum game with payoff matrix $\{-L_{(t)}^{(j_1,j_2)}\}_{j_1 \in [k+1], j_2 \in [m_0+k]}$. In turn, based only on the function weights $\{o^{(j)}\}_{j \in [m_0+k]}$, the new representation is updated to $R_{(t)}^{(k+1)}$, which then changes the pay-off matrix at the next iteration. It is well known that the MWU solved two-player zero-sum game (Freund and Schapire, 1999), in which the representation player can choose the weights and the function player can choose the function.

This loop iterates for $T_{\mathrm{stop}}$ iterations, and then the optimal weights are given by the average over the last $T_{\mathrm{avg}}$ iterations (Freund and Schapire, 1999), i.e.,

$$p_*^{(j)} = \frac{1}{T_{\mathrm{avg}}} \sum_{t=T_{\mathrm{stop}}-T_{\mathrm{avg}}+1}^{T_{\mathrm{stop}}} p_{(t)}^{(j)}, \tag{174}$$

and

$$o_*^{(j)} = \frac{1}{T_{\mathrm{avg}}} \sum_{t=T_{\mathrm{stop}}-T_{\mathrm{avg}}+1}^{T_{\mathrm{stop}}} o_{(t)}^{(j)}. \tag{175}$$

In the last $T_R - T_{\mathrm{stop}}$ iterations, only the representation $R_{(t)}^{(k+1)}$ and the predictors are updated. The algorithm then outputs $R_{(T)}^{(k+1)}$ as the new representation and the weights $\{p_*^{(j)}\}_{j \in [k+1]}$.

**Design choices and possible variants of the basic algorithm**   At initialization, we have chosen a simple random initialization for $R_{(1)}^{(k+1)}$, but it may also be initialized based on some prior knowledge of the desired new representation. The initial predictors $\{Q_{(1)}^{(k+1,j_2)}\}_{j_2 \in [m_0+k]}$ will then be initialized as the optimal predictors for $R_{(1)}^{(k+1)}$ and $\{f^{(j_2)}\}_{j_2 \in [m_0+k]}$. We have initialized the

---

**Algorithm 3** A procedure for finding a new representation $R^{(k+1)}$ via the solution of (24)

---

1: **procedure** PHASE 2 SOLVER($\{R^{(j_1)}\}_{j\in[k]}, \{f^{(j_2)}\}_{j_2\in[m_0+k]}, \mathcal{R}, \mathcal{F}, \mathcal{Q}, d, r, P_{\boldsymbol{x}}$)
2:     **begin**
3:       **initialize** $T_R, T_{\text{stop}}, T_{\text{avg}}$                                      ▷ Number of iterations parameters
4:       **initialize** $\eta_R$                                            ▷ Step size parameter
5:       **initialize** $\beta \in (0,1)$                               ▷ Inverse temperature parameter
6:       **initialize** $f_{(1)} \in \mathcal{F}$                        ▷ Function initialization, e.g., at random
7:       **initialize** $p_{(1)}^{(j)} \leftarrow 0$ for $j \in [k]$ and $p_{(1)}^{(k+1)} \leftarrow 0$     ▷ A uniform weight initialization for the representations
8:       **initialize** $o_{(1)}^{(j_2)} \leftarrow \frac{1}{m_0+k}$ for $j_2 \in [k]$     ▷ A uniform weight initialization for the functions
9:       **for** $j_1 = 1$ to $k$ **do**
10:          **for** $j_2 = 1$ to $m_0 + k$ **do**
11:              **set** $Q^{(j_1,j_2)} \leftarrow \operatorname{argmin}_{Q\in\mathcal{Q}} \mathbb{E}\left[\text{loss}(f^{(j_2)}(\boldsymbol{x}), Q(R^{(j_1)}(\boldsymbol{x})))\right]$
    ▷ Optimal predictors for existing representations and input functions
12:              **set** $L^{(j_1,j_2)} \leftarrow \min_{Q\in\mathcal{Q}} \mathbb{E}\left[\text{loss}(f^{(j_2)}(\boldsymbol{x}), Q^{(j_1,j_2)}(R^{(j_1)}(\boldsymbol{x})))\right]$   ▷ The minimal loss
13:          **end for**
14:       **end for**
15:       **for** $j_2 = 1$ to $m_0 + k$ **do**
16:          **initialize** $R_{(1)}^{(k+1)}$                                   ▷ Arbitrarily, e.g., at random
17:          **set** $Q_{(1)}^{(k+1,j_2)} \leftarrow \operatorname{argmin}_{Q\in\mathcal{Q}} \mathbb{E}\left[\text{loss}(f^{(j_2)}(\boldsymbol{x}), Q(R^{(k+1)}(\boldsymbol{x})))\right]$ for $j_2 \in [m_0+k]$
    ▷ Optimal predictors for new representation and input functions
18:       **end for**
19:       **for** $t = 2$ to $T_R$ **do**
20:          **update**                                 ▷ A gradient update of the new representation

$$R_{(t-1/2)}^{(k+1)} = R_{(t-1)}^{(k+1)} + \eta_R \cdot \sum_{j_2\in[m_0+k]} o_{(t-1)}^{(j_2)} \cdot \nabla_{R^{(k+1)}} \mathbb{E}\left[\text{loss}(f^{(j_2)}(\boldsymbol{x}), Q^{(k+1,j_2)}(R_{(t-1)}^{(k+1)}(\boldsymbol{x})))\right]$$

(176)

21:          **project** $R_{(t)}^{(k+1)} = \Pi_{\mathcal{R}}(R_{(t-1/2)}^{(k+1)})$                ▷ Standardization based on the class $\mathcal{R}$
22:          **for** $j = 1$ to $k$ **do**
23:              **set** $Q^{(k+1,j_2)} \leftarrow \operatorname{argmin}_{Q\in\mathcal{Q}} \mathbb{E}\left[\text{loss}(f^{(j_2)}(\boldsymbol{x}), Q(R_{(t)}^{(k+1)}(\boldsymbol{x})))\right]$
    ▷ Update of predictors for the new representation
24:              $L_{(t)}^{(k+1,j_2)} \leftarrow \mathbb{E}\left[\text{loss}((f^{(j_2)}(\boldsymbol{x}), Q^{(k+1,j_2)}(R_{(t)}^{(k+1)}(\boldsymbol{x})))\right]$         ▷ Compute loss
25:          **end for**
26:          **set** $L_{(t)}^{(j_1,j_2)} \leftarrow L^{(j_1,j_2)}$ for $j_1 \in [k]$ and $j_2 \in [m_0+k]$
27:          **if** $t < T_{\text{stop}}$ **then**
28:              **update** $p_{(t)}^{(j)} \leftarrow \frac{p_{(t-1)}^{(j)} \cdot \beta^{L^{(j)}}}{\sum_{\tilde{j}\in[k]} p_{(t-1)}^{(\tilde{j})} \cdot \beta^{L^{(\tilde{j})}}}$ for $j \in [k]$               ▷ A MWU
29:              **update** $o_{(t)}^{(j)} \leftarrow \frac{o_{(t-1)}^{(j)} \cdot \beta^{-L^{(j)}}}{\sum_{\tilde{j}\in[m_0+k]} o_{(t-1)}^{(\tilde{j})} \cdot \beta^{-L^{(\tilde{j})}}}$ for $j \in [m_0+k]$       ▷ A MWU
30:          **else if** $t = T_{\text{stop}}$ **then**
31:              **update** $p_{(t)}^{(j)} = p_{(t)}^{(j)} \leftarrow \frac{1}{T_{\text{avg}}} \sum_{t=T_{\text{stop}}-T_{\text{avg}}+1}^{T_{\text{stop}}} p_{(t)}^{(j)}$ for $j \in [k]$
    ▷ Optimal weights by averaging last $T_{\text{avg}}$ iterations
32:              **update** $o_{(t)}^{(j)} \leftarrow \frac{1}{T_{\text{avg}}} \sum_{t=T_{\text{stop}}-T_{\text{avg}}+1}^{T_{\text{stop}}} o_{(t)}^{(j)}$ for $j \in [m_0+k]$
    ▷ Optimal weights by averaging last $T_{\text{avg}}$ iterations
33:          **else**
34:              **update** $p_{(t)}^{(j)} \leftarrow p_{(t-1)}^{(j)}$ for $j \in [k]$          ▷ No update for the last $T - T_{\text{stop}}$ iterations
35:              **update** $o_{(t)}^{(j)} \leftarrow o_{(t-1)}^{(j)}$ for $j \in [m_0+k]$ ▷ No update for the last $T - T_{\text{stop}}$ iterations
36:          **end if**
37:          **return** $R_{(T)}^{(k+1)}$ **and** $\{p_{(T_R)}^{(j)}\}_{j\in[k+1]}$
38:       **end for**
39: **end procedure**

---

Table 1: Hardware details

| CPU | RAM | GPU |
|---|---|---|
| Intel i9 13900k | 64GB | RTX 3090 Ti |

representation and function weights uniformly. A possibly improved initialization for the function weights is to put more mass on the more recent functions, that is, for large values of $j_2$, or to use the minimax strategy of the function player in the two-player zero-sum game with payoff matrix $\{-L_{(t)}^{(j_1,j_2)}\}_{j_1 \in [k], j_2 \in [m_0+k]}$ (that is, a game which does not include the new representation). As in the Phase 1 algorithm, the gradient update of the new representation can be replaced by a more sophisticated algorithm, the computation of the optimal predictors can be replaced with (multiple) update steps, and the step size may also be adjusted. For the MWU update, we use the proposed scaling proposed by Freund and Schapire (1999)

$$\beta = \frac{1}{1 + \sqrt{\frac{c \ln m}{T}}} \tag{177}$$

for some constant $c$. It is well known that using the last iteration of a MWU algorithm may fail (Bailey and Piliouras, 2018), while averaging the weights value of all iterations provides the optimal value of a two-player zero-sum games (Freund and Schapire, 1999). For improved accuracy, we compute the average weights over the last $T_{\text{avg}}$ iterations (thus disregarding the initial iterations). We then halt the weights update and let the function and predictor update to run for $T - T_{\text{stop}}$ iterations in order to improve the convergence of $R^{(k+1)}$. Finally, the scheduling of the steps may be more complex, e.g., it is possible that running multiple gradient steps follows by multiple MWU steps may improve the result.

**Running-time complexity analysis** Algorithm 3 is more complicated than Algorithm 2, but the computational complexity analysis is similar. It runs for $T_R$ iterations and the total cost is roughly on the order of $T_R k^2 g C_1$ (taking $g$ gradient steps for the predictor optimization; $k^2$ is the number of representations, and is controlled by the learner; $C_1$ is determined by the computer, and $g$ should be large enough to assure quality results).

## H DETAILS FOR THE EXAMPLES OF ALGORITHM 1 AND ADDITIONAL EXPERIMENTS

As mentioned, the solvers of the Phase 1 and Phase 2 problems of Algorithm 1 require the gradients of the regret (1) as inputs, as well as initial representation and set of adversarial functions. We next provide these details for the examples in Section 4. The code for the experiments was written in `Python 3.6` and is available at this link. The optimization of hyperparameters was done using the `Optuna` library. The hardware used is standard and detailed appear in Table 1.

### H.1 DETAILS FOR EXAMPLE 6: THE LINEAR MSE SETTING

In this setting, the expectation over the feature distribution can be carried out analytically, and the regret is given by

$$\text{regret}(R, f \mid \Sigma_{\boldsymbol{x}}) = \mathbb{E}\left[\left(f^\top \boldsymbol{x} - q^\top R^\top \boldsymbol{x}\right)^2\right] \tag{178}$$

$$= f^\top \Sigma_{\boldsymbol{x}} f - 2q^\top R^\top \Sigma_{\boldsymbol{x}} f + q^\top R^\top R q. \tag{179}$$

The regret only depends on the feature distribution $P_{\boldsymbol{x}}$ via $\Sigma_{\boldsymbol{x}}$. For each run of the algorithm, the covariance matrix $\Sigma_{\boldsymbol{x}}$ was chosen to be diagonal with elements drawn from a log-normal distribution, with parameters $(0, \sigma_0)$, and $S = I_d$.

**Regret gradients** The gradient of the regret w.r.t. the function $f$ is given by

$$\nabla_f \mathbb{E}\left[\left(f^\top \boldsymbol{x} - q^\top R^\top \boldsymbol{x}\right)^2\right] = 2f^\top \Sigma_{\boldsymbol{x}} - 2q^\top R^\top \Sigma_{\boldsymbol{x}} \tag{180}$$

Table 2: Parameters for linear MSE setting example

| Parameter | $\beta_r$ | $\beta_f$ | $\eta_r$ | $\eta_f$ |
|-----------|-----------|-----------|----------|----------|
| Value | 0.94 | 0.653 | 0.713 | 0.944 |

| Parameter | $T_R$ | $T_f$ | $T_{\text{avg}}$ | $T_{\text{stop}}$ |
|-----------|-------|-------|------------------|-------------------|
| Value | 100 | until convergence | 10 | 80 |

and the projection on $\mathcal{F}_S$ is

$$\Pi_{\mathcal{F}}(f) = \begin{cases} \frac{f}{\|f\|_S}, & \|f\|_S \geq 1 \\ f, & \|f\|_S < 1 \end{cases}. \tag{181}$$

However, we may choose to normalize by $\frac{f}{\|f\|_S}$ even if $\|f\|_S \leq 1$ since in this case the regret is always larger if $f$ is replaced by $\frac{f}{\|f\|_S}$ (in other words, the worst case function is obtained on the boundary of $\mathcal{F}_S$). The gradient w.r.t. the predictor $q$ is given by

$$\nabla_q \mathbb{E}\left[\left(f^\top \boldsymbol{x} - q^\top R^\top \boldsymbol{x}\right)^2\right] = \left[-2f^\top \Sigma_{\boldsymbol{x}} R + 2q^\top R^\top \Sigma_{\boldsymbol{x}} R\right]. \tag{182}$$

Finally, to derive the gradient w.r.t. $R$, let us denote $R := [R_1, R_2, \ldots, R_r] \in \mathbb{R}^{d \times r}$ where $R_i \in \mathbb{R}^d$ is the $i$th column ($i \in [r]$), and $q^\top = (q_1, q_2, \ldots, q_r)$. Then, $q^\top R^\top \boldsymbol{x} = \sum_{i \in [d]} q_i R_i^\top \boldsymbol{x}$ and the loss function is

$$\mathbb{E}\left[\left(f^\top \boldsymbol{x} - q^\top R^\top \boldsymbol{x}\right)^2\right] = \mathbb{E}\left[\left(f^\top \boldsymbol{x} - \sum_{i \in [d]} q_i \boldsymbol{x}^\top R_i\right)^2\right] \tag{183}$$

$$= f^\top \Sigma_{\boldsymbol{x}} f - 2q^\top R^\top \Sigma_{\boldsymbol{x}} f + q^\top R^\top \Sigma_{\boldsymbol{x}} R q. \tag{184}$$

The gradient of the regret w.r.t. $R_k$ is then given by

$$\nabla_{R_k}\left\{\mathbb{E}\left[\left(f^\top \boldsymbol{x} - q^\top R^\top \boldsymbol{x}\right)^2\right]\right\} = -2\mathbb{E}\left[\left(f^\top \boldsymbol{x} - q^\top R^\top \boldsymbol{x}\right) \cdot q_k \boldsymbol{x}^\top\right] \tag{185}$$

$$= -2q_k\left(f^\top \Sigma_{\boldsymbol{x}} - q^\top R^\top \Sigma_{\boldsymbol{x}}\right), \tag{186}$$

hence, more succinctly, the gradient w.r.t. $R$ is

$$\nabla_R\left\{\mathbb{E}\left[\left(f^\top \boldsymbol{x} - q^\top R^\top \boldsymbol{x}\right)^2\right]\right\} = -2q\left(f^\top \Sigma_{\boldsymbol{x}} - q^\top R^\top \Sigma_{\boldsymbol{x}}\right). \tag{187}$$

We remark that in the algorithm these gradients are multiplied by weights. We omit this term whenever the weight is common to all terms in order to keep the effective step size constant.

**Initialization** Algorithm 1 requires an initial representation $R^{(1)}$ and an initial set of functions $\{f^{(j)}\}_{j \in [m_0]}$. In the MSE setting, each function $f \in \mathbb{R}^d$ is also a single column of a representation matrix $R \in \mathbb{R}^{d \times r}$. A plausible initialization matrix $R^{(1)} \in \mathbb{R}^{d \times r}$ is therefore the worst $r$ functions. These, in turn, can be found by running Algorithm (1) to obtain $\tilde{m} = r$ functions, by setting $\tilde{r} = 1$. A proper initialization for this run is simply an all-zero representation $\tilde{R}^{(1)} = 0 \in \mathbb{R}^{d \times 1}$. The resulting output is then $\{\tilde{R}^{(j)}_{(T)}\}_{j \in [r]}$ which can be placed as the $r$ columns of $R^{(1)}$. This initialization is then used for Algorithm 1.

**Algorithm parameters** The algorithm parameters used for Example 6 are shown in Table 2. The parameters were optimally tuned for $\sigma_0 = 1$.

**Additional results** The learning curve for running Algorithm 1 for Example 6 is shown in Figure 4, which shows the improvement in regret in each iteration, for which an additional matrix is added to the set of representations. It can be seen that mixing roughly 10 matrices suffice to get close to the

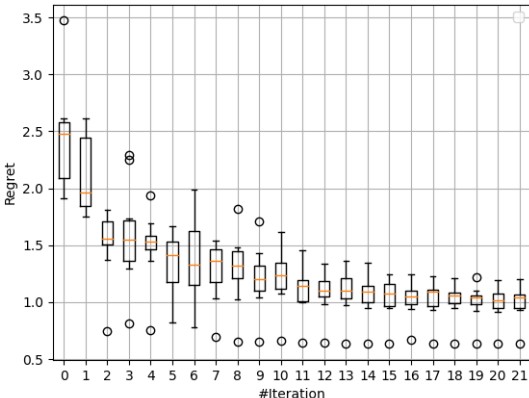

Figure 4: The learning curve for Algorithm 1 in the linear MSE setting: $d = 20$, $r = 3$, $\sigma = 1$.

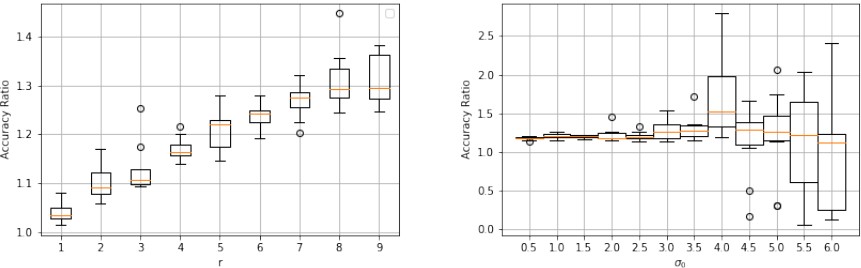

Figure 5: The ratio between the regret achieved by Algorithm 1 and the theoretical regret in the linear MSE setting. Left: $d = 20$, $\sigma_0 = 1$, varying $r$. Right: $r = 5$, $d = 20$, varying $\sigma_0$.

minimal regret attained by the algorithm, compared to the potential number of $\binom{d}{r} = \binom{20}{3} = 1140$ representation matrices determined by $\overline{\mathcal{A}}$.

Additional results of the accuracy of the Algorithm 1 in the linear MSE setting are displayed in Figure 5. The left panel of Figure 5 shows that the algorithm output is accurate for small values of $r$, but deteriorates as $r$ increases. This is because when $r$ increases then so is $\ell^*$ and so is the required number of matrices in the support of the representation rule (denoted by $m$). Since the algorithm gradually adds representation matrices to the support, an inaccurate convergence at an early iteration significantly affects later iterations. One possible way to remedy this is to run each iteration multiple times, and choose the best one, before moving on to the next one. Another reason is that given large number of matrices in the support (large $m$), it becomes increasingly difficult for the the MWU to accurately converge. Since the iterations of the MWU do not converge to the equilibrium point, but rather their average (see discussion in Appendix A) this can only be remedied by allowing more iterations for convergence (in advance) for large values of $m$. The right panel of Figure 5 shows that the algorithm output is accurate for a wide range of the condition number of the covariance matrix. This condition number is determined by the choice of $\sigma_0$, where low values typically result covariance matrices with condition number that is close to 1, while high values will typically result large condition number. The right panel shows that while the hyperparameters were tuned for $\sigma_0 = 1$, the result is fairly accurate for a wide range of $\sigma_0$ values, up to $\sigma_0 \approx 5$. Since for $Z \sim N(0, 1)$ (standard normal) it holds that $\mathbb{P}[-2 < Z < 2] \approx 95\%$, the typical condition number of a covariance matrix drawn with $\sigma_0 = 5$ is roughly $\frac{e^{2\sigma_0}}{e^{-2\sigma_0}} \approx 4.85 \cdot 10^8$, which is a fairly large range.

## H.2 DETAILS FOR EXAMPLE 8: THE LINEAR CROSS-ENTROPY SETTING

In this setting,

$$\text{regret}(R, f \mid P_{\boldsymbol{x}}) = \min_{q \in \mathbb{R}^r} \mathbb{E}\left[D_{\text{KL}}\left([1 + \exp(-f^\top \boldsymbol{x})]^{-1} \mid\mid [1 + \exp(-q^\top R^\top \boldsymbol{x})]^{-1}\right)\right], \quad (188)$$

and the expectation over the feature distribution typically cannot be carried out analytically. We thus tested Algorithm 1 on empirical distributions of samples drawn from a high-dimensional normal distribution. Specifically, for each run, $B = 1000$ feature vectors were drawn from an isotropic normal distribution of dimension $d = 15$. The expectations of the regret and the corresponding gradients were then computed with respect to (w.r.t.) the resulting empirical distributions.

**Regret gradients**  We use the facts that

$$\frac{\partial}{\partial p_1} D_{\text{KL}}(p_1 \mid\mid p_2) = \log \frac{p_1(1 - p_2)}{p_2(1 - p_1)} \quad (189)$$

and

$$\frac{\partial}{\partial p_2} D_{\text{KL}}(p_1 \mid\mid p_2) = \frac{p_2 - p_1}{p_2(1 - p_2)}. \quad (190)$$

For brevity, let us next denote

$$p_1 := \frac{1}{1 + \exp(-f^\top \boldsymbol{x})} \quad (191)$$

and

$$p_2 := \frac{1}{1 + \exp(-q^\top R^\top \boldsymbol{x})}. \quad (192)$$

We next repeatedly use the chain rule for differentiation. First,

$$\nabla_f p_1 = \nabla_f \left[\frac{1}{1 + \exp(-f^\top \boldsymbol{x})}\right] = \frac{\exp(-f^\top \boldsymbol{x}) \cdot \boldsymbol{x}}{[1 + \exp(-f^\top \boldsymbol{x})]^2} = p_1(1 - p_1) \cdot \boldsymbol{x}. \quad (193)$$

and

$$\nabla_q p_2 = \nabla_q \left[\frac{1}{1 + \exp(-q^\top R^\top \boldsymbol{x})}\right] = \frac{\exp(-q^\top R^\top \boldsymbol{x}) \cdot R^\top \boldsymbol{x}}{[1 + \exp(-q^\top R^\top \boldsymbol{x})]^2} = p_2(1 - p_2) \cdot R^\top \boldsymbol{x}. \quad (194)$$

So, assuming that $P_{\boldsymbol{x}}$ is such that the order of differentiation and expectation may be interchanged (this can be guaranteed using dominated/monotone convergence theorems), the gradient of the regret w.r.t. $f$ is

$$\nabla_f \text{regret}(R, f \mid P_{\boldsymbol{x}}) = \mathbb{E}\left[\frac{\partial}{\partial p_1} D_{\text{KL}}(p_1 \mid\mid p_2) \times \nabla_f p_1\right] \quad (195)$$

$$= \mathbb{E}\left[\log\left(\frac{p_1(1 - p_2)}{p_2(1 - p_1)}\right) \cdot p_1(1 - p_1) \cdot \boldsymbol{x}\right] \quad (196)$$

$$= \mathbb{E}\left[(f^\top - q^\top R^\top)\boldsymbol{x} \frac{\exp(-f^\top \boldsymbol{x})}{[1 + \exp(-f^\top \boldsymbol{x})]^2} \cdot \boldsymbol{x}\right] \quad (197)$$

$$= \mathbb{E}\left[\frac{\exp(-f^\top \boldsymbol{x})}{[1 + \exp(-f^\top \boldsymbol{x})]^2} \cdot \boldsymbol{x}^\top (f - Rq)\boldsymbol{x}\right]. \quad (198)$$

Next, under similar assumptions, the gradient of the regret w.r.t. the predictor $q$ is

$$\nabla_q \text{regret}(R, f \mid P_{\boldsymbol{x}}) = \mathbb{E}\left[\frac{\partial}{\partial p_2} D_{\text{KL}}(p_1 \mid\mid p_2) \times \nabla_q p_2\right] \quad (199)$$

$$= \mathbb{E}\left[\left(\frac{1}{1 + \exp(-q^\top R^\top \boldsymbol{x})} - \frac{1}{1 + \exp(-f^\top \boldsymbol{x})}\right) \cdot R^\top \boldsymbol{x}\right]. \quad (200)$$

Table 3: Parameters for linear cross entropy setting example

| Parameter | $\beta_r$ | $\beta_f$ | $\eta_r$ | $\eta_f$ |
|-----------|-----------|-----------|----------|----------|
| Value | 0.9 | 0.9 | $10^{-3}$ | $10^{-1}$ |
| Parameter | $T_R$ | $T_f$ | $T_{\text{avg}}$ | $T_{\text{stop}}$ |
| Value | 100 | 1000 | 25 | 50 |

Finally, as for the MSE case, to derive the gradient w.r.t. $R$, we denote $R := [R_1, R_2, \ldots, R_r] \in \mathbb{R}^{d \times r}$ where $R_i \in \mathbb{R}^d$ is the $i$th column ($i \in [r]$), and $q^\top = (q_1, q_2, \ldots, q_r)$. Then, $q^\top R^\top x = \sum_{i \in [d]} q_i R_i^\top x$ and

$$p_2 = \frac{1}{1 + \exp(-\sum_{i \in [d]} q_i R_i^\top x)}. \tag{201}$$

Then, the gradient of $p_2$ w.r.t. $R_k$ is then given by

$$\nabla_{R_k} p_2 = p_2 (1 - p_2) \cdot q_i x, \tag{202}$$

hence, more succinctly, the gradient w.r.t. $R$ is

$$\nabla_R p_2 = p_2 (1 - p_2) \cdot x q^\top. \tag{203}$$

Hence,

$$\nabla_R \mathsf{regret}(R, f \mid P_x) = \mathbb{E}\left[ \frac{\partial}{\partial p_2} D_{\mathsf{KL}}(p_1 \parallel p_2) \times \nabla_R p_2 \right] \tag{204}$$

$$= \mathbb{E}\left[ (p_2 - p_1) \cdot x q^\top \right] \tag{205}$$

$$= \mathbb{E}\left[ \left( \frac{1}{1 + \exp(-q^\top R^\top x)} - \frac{1}{1 + \exp(-f^\top x)} \right) \cdot x q^\top \right]. \tag{206}$$

**Initialization** Here the initialization is similar to the linear MSE setting, except that since a column of the representation cannot ideally capture even a single adversarial function, the initialization algorithm only searches for a single adversarial function ($\tilde{m} = 1$). This single function is then used to produce $R^{(1)}$ as the initialization of Algorithm 1.

**Algorithm parameters** The algorithm parameters used for Example 8 are shown in Table 3.

### H.3 DETAILS FOR EXAMPLE 9: AN EXPERIMENT WITH A MULTI-LABEL CLASSIFICATION OF IMAGES AND A COMPARISON TO PCA

We next present the setting of Example 9, which shows that large reduction in the representation dimension can be obtained if the function is known to belong to a finite class.

**Definition 23** (The multi-label classification setting). Assume that $\mathcal{X} = \mathbb{R}^{\sqrt{d} \times \sqrt{d}}$ where $d = 625$, and $x$ represents an image. The distribution $P_x$ is such that $x$ contains 4 shapes selected from a dictionary of 6 shapes in different locations, chosen with a uniform probability; see Figure 6. The output is a binary classification $\mathcal{Y} = \{\pm 1\}$ of the image. Assume that the class of representation is linear $z = R(x) = R^\top x$ for some $R \in \mathcal{R} := \mathbb{R}^{d \times r}$ where $d > r$. The response function belongs to a class of 6 different functions $\mathcal{F} = \{f_1, \ldots f_6\}$, where $f_j : \mathcal{X} \to \mathcal{Y}$ indicates whether the $i$th shape appears in the image or not. Assume the cross-entropy loss function, where given that the prediction that $y = 1$ with probability $q$ results the loss $\mathsf{loss}(y, q) := -\frac{1}{2}(1 + y) \log q - \frac{1}{2}(1 - y) \log(1 - q)$. The set of predictor functions is $\mathcal{Q} := \{Q(z) = 1/[1 + \exp(-q^\top z)], \ q \in \mathbb{R}^r\}$, and the regret is then given by the expected binary Kullback-Leibler (KL) divergence as in Definition 7.

**Simplifying Algorithm 1** In the the multi-label classification setting of Definition 23, Algorithm 1 can be simplified as follows. First, since the number of response functions in the class $\mathcal{F}$ is finite, the Phase 1 problem (22) in Algorithm 1 algorithm is simple, since the adversarial function can

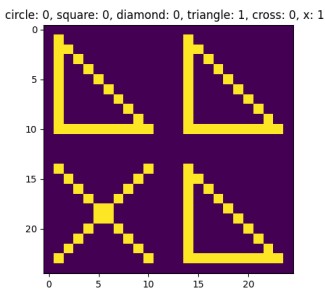

Figure 6: An image in the dataset for the multi-label classification setting (Definition 23).

be found by a simple maximization over the 6 functions. Then, the phase 2 step simply finds for each function $f^{(j_2)}$ in $\mathcal{F}$, $j_2 \in [6]$, the best representation-predictor $(R^{(j_1)}, Q^{(j_1, j_2)})$ using gradient descent, where $R^{(j_1)} : \mathcal{X} \to \mathbb{R}^r$ is a linear representation $z =$, and $Q$ is logistic regression. This results is a payoff matrix of $\mathbb{R}^{6 \times 6}$. Then, the resulting game can be numerically solved as a linear program, thus obtaining the probability that each representation should be played. The resulting loss of this minimax rule $\boldsymbol{R}^*$ is the loss of our representation. This representation is then compared with a standard PCA representation, which uses the projections on the first $r$ principle directions of $R = V_{1:r}(\Sigma_{\boldsymbol{x}})^\top$ as the representation (without randomization). The results of the experiment are shown in Figure 3 in the paper.

### H.4   AN EXPERIMENT WITH A NN ARCHITECTURE

In the analysis and the experiments above we have considered basic linear functions. As mentioned, since the operation of Algorithm 1 only depends on the gradients of the loss function, it can be easily generalized to representations, response functions and predictors for which such gradients (or subgradients) can be provided. In this section, we exemplify this idea with a simple NN architecture. For $x \in \mathbb{R}^d$, we let the rectifier linear unit (ReLU) be denoted as $(x)_+$.

**Definition 24** (The NN setting). Assume the same setting as in Definitions 1 and 7, except that the class of representation, response and predictors are NN with $c$ hidden layers of sizes $h_R, h_f, h_q \in \mathbb{N}_+$, respectively, instead of linear functions. Specifically: (1) The representation is

$$R(x) = R_c^\top \left( \cdots \left( R_1^\top (R_0^\top x)_+ \right)_+ \right)_+ \tag{207}$$

for some $(R_0, R_1, \cdots R_c) \in \mathcal{R} := \{\mathbb{R}^{d \times h_R} \times \mathbb{R}^{h_R \times h_R} \cdots \mathbb{R}^{h_R \times h_R} \times \mathbb{R}^{h_R \times r}\}$ where $d > r$. (2) The response is determined by

$$f(x) = f_c^\top \left( \cdots \left( F_1^\top (F_0^\top x)_+ \right)_+ \right)_+ \tag{208}$$

where $(F_0, F_1, \ldots, f_c) \in \mathcal{F} := \{\mathbb{R}^{d \times h_f} \times \mathbb{R}^{h_f \times h_f} \cdots \mathbb{R}^{h_f \times h_f} \times \mathbb{R}^{h_f}\}$. (3) The predictor is determined by for some

$$q(z) = q_c^\top \left( \cdots \left( Q_1^\top (Q_0^\top z)_+ \right)_+ \right)_+ \tag{209}$$

where $(Q_0, Q_1, \ldots, q_c) \in \mathcal{Q} := \{\mathbb{R}^{r \times h_q} \times \mathbb{R}^{h_q \times h_q} \cdots \mathbb{R}^{h_q \times h_q} \times \mathbb{R}^{h_q}\}$.

**Regret gradients**   Gradients were computed using `PyTorch` with standard gradients computation using backpropagation for an SGD optimizer.

**Initialization**   The initialization algorithm is similar to the initialization algorithm used in the linear cross-entropy setting.

**Algorithm parameters**   The algorithm parameters used for the example are shown in Table 4.

**Results**   For a single hidden layer, Figure 7 shows the reduction of the regret with the iteration for the cross-entropy loss.

Table 4: Parameters for the NN cross-entropy setting.

| Parameter | $c$ | $h_R$ | $h_f$ | $h_q$ | |
| --- | --- | --- | --- | --- | --- |
| Value | 1 | $d$ | $d$ | $d$ | |
| Parameter | $\beta_r$ | $\beta_f$ | $\eta_r$ | $\eta_f$ | $\eta_q$ |
| Value | 0.9 | 0.9 | $10^{-3}$ | $10^{-1}$ | $10^{-1}$ |
| Parameter | $T_R$ | $T_f$ | $T_Q$ | $T_{\text{avg}}$ | $T_{\text{stop}}$ |
| Value | 100 | 1000 | 100 | 10 | 80 |

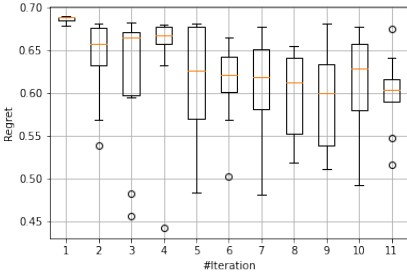

Figure 7: The regret achieved by Algorithm 1 in the NN cross-entropy setting as a function of the iteration $m$.

