# OpenReview forum: "A representation-learning game for classes of prediction tasks"
_ICLR.cc/2024/Conference — ICLR 2024 poster_

### Official Review · Reviewer_eNwZ · 2023-10-22

**Soundness:** 3 good
**Presentation:** 3 good
**Contribution:** 3 good
**Rating:** 8
**Confidence:** 3

**Summary:**

The submission proposed a two-person game to solve the representation learning problem. It targets on a given class of tasks. A theoretical justification was made for the linear setting. A series of discussions were provided following the proposed algorithm for the general setting.

**Strengths:**

a) The submission proposed a game to solve the representation learning for a given class of tasks. The formulation is practical as one can control the relevant tasks.
b) The submission provided theoretical justifications for the linear setting to characterize the learned representations and the performance bounds (Theorems 2 and 3).

**Weaknesses:**

c) An algorithm is proposed to learn the representation for the general case. However, the evidence, from the theoretical or the practical perspective, is insufficient to determine the applicability of the proposed method.

**Questions:**

d) The analysis for the linear case seems to be adapted from the game theory. Could you please clarify the technical contribution, if any, in the analysis?
e) Is it possible to show the game will reach some situation, such as an equilibrium? Representations from the equilibrium may represent a negotiated outcome from both players.
f) Although Examples 6 -- 9 help us to understand the nature of Algorithm 1, one still cannot know Algorithm 1's applicability. Would it be better to compare Algorithm 1 with baselines to show the representations output from Algorithm 1 are better than the existing methods?

---

> ### Author Response · Authors · 2023-11-14
> **Answers to comments**
>
> # Answer to c)
>
> Our main goal here is to propose a principled framework for learning representations based on partial knowledge of downstream prediction tasks. We aimed for analytical exploration of its fundamental properties, and thus focused on analysis of the linear-MSE setting, specifically Theorem 3. We believe that this provides a strong evidence for the validity of the framework. The proof of this result is rather convoluted (see our answer to d) below), and thus is also a main contribution of the paper. Algorithm 1 was developed to show that the framework is viable beyond the linear case, even in general cases (loss function, representation rules). Following this approach, we have focused the paper on the core of the framework and basic settings, rather than on practical baselines. Going beyond basic settings inherently requires some domain-knowledge, e.g., determining the class of functions $\{\cal F\}$, and we believe that this deserves a separate inquiry. We hope that this would be achieved in future research, both by us, and by experts in various domains. We note, however, that the incremental algorithmic approach we take follows a widely acceptable solution in various ML problems, as we discuss in depth on page 2 (“Algorithmic contribution”), and in Appendix B, pages 18-19 (“Game-theoretic formulations in statistics and machine-learning” and “Incremental learning of mixture models”).
>
> # Answer to d)
>
> The main part of the analysis in the linear case is the proof of Theorem 3 (which is rather long, about 7 pages). Its goal is to find the optimal solution for the representation in mixed strategies. The main technical difficulty is that solving the mixed minimax problem directly, that is, solving
>
> $\min\_{\boldsymbol{R}}\max_{f\in{{\mathcal F}_{S}}} \mathbb{E}\left[\mathsf{regret}(\boldsymbol{R},f \mid \Sigma\_{\boldsymbol{x}})\right]$
>
> is intractable. However, the minimax theorem from game theory implies that its value equals the maximin problem, that is, to
> $$\max_{\boldsymbol{f}}\min_{\boldsymbol{R}}\mathbb{E}\left[\mathsf{regret}(\boldsymbol{R},f\mid\Sigma_{\boldsymbol{x}})\right]$$
> where now the function can be random too. This problem is still not trivial, but we managed to solve it. This resulted $\boldsymbol{f}^{\*}$, the function part of the saddle-point $(\boldsymbol{R}^{\*},\boldsymbol{f}^{\*})$, which solves both the minimax and maximim problems. We then exploited the maximin solution as a certificate to find the optimal representation $\boldsymbol{R}^{\*}$, because the minimax value is never smaller than the maxinim value. Thus, if one finds a random representation for which
>
> $\max_{f\in{\cal {\cal F}_{S}}}\mathbb{E}\left[\mathsf{regret}(\boldsymbol{R},f\mid\Sigma\_{\boldsymbol{x}})\right]$
>
> equals the maxinim value, this guarantees that this random representation is the sought $\boldsymbol{R}^{\*}$. We find this by first making an educated guess regarding a simple structure that $\boldsymbol{R}^{\*}$ should posses. We then optimized over this structure, and shown it is reduced to solving an equation $Ap=b$. We proved that any solution to this equation leads to a representation whose regret is no larger than the maximin value, and is thus $\boldsymbol{R}^{\*}$, the saddle-point representation. To complete the proof, we show that a solution to $Ap=b$ must exist, and that the support of this solution is at most $\ell^{*}+1$. This leads to an optimal random representation $\boldsymbol{R}^{\*}$ that is drastically simpler than a general probability distribution over $\mathbb{R}^{d\times r}$, and reinforces the effectiveness of the approach.
>
> # Answer to e)
>
> As follows from our answer to d), the obtained solution $(\boldsymbol{f}^{\*},\boldsymbol{R}^{\*})$ is exactly the equilibrium of the game. If both sides agree to use this solution, no side can improve its payoff by a one-sided change of his/her strategy.
>
> # Answer to f)
>
> As we have answered to point c), we focused this paper on the general approach, theoretical guarantees on fundamental settings, and a general algorithm. However, even though it is just a basic setting, in Example 9 we compare our method to PCA and show a significant improvement. PCA is also clearly not the state-of-the-art, but it is a classic and principled method that enables initial comparisons, and evidence of the potential of the method.
>
> In summary, our main contributions are a general framework, an analytic solution in basic setting (linear-MSE) and its proof, and a “meta-algorithm” which shows the viability of the method in the general cases. Hopefully, applications will be explored in depth in the future. In light of the above clarification, we would appreciate if you could re-evaluate our contributions, and finally thank you for your effort in reviewing the paper.

---

> > ### Comment · Reviewer_eNwZ · 2023-11-15
> > **Thank you for your reply.**
> >
> > I believe the submission aims at answering an important problem: "Is there a generic strategy to learn good representations, or is problem-by-problem representation learning unavoidable?" But, my understanding of the submission and your reply tells me, "You have a general methodology, but only compare the performance with PCA." If the statement is correct, it cannot serve as a legitimate answer to the topmost question since a generic methodology outputting suboptimal representations might not be able to replace a problem-tailored representation learner.
> >
> > [Question g)] Given the current submission, can you provide evidence that it at least produces good representations for a set of problems? The evidence could be proofs of convergence rates of a wide range of problems or experimental outcomes compared with existing methods. The current result (Theorem 3) characterizes the performance via regret, which I don't know how to use to compare with existing methods. [Question h)] Could you please elaborate on how we can compare methods (the proposed and the existing ones) fairly? I really do appreciate your devotion to this project and the non-trivial results provided. But maybe representation learning cannot be analyzed only mathematically; its effects must be tested. Maybe it is just not as simple as proving a convergence rate. I won't say my perspective is correct; I welcome better viewpoints that could provide fair means to justify a representation learning method.
> >
> > Also, thank you for pointing out the technical contributions for the proofs. [Question i)] Are they of independent interests, or are they just tailored for the specific goal?

---

> > > ### Author Response · Authors · 2023-11-16
> > > **Additional answers**
> > >
> > > Thank you for your response!
> > > # Answer to the comment and question g)
> > > The paper roughly follows the question you stated, but we feel that the description that follows it is inaccurate, so we kindly ask to clarify. Our main motivation is compression, in which unlabeled feature vectors $x$ of high dimension $d$ are collected, but should be compressed to vectors $z$ of much smaller dimension $r$. The aim is that labels $y$ will be collected in the future for each $z$, and a predictor from $z$ to $y$ will be learned. Obviously, predicting from $z$ is less informative compared to predicting from $x$. The question is thus how to compress $x$ while maximizing future prediction capabilities.
> > >
> > > Let us consider two extreme cases. First, nothing informative is known about the subsequent prediction task. This calls for unsupervised learning methods, in which $z$ is chosen to allow an accurate reconstruction of $x$ from $z$. Such methods aim to look for “directions” in which the features have significant variance. In PCA these are standard directions in $\mathbb{R}^{d}$, and this is generalized by advanced non-linear methods, such as auto-encoders, to sort of non-linear directions. Second, suppose that the prediction task $y=f(x)$ is actually known or labels are given, before $x$ is compressed to $z$ (here f is random). This is an easy case since $z$ will be optimized for the given function $f$.
> > >
> > > We aim to bridge between these two extreme cases: How to effectively compress unlabeled $x$, when there is a *partial* knowledge on future prediction tasks, but neither exact knowledge nor labeled data? We assume that this knowledge is given to the learner in the form of a class of possible (random) functions ${\cal F}$ from $x$ to $y$. The representation $z$ thus should be chosen to uniformly allow efficient prediction from $z$ for all functions in ${\cal F}$.
> > >
> > > Our method assumes that the representation is also chosen from a class of rules, e.g., linear, auto-encoder, etc., and the goal is to optimize the representation within this class, when tailored to the function class ${\cal F}$. This is a relative criterion. If the class is linear the criterion extracts the best linear representation. If a more complicated class of representations is considered, we will get the best within that class. Our method provides a principled way to achieve that.
> > >
> > > Accordingly, performance is measured by the regret. To explain this, consider, for example, a linear setting $y=f^{\top}x+\epsilon$. A learner that uses $x$ directly, will have a prediction loss given by the noise variance $\mathbb{E}[\epsilon^{2}]$. If, instead of $x$, the learner must use a compressed version $z$, this will increase the minimal possible loss. The regret measures this increase, and if the regret is $v$ then it is assured that the loss in any future prediction tasks $f\in{\cal F}$ based on the representation $z$ will have a loss of at most $v+\mathbb{E}[\epsilon^{2}]$.
> > >
> > > # Answer to question h)
> > >
> > > Practical use of the method could be as follows. Suppose that a representation architecture is efficient in some prediction application (e.g., a specific form of auto-encoder). This defines a class of possible representations, determined by the architecture weights. We would like to use this architecture when collecting new data, when the weights are chosen before labels are collected, so they cannot be fully optimized for specific prediction problem. For example, suppose that very detailed medical images should be compressed, only partially knowing what they will be used to diagnose in the future. Our representation can be compared with a representation that does not use this information. Indeed, Example 9 considers a class of prediction problems, and assume a linear representation. A naive choice of a linear representation is PCA, but we show that our optimized representation is much better, which is expected as it was optimized for the class of prediction problems. We agree that the next step should be to consider more complicated representation rules. We have a preliminary experiment in Section H.4 that uses a NN architecture, and are eager to expand this in the future to practical applications. However, this will depend on domain knowledge to determine a practically relevant class ${\cal F}$, and admittedly, some computational compromises in the instantiation of the algorithm. We thus believe that this paper should focus on a clean presentation of the method, in its most broadest form.
> > >
> > > # Answer to question i)
> > > The proof has a few components of independent interest: The idea that a uniform guarantee (worst case, minimax solution) can be obtained by solving a maximin problem was widely used in classic statistics, but we believe is less prevalent in modern ML. Also, the fact that the optimal randomization requires a rather small number of atoms is crucial for implementation, and our argument that proves this could be used for other problems.

---

> > > > ### Comment · Reviewer_eNwZ · 2023-11-21
> > > > **Thank you for clarifying**
> > > >
> > > > From a mathematical perspective, the submission did a good job of proposing a model pretraining methodology and characterizing the performance. IMHO, practical applications are still better places for testing the effectiveness of a representation learning method than math. I will revise my score, but it is still subject to change during the discussion phase. Thank you.

---

> > > > > ### Author Response · Authors · 2023-11-21
> > > > > **Thank you**
> > > > >
> > > > > We agree that practical applications are of prime importance, and hope that this submission will spur interest in this direction.
> > > > > We appreciate your responsiveness and the time devoted for the review!

---

### Official Review · Reviewer_yB5J · 2023-10-31

**Soundness:** 3 good
**Presentation:** 3 good
**Contribution:** 3 good
**Rating:** 8
**Confidence:** 2

**Summary:**

They propose a game-based methodology to learn the classical machine learning problem of dimensionaity-reducing representations of feature vectors. On the machine learning they consider the linear setting and use the mean square error loss function. On the game theoretic side, they used mixed and pure strategies.

**Strengths:**

They propose an interesting way to study a very classical problem. The paper is well-written, and clear

**Weaknesses:**

The computation can be complex such as in solving Ap = b for the probability p in the mixed minmax strategy.

**Questions:**

The goal of RL is the minimize regret, while the goal of this seems to be to minimize MSE loss. There are some results relating to regret, but how does this relate to the MSE that results in applying this method?

Also for Theorem 3, does this assume there are only finitely many possible feature vectors (or does it just assume that the dimension of the feature vectors is finite). Just to check my understanding, I assume the dimension of the covariance matrix matches the dimension of the feature vectors learned correct?

---

> ### Author Response · Authors · 2023-11-14
> **Answers to comments**
>
> # **Response to the weakness**
> As we mention in Remark 1 on page 5, the
> direct solution of $Ap=b$ is infeasible in high dimensions, since
> it requires solving a linear program with very large dimension ${\ell^{\*} \choose r}=\Theta((\ell^{*})^{r})$.
> In the paper, we mentioned that this can be circumvented by using
> our general algorithm to solve the linear case. This approach is indeed
> effective, as shown in Example 6 on page 8. However, we actually developed
> a computationally effective algorithm to directly solve $Ap=b$, which
> allows to use the closed-form solution, but haven't included this algorithm
> in the paper in order not to burden the reader. It is reminiscent
> of the ``basis pursuit'' algorithm, and operates as follows: Recall
> that $A$ has $\ell^{\*}$ rows and ${\ell^{\*} \choose r}$ columns.
> Theory guarantees $Ap=b$ has a solution $p$ that has less than $\ell^{\*}+1$
> non-zero entries. Thus, the algorithm needs to find the support of
> this solution, that is, $\ell^{\*}+1$ column indices out of the possible
> ${\ell^{\*} \choose r}$ columns of $A$ with non-zero weights. This
> can be done iteratively. First, we begin with an initial guess of
> $\ell^{\*}+1$ columns. It could be arbitrary, but as a general rule-of-thumb
> it is better to choose columns with $\ell^{\*}-r$ ones on the larger
> eigenvalues (and zeros for other columns). Let us denote the sub-matrix
> of $A$ comprised from these $\ell^{\*}+1$ columns by $A_{1}$, and
> the vector $p$ reduced to these indices by $p_{1}$. Then, a solution
> to $A_{1}p_{1}=b$ is sought that only uses this support. Specifically,
> since an exact solution perhaps cannot be found, we instead solve
> $\min_{p_{1}}\||A_{1}p_{1}-b\||\_{1}$, which is a linear program.
> Let the solution be $p_{1}^{\*}$. Then, the error is given by $e_{1}=A_{1}p_{1}^{\*}-b$.
> If $e_{1}=0$ then this means that we have found a solution to $Ap=b$
> with a support of at most $\ell^{\*}+1$, as theory predicts. Otherwise,
> the error $e_{1}$ is rounded towards a column of $A$. Specifically,
> we find the $\ell^{\*}-r$ coordinates of $e_{1}$ with maximal value,
> and choose a column of $A$ with ones on these columns (and zero otherwise).
> This new column is added to $A_{1}$ to create $A_{2}$. Next, we
> similarly solve $\min_{p_{2}}\||A_{2}p_{2}-b\||\_{1}$ and obtain error $e_{2}$.
> If $e_{2}=0$ that this means we have solved $Ap=b$ with $\ell^{\*}+2$
> columns, which is sub-optimal, but still rather effective. We continue
> to more such iterations, adding columns to $A_{i}$ as necessary,
> one column at a time, until $e_{i}=0$. If the number of iterations
> is not too large, then this only requires solving linear programs
> of dimensions much smaller than ${\ell^{\*} \choose r}$. As an example,
> in case $d=100$, $r=20$, and the eigenvalues are $\lambda_{i}\propto1/\sqrt{i}$
> it turns out that $\ell^{\*}=61$. While ${\ell^{\*} \choose r}={61 \choose 20}=6\cdot10^{15}$
> is huge, this method finds a solution with $120\approx2\ell^{\*}$
> columns, and thus feasible, and also rather effective.
>
> #  **Response to the questions**
>
> * The regret and the MSE (or more generally, the loss) are tightly related.
> For example, in the linear-MSE setting, the regret is the MSE of the
> proposed solution, minus the noise variance. The variance of the noise
> represents an inevitable loss that cannot be reduced by any representation.
> It is thus subtracted from the MSE in order to focus on the part of
> the MSE that can be reduced by optimization of the representation.
> * In Theorem 3, the feature vector is $\boldsymbol{x}\in\mathbb{R}^{d}$
> with $d<\infty$ and its covariance matrix is $\Sigma_{\boldsymbol{x}}\in\mathbb{R}^{d\times d}$,
> so it is a finite-dimensional continues vector, which takes infinite
> possible values. A pure representation of $\boldsymbol{x}$ is a matrix
> $R\in\mathbb{R}^{d\times r}$ so that the representation is $\boldsymbol{z}=R^{\top}\boldsymbol{x}\in\mathbb{R}^{r}$,
> which is also a continues vector. A mixed representation allows to
> choose the matrix $R$ randomly. In Theorem 3, we show that this randomization
> may be supported on a finite set of matrices $\\{R_{1},R_{2},\ldots,R_{\ell^{*}+1}\\}$
> so that the representation is $\boldsymbol{z}=R_{j}^{\top}\boldsymbol{x}$
> with probability $p_{j}$. The theorem provides the optimal choice
> of set of matrices and the probabilities $\\{p_{j}\\}$.
>
> We hope that this clarifies your questions, and will be happy to take further ones. We thank you for your effort in evaluating the paper
> and the positive assessment.

---

### Official Review · Reviewer_HEJP · 2023-11-01

**Soundness:** 3 good
**Presentation:** 2 fair
**Contribution:** 4 excellent
**Rating:** 6
**Confidence:** 2

**Summary:**

The authors propose a new game theoretic framework for learning low-dimensional representations $z= R(x) \in \mathbb{R}^r$ of unlabelled data $\\{\vec{x}_i\\} \subset \mathbb{R}^d$ (where $r \ll d$) in such a way that the learned representation would be useful for a variety of downstream learning tasks (specified by a class $\mathcal{F}$ of response functions).

The contributions are both theoretical (optimal linear representations for linear response etc) and practical/algorithmic (more general representations and response functions). This paper uses a different form of game than the one used by (Dubois et al 2020).
1. In (Dubois et al 2020), Player 1 chooses the learning task (input distribution, response function) and the score function. Then Player 2 (knowing the above) trains a representation for the input data. Finally, Player 1 evaluates the representation, using the chosen score function on an ERM classifier trained using the representation given by Player 2 (with IID data ~ (input distribution, response function)).

2. In this paper the game is a 2PZS game with: (i) Player 1 (representation player) chooses the representation mapping $R \in \mathcal{R}$ (ii) Player 2 chooses the response function $f \in \mathcal{F}$, where the data distribution ($P_{\mathbf{x}}$), loss function ($\mathsf{loss}$), and the class of prediction rules ($\mathcal{Q}$) are fixed. The payoff is given by
$$\mathsf{Payoff}(R,f) = \mathsf{Regret}(R, f | P_{\mathbf{x}}, \mathsf{loss}, \mathcal{Q}) = \min_{Q \in \mathcal{Q} \text{ on } \mathbb{R}^r} \mathbb{E}[\mathsf{loss}(f(\mathbf{x}),Q(R(\mathbf{x})) )] - \min_{Q \in \mathcal{Q} \text{ on } \mathbb{R}^d} \mathbb{E}[\mathsf{loss}(f(\mathbf{x}),Q(\mathbf{x}) )]$$

The game in this paper interchanges the order of the players (in a still useful way) and also abstracts the evaluation of the representation in order to make use of the 2PZS/saddle-point framework.

They then consider the minimax and maximin regret in terms of mixed strategies, which are equal due to the minimax theorem. The former is given by $$\min_{\mathcal{D}^{\mathrm{rep}} \in \text{distributions on } \mathcal{R}} \max_{f \in \mathcal{F}} \mathbb{E}_{R \sim \mathcal{D}^{\mathrm{rep}}} \mathsf{Regret}(R,f)$$

whereas the latter is given by
$$\max_{\mathcal{D}^{\mathrm{fn}} \in \text{distributions on } \mathcal{F}} \min_{R \in \mathcal{R}} \mathbb{E}_{f \sim \mathcal{D}^{\mathrm{fn}}} \mathsf{Regret}(R,f)$$

with the goal of characterizing the optimal minimax (= maximin) regret as well as the optimal $\mathcal{D}^{\mathrm{fn}}$ (parametrized) and as well as the optimal $\mathcal{D}^{\mathrm{rep}}$ (parametrized) that lead to the optimal regret. They are able to do this in the linear MSE case. In the general case, they propose an algorithm to find distributions (mixtures) over finitely many functions and representations, with no theoretical guarantees.

They also consider the minimax regret in pure strategies, given by
$$\min_{R \in \mathcal{R}} \max_{f \in \mathcal{F}} \mathsf{Regret}(R,f)$$
with the goal of finding an optimal saddle point representation $(R^\ast, f^\ast)$ in the linear MSE case.

The results in the paper are as follows:

**The linear MSE setting:** Where the data $\mathbf{x}$ is non-degenerate with zero-mean and covariance $\Sigma_{\mathbf{x}}$, the representations are linear ($R(\mathbf{x}) = R^\top \mathbf{x}$ for $R \in \mathbb{R}^{d \times r}$), the response functions are linear (response class is $\mathcal{F}_S$, consisting of functions of the form $f(\mathbf{x}) = f^\top \mathbf{x} + \varepsilon$, where $\varepsilon$ is heteroskedastic, mean-zero, noise, and the coefficient vector $f \in \mathbb{R}^d$ lies in the ellipsoid given by a positive definite matrix $S$), the loss function is mean squared error (MSE), and the prediction functions are also linear ($Q(\mathbf{x}) = q^\top \mathbf{x}$).

* Here they characterize the minimax pure regret, as well as the optimal saddle point pair $(R^\ast, f^\ast)$ giving the minimax regret w.r.t pure strategies (Theorem 2). Interestingly, the optimal representation involves _whitening_ the input vector and then projecting the result on the _top-$r$ eigenvectors_ of the $S$-adjusted version of the data covariance $\Sigma_{\mathbf{x}}$.
* They also characterize the minimax/maximin mixed regret, as well as the optimal mixed strategies $(\mathcal{D}^{\mathrm{fn}}, \mathcal{D}^{\mathrm{rep}})$ leading to this regret (Theorem 3).

The characterization results also generalize to infinite dimensional feature spaces $\mathcal{X}$ (rather than $\mathbb{R}^d$) with some more assumptions (independent noise). This is only done in Appendix F.

**General case**: Here they do implicitly assume some finite dimensional, differentiable, representation for the representations and response functions such that the saddle point problem giving minimax regret can be approximately-solved using an iterative procedure (proposed). The algorithm is motivated by the application of some theoretically-applicable concepts (for saddle point problems) --- iteratively adding to (and hopefully improving) the sets of representations and the response functions alternately,  using projected gradient descent on the regret w.r.t one (keeping the other set fixed), and then adjusting the weights assigned to the various representations and functions in the currently explored set using MWU (as in Freund and Schapire's adaptive game playing framework). However, there are understandably no theoretical guarantees, since the saddle-point problem involved is not convex-concave in general.

**Strengths:**

* The game-theoretic framework proposed by the paper is very interesting and is novel (in terms of application to representation learning) to the best of my knowledge.
* The results in the linear MSE case are very precise and complete. They give the intuitive expected results when the response class is $\mathcal{F}_S, S=\\{I_d\\}$ = unit-norm linear functions (Example 5) and when the response class is $\mathcal{F} = \\{I_d\\}$ = identity function (Appendix E).
* The linear MSE results also seem to have some interesting consequences (depending on the structure and correlations of $S^{-1}$ and $\Sigma_{\mathbf{x}}^{-1}$) in general (interpreting the entries as some form of feature importance/correlation weights in the input data and in the prediction task respectively), which are very novel to the best of my knowledge, and could well be exploited in other works.
* The representation learning algorithm for the general case is a substantial contribution which is well-motivated using existing theoretical frameworks, even in the understandable absence of theoretical guarantees.
* The examples in the main paper are very useful in conveying the gist of the ideas.
* The experiments, while limited, are well-thought-out (validating the general algorithm in the linear MSE case, comparison with standard PCA etc).

**Weaknesses:**

* The experiments are very limited given the scope of the paper ("learning good representations for general prediction tasks", as it may said-to-be). The experiments do not compare to any state-of-the-art practical methods for learning representations at all, especially when applying the representations to different tasks (which I feel is important given the lack of theoretical guarantees for the general case algorithm). However, this paper may be viewed as introducing a _novel framework_ for representation learning/evaluation that links it to saddle-point-game theory (in a way that prior works do not) and hence make it possible for any advances in solving hard saddle-point-games to lead to better approaches for learning representations.
* The meat of the paper is mostly in what is effectively the supplementary material (after the references). The authors have albeit put substantial effort in trying to condense the ideas involved in the proofs as well as illustrate using good examples in the main section. However, it seems to be a losing battle, and any useful perusal of this paper must involve substantial parts of the supplementary.

**Questions:**

None

---

> ### Author Response · Authors · 2023-11-14
> **Answers to comments**
>
> Our answers to the mentioned weaknesses are as follows:
>
> * Indeed, our main goal in this paper was to introduce this framework and explore its basic principles. We consider our proposed algorithm as a meta-algorithm, which demonstrates that the proposed approach is not limited to the linear setting, and agree that it requires extensive experimentation in applications. We have some ideas for such applications in mind, for example, using prior knowledge on region of interest in computer vision, or using prior knowledge on non-coding DNA in genomics. However, this requires domain-knowledge in determining the class of functions ${\cal F}$, and various other details. This will divert the paper to these applications, rather than to the broad scope of representation learning. We hope that extensive experimentation would be achieved in future research, both by us, and by experts in various domains.
>
> * Our approach in writing the paper was to first present the general framework and its importance; second, explore it theoretically in basic linear settings; and third, propose an efficient algorithmic for the general case, which cannot be covered by theoretical analysis. We believed that all these components are vital, and thus deferred the technical aspects to the appendix, leaving them to in-depth readers. We understand that this comes at a cost, and we will re-inspect the paper from this aspect. We would be happy to take any suggestions for improvement (e.g., what important points are missing in the body of the paper), and will incorporate them into an updated version.
>
> Thank you for a careful reading of the paper, a detailed summary, and a positive assessment.

---

### Official Review · Reviewer_crzZ · 2023-11-06

**Soundness:** 3 good
**Presentation:** 3 good
**Contribution:** 3 good
**Rating:** 8
**Confidence:** 4

**Summary:**

The paper considers the online prediction problem of predicting features for learning functions. First, the paper focuses on the problem of online linear regression with reduced dimensionality. Then, it considers more general settings, including mixed representations or/and logistic regression. Finally some preliminary experimental results are shown.

**Strengths:**

The problem is well-motivated and suited for the conference. The problem formulations are reasonable. Although the first problem might look somewhat elementary, the theoretical results are solid.

**Weaknesses:**

So far the current results only focus on simple cases where the classifiers are linear. Ideally, some attempts to cope with nonlinear classifiers or nonlinear feature mapping would be appreciated.

**Questions:**

Is it possible to extend the first result (linear regression) to kernelized classifiers? If not, can you explain why?

---

> ### Author Response · Authors · 2023-11-14
> **Answers to comments**
>
> * **Response to the weakness**: Our proposed Algorithm 1 is applicable to any differential representation, not just linear. This algorithm alternates between optimizing an adversarial function (phase 1) and a new representation rule (phase 2). We have not focused on this in the body of the paper, but the optimization of the new representation rule in phase 2 is discussed in Appendix G.2 in detail. Algorithm 3 proposed there for the phase 2 optimization of the representation is based on computing gradients of the loss with respect to the representation rule (line 20). Therefore, the algorithm is applicable as long as these gradients can be computed (sub-gradients may also suffice). In Appendix H.4 we conducted a preliminary experiment with a one hidden-layer NN architecture, and showed the improvement of the regret. In future research we plan to explore our algorithm for non-linear representations in depth.
>
> * **Response to the question**: Yes, extension to representations that are based on infinite-dimensional features is possible. In Appendix F, we provide one possible such extension, to a Hilbert space setting (this is mentioned on page 5, in the remark before Example 4). In this setting, the original feature vector $x\in\mathbb{R}^{d}$ is mapped to a vector $R(x)=(\psi_{1}(x),\ldots,\psi_{r}(x))^{\top}\in\mathbb{R}^{r}$ where $\\{\psi_{i}(x)\\}$ are non-linear feature-maps, taken from a Hilbert space of such maps. Our theoretical results on pure and mixed representations are generalized to this setting in Theorems 19 and 20. One can also consider the opposite setting – a linear representation followed by a non-linear (kernelized) predictor. Generalizing the theoretical results to this case is interesting, but appears to be challenging. This is because it requires obtaining a closed-form expression for $\mathsf{regret}(R,f\mid P_{\boldsymbol{x}})$, as was obtained for the linear case in Lemma 16, Appendix E.2. In turn, this requires understanding how the non-linear feature map affects a linear representation $z=Rx$, which is difficult in general. In future research, it would be interesting to furnish conditions that allow to analyze such cases too. By the way, referring to the weakness above, this setting is another case in which the representation is non-linear.
>
> Thank you for your effort in evaluating the paper and the positive assessment.

---

### Author Response · Authors · 2023-11-23
**A general summary and outlook**

We have responded to all the comments made by the reviewers, which were overall positive.

We highlight here that we focused here on the introduction of the framework, on its analytical properties in the elementary linear-MSE setting, and on a general algorithmic approach. We believe that these findings could be of interest to a wide audience: To theoreticians, who can generalize our results to advanced, non-linear representations; To researches of iterative (online) minimax algorithms, who can further explore the link we introduced from representation learning to saddle-points of certain minimax games; To practitioners, who may improve data compression and dimensionality reduction techniques, and would face the challenge of specifying domain-specific classes of tasks, and perhaps find methods to automatically learn them from past experience.

Finally, our formulation bares an interesting relation to the pretraining approach of foundation models. Typically, such models are pre-trained and work well without any explicit prior assumption on downstream tasks. It is thus interesting to investigate if they _implicitly_ induce such assumptions, or whether incorporating such assumptions can improved the pretraining procedure.

We thank you and the reviewers for your effort in evaluating our paper.

---

### Meta-Review · Area_Chair_ro9r · 2023-12-06

**Metareview:**

This paper proposes a game-theoretic framework for learning an low-dimensional representation of feature vectors, characterized optimal linear representations in some cases, and illustrated the effectiveness of the proposed research in experiments.

Pros: the approach is quite novel and interesting

Cons: two reviewers (HEJP and eNwZ) are not completely convinced by the effectiveness of the proposed methods.

**Justification For Why Not Higher Score:**

While the overall scores are high, two reviewers (with scores 6 and 8) are completely convinced that the proposed method work. Their concerns were not addressed after the rebuttal and the discussion.

**Justification For Why Not Lower Score:**

All reviewers are positive about the paper and liked the novelty of the proposed approach.

---

### Decision · Program_Chairs · 2024-01-16

Accept (poster)